# Text embedding models yield detailed conceptual knowledge maps derived from short multiple-choice quizzes

**Paxton C. Fitzpatrick, Andrew C. Heusser** ⑩ **& Jeremy R. Manning** ⑩ ✉

Real-world conceptual knowledge is complex, multifaceted, and substantially over-simplified in most laboratory studies. Here we develop a mathematical framework, based on natural language processing models, for tracking and characterizing the acquisition of real-world conceptual knowledge. Our approach embeds each concept in a high-dimensional representation space where nearby coordinates reflect similar or related concepts. We test our approach using behavioral data from participants who answered small sets of multiple-choice quiz questions interleaved between watching two Khan Academy course videos. We apply our framework to the videos' transcripts and the text of the quiz questions to quantify the content of each moment of video and each quiz question. We use these embeddings, along with participants' quiz responses, to track how the learners' knowledge changed after watching each video and predict their success on individual quiz questions. Our findings show how a small set of quiz questions may be used to obtain rich and meaningful insights into what each learner knows, and how their knowledge changes over time as they learn.

Suppose that a teacher had access to a complete, tangible map of everything a student knows. Defining what such a map might even look like, let alone how it might be constructed or filled in, is itself a non-trivial problem. But if a teacher were to gain access to such a map, how might it change their ability to teach that student? Perhaps they might start by checking how well the student knows the to-be-learned information already, or how much they know about related concepts. For some students, they could potentially optimize their teaching efforts to maximize efficiency by focusing primarily on not-yet-known content. For other students (or other content areas), it might be more effective to optimize for direct connections between already known content and new material. Observing how the student's knowledge changed over time, in response to their teaching, could also help to guide the teacher towards the most effective strategy for that individual student.

A common approach to assessing a student's knowledge is to present them with a set of quiz questions, calculate the proportion they answer correctly, and provide them with feedback in the form of a simple numeric or letter grade. While such a grade can provide some indication of whether the student has mastered the to-be-learned material, any univariate measure of performance on a complex task sacrifices certain relevant information, risks conflating underlying factors, and so on. For example, consider the relative utility of the theoretical map described above that characterizes a student's knowledge in detail, versus a single annotation saying that the student answered 85% of their quiz questions correctly, or that they received a 'B'. Here we show that the same quiz data required to compute proportion-correct scores or letter grades can instead be used to obtain far more detailed insights into what a student knew at the time they took the quiz.

Designing and building procedures and tools for mapping out knowledge touches on deep questions about what it means to learn. For example, how do we acquire conceptual knowledge? Memorizing course lectures or textbook chapters by rote can lead to the superficial appearance of understanding the underlying content, but achieving true conceptual understanding seems to require something deeper and richer. Does conceptual understanding entail connecting newly

Department of Psychological and Brain Sciences, Dartmouth College, Hanover, NH, USA. ✉e-mail: Jeremy.R.Manning@Dartmouth.edu

acquired information to the scaffolding of one's existing knowledge or experience[1–6]? Or weaving a lecture's atomic elements (e.g., its component words) into a structured network that describes how those individual elements are related[7,8]? Conceptual understanding could also involve building a mental model that transcends the meanings of those individual atomic elements by reflecting the deeper meaning underlying the gestalt whole[9–12].

The difference between "understanding" and "memorizing," as framed by researchers in education, cognitive psychology, and cognitive neuroscience (e.g.,[10,11,13–15]), has profound analogs in the fields of natural language processing and natural language understanding. For example, considering the raw contents of a document (e.g., its constituent symbols, letters, and words) might provide some clues as to what the document is about, just as memorizing a passage might provide some ability to answer simple questions about it. However, text embedding models (e.g.,[16–23]) also attempt to capture the deeper meaning underlying those atomic elements. These models consider not only the co-occurrences of those elements within and across documents, but (in many cases) also patterns in how those elements appear across different scales (e.g., sentences, paragraphs, chapters, etc.), their temporal and grammatical properties, and other high-level characteristics of how they are used[24,25]. To be clear, this is not to say that text embedding models themselves are capable of "understanding" deep conceptual meaning in any traditional sense. But rather, their ability to capture the underlying structure of text documents beyond their surface-level contents provides a computational framework through which those documents' deeper conceptual meanings may be quantified, explored, and understood. According to these models, the deep conceptual meaning of a document may be captured by a feature vector in a high-dimensional representation space, wherein nearby vectors reflect conceptually related documents. A model that succeeds at capturing an analogue of "understanding" is able to assign nearby feature vectors to two conceptually related documents even when the specific words contained in those documents have limited overlap. In this way, "concepts" are defined implicitly by the model's geometry (e.g., how the embedding coordinate of a given word or document relates to the coordinates of other text embeddings[26]).

Given these insights, what form might a representation of the sum total of a person's knowledge take? First, we might require a means of systematically describing or representing (at least some subset of) the nearly infinite set of possible things a person could know. Second, we might want to account for potential associations between different concepts. For example, the concepts of "fish" and "water" might be associated in the sense that fish live in water. Third, knowledge may have a critical dependency structure, such that knowing about a particular concept might require first knowing about a set of other concepts. For example, understanding the concept of a fish swimming in water first requires understanding what fish and water are. Fourth, as we learn, our current state of knowledge should change accordingly. Learning new concepts should both update our characterizations of what is known and also unlock any now-satisfied dependencies of those newly learned concepts so that they are considered available for future learning.

Here we develop a framework for modeling how conceptual knowledge is acquired during learning. The central idea behind our framework is to use text embedding models to define the coordinate systems of two maps: a "knowledge map" that describes the extent to which each concept is currently known, and a "learning map" that describes changes in knowledge over time. Each location on these maps represents a single concept, and the maps' geometries are defined such that related concepts are located nearby in space. We use this framework to analyze and interpret behavioral data collected from an experiment that had participants answer sets of multiple-choice questions about a series of recorded course lectures.

Our primary research goal is to advance our understanding of what it means to acquire deep, real-world conceptual knowledge. Traditional laboratory approaches to studying learning and memory (e.g., list-learning studies) often draw little distinction between memorization and understanding. Instead, these studies typically focus on whether information is effectively encoded or retrieved, rather than whether the information is understood. Approaches to studying conceptual learning, such as category learning experiments, can begin to investigate the distinction between memorization and understanding, often by training participants to distinguish arbitrary or random features in otherwise meaningless categorized stimuli[27–32]. However, the objective of real-world training, or learning from life experiences more generally, is often to develop new knowledge that may be applied in useful ways in the future. In this sense, the gap between modern learning theories and modern pedagogical approaches that inform classroom learning strategies is enormous: most of our theories about how people learn are inspired by experimental paradigms and models that have only peripheral relevance to the kinds of learning that students and teachers actually seek[10,15]. To help bridge this gap, our study uses course materials from real online courses to inform, fit, and test models of real-world conceptual learning. We show that these models recover meaningful relationships between concepts presented during course lectures and tested by assessments, and that these relationships can be leveraged to predict students' success on individual quiz questions. We also provide a demonstration of how our models can be used to construct maps of what students know, and how their knowledge changes with training. In addition to helping to visually capture knowledge (and changes in knowledge), we hope that such maps might lead to real-world tools for improving how we educate. Taken together, our work shows that existing course materials and evaluative tools like short multiple-choice quizzes may be leveraged to gain highly detailed insights into what students know and how they learn.

## Results

At its core, our main modeling approach is based around a simple assumption that we sought to test empirically: all else being equal, knowledge about a given concept is predictive of knowledge about similar or related concepts. From a geometric perspective, this assumption implies that knowledge is fundamentally "smooth." In other words, as one moves through a space representing an individual's knowledge (where similar concepts occupy nearby coordinates), their level of knowledge should change relatively gradually. To begin to test this smoothness assumption, we sought to track participants' knowledge and how it changed over time in response to training. Two overarching goals guide our approach. First, we want to gain detailed insights into what learners know at different points in their training. For example, rather than simply reporting on the proportions of questions participants answer correctly (i.e., their overall performance), we seek estimates of their knowledge about a variety of specific concepts. Second, we want our approach to be potentially scalable to large numbers of diverse concepts, courses, and students. This requires that the conceptual content of interest be discovered automatically, rather than relying on manually produced ratings or labels.

We asked participants in our study to complete brief multiple-choice quizzes before, between, and after watching two lecture videos from the Khan Academy[33] platform (Fig. 1). The first lecture video, entitled *Four Fundamental Forces*, discussed the four fundamental forces in physics: gravity, strong and weak interactions, and electromagnetism. The second, entitled *Birth of Stars*, provided an overview of our current understanding of how stars form. We selected these particular lectures to satisfy three general criteria. First, we wanted both lectures to be accessible to a broad audience (i.e., with minimal prerequisite knowledge) so as to limit the impact of prior training on

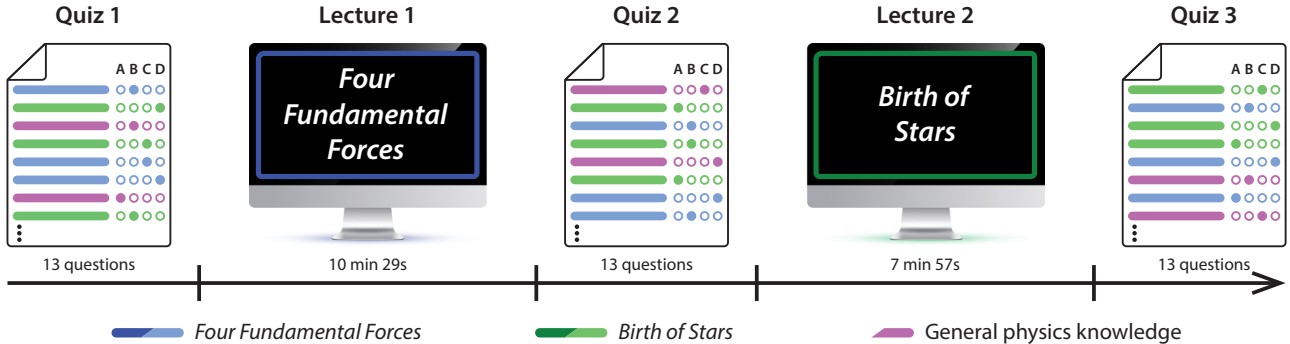

**Fig. 1 | Experimental paradigm.** Participants alternate between completing three 13-question multiple-choice quizzes and watching two Khan Academy lectures. Each quiz contains a mix of 5 questions about Lecture 1, 5 questions about Lecture 2, and 3 questions about general physics knowledge. The specific questions appearing on each quiz, and the orders of each quiz's questions, were randomized across participants.

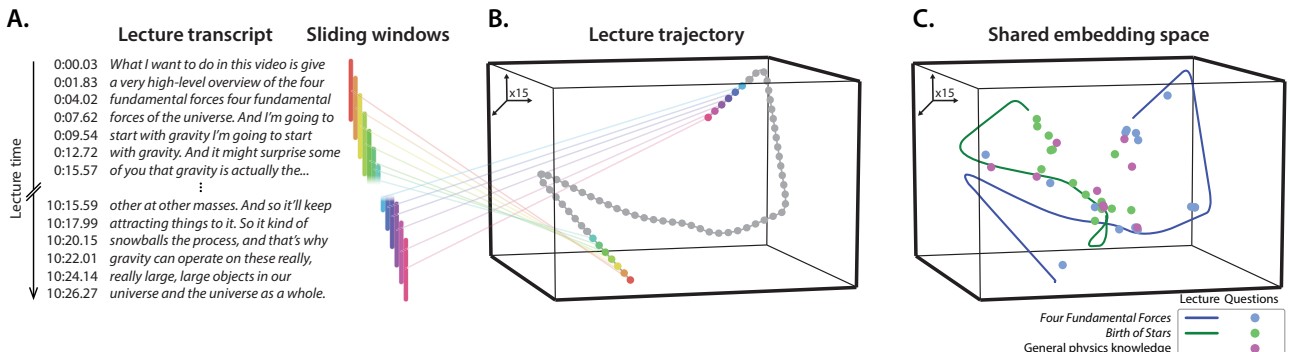

**Fig. 2 | Modeling course content. A Building a document pool from sliding windows of text.** We decompose each lecture's transcript into a series of overlapping sliding windows. The full set of transcript snippets (across all windows) may be treated as a set of documents for training a text embedding model.
**B Constructing lecture content trajectories.** After training the model on the sliding windows from both lectures, we transform each lecture into a trajectory through text embedding space by joining the embedding coordinates of successive sliding windows parsed from its transcript. **C** We apply the same model (trained on the two lectures' windows) to both lectures, along with the text of each question in our pool (Supplementary Table 1), to project them into a shared text embedding space. This results in one trajectory per lecture and one coordinate for each question. Here, we have projected the 15-dimensional embeddings onto their first 3 principal components for visualization.

participants' abilities to learn from the lectures. To this end, we selected two introductory videos that were intended to be viewed at the start of students' training in their respective content areas. Second, we wanted the two lectures to have some related content so that we could test our approach's ability to distinguish similar conceptual content. To this end, we chose two videos from the same Khan Academy course domain, "Cosmology and Astronomy." Third, we sought to minimize dependencies and specific overlap between the videos. For example, we did not want participants' abilities to understand one video to (directly) influence their abilities to understand the other. To satisfy this last criterion, we chose videos from two different lecture series (Lectures 1 and 2 were from the "Scale of the Universe" and "Stars, Black Holes, and Galaxies" series, respectively).

We also wrote a set of multiple-choice quiz questions that we hoped would enable us to evaluate participants' knowledge about each individual lecture, along with related knowledge about physics concepts not specifically presented in either video (see Supplementary Table 1 for the full list of questions in our stimulus pool). Participants answered questions randomly drawn from each content area (Lecture 1, Lecture 2, and general physics knowledge) on each of the three quizzes. Quiz 1 was intended to assess participants' baseline knowledge before training, Quiz 2 assessed knowledge after watching the *Four Fundamental Forces* video (i.e., Lecture 1), and Quiz 3 assessed knowledge after watching the *Birth of Stars* video (i.e., Lecture 2).

To study in detail how participants' conceptual knowledge changed over the course of the experiment, we first sought to model the conceptual content presented to them at each moment throughout each of the two lectures. We adapted an approach we developed in prior work[34] to identify the latent themes in the lectures using a topic model[18]. Briefly, topic models take as input a collection of text documents, and learn a set of "topics" (i.e., latent themes) from their contents. Once fit, a topic model can be used to transform arbitrary (potentially new) documents into sets of "topic proportions" describing the weighted blend of learned topics reflected in their texts. We parsed automatically generated transcripts of the two lectures into overlapping sliding windows, where each window contained the text of the lecture transcript from a particular time span. We treated the set of text snippets (across all of these windows) as documents to fit the model (Fig. 2A; see *Constructing text embeddings of multiple lectures and questions*). Transforming the text from every sliding window with the model yielded a number-of-windows by number-of-topics (15) topic-proportions matrix describing the unique mixture of broad themes from both lectures reflected in each window's text. Each window's "topic vector" (i.e., row of the topic-proportions matrix) is analogous to a coordinate in a 15-dimensional space whose axes are topics discovered by the model. Within this space, each lecture's sequence of topic vectors (i.e., corresponding to its transcript's overlapping text snippets across sliding windows) forms a trajectory that captures how its conceptual content unfolds over time (Fig. 2B). We resampled these

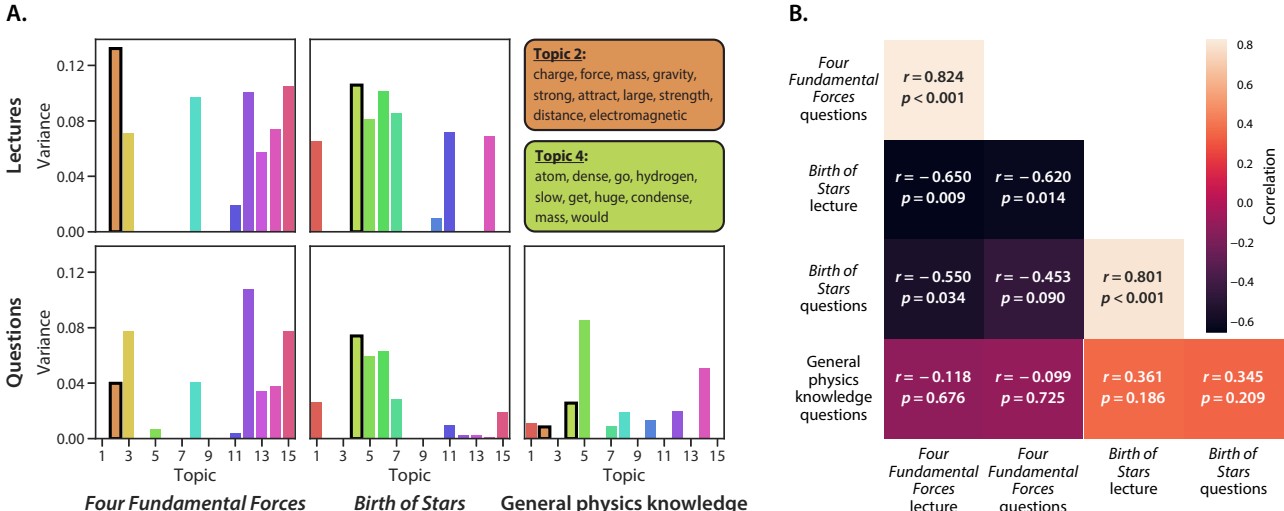

**Fig. 3 | Lecture and question topic overlap. A Topic weight variability.** The bar plots display the variance in each topic's weight across lecture timepoints (top row) and questions (bottom row); colors denote topics. The top-weighted words from the most expressive (i.e., variable across observations) topic from each lecture are displayed in the upper right (orange: topic 2; yellow-green: topic 4). The top-weighted words from the full set of topics may be found in Supplementary Table 2. **B Relationships between topic weight variability.** Pairwise correlations between the distributions of topic weight variance for each lecture and question set. Each row and column corresponds to a bar plot in (**A**).

trajectories to a resolution of one topic vector for each second of video (i.e., 1 Hz).

We hypothesized that a topic model trained on transcripts of the two lectures should also capture the conceptual knowledge probed by each quiz question. If indeed the topic model could capture information about the deeper conceptual content of the lectures (i.e., beyond surface-level details such as particular word choices), then we should be able to recover a correspondence between each lecture and questions about each lecture. Importantly, such a correspondence could not arise solely from superficial text matching between lecture transcripts and questions, since the lectures and questions often used different words (Supplementary Fig. 11) and phrasings. Simply comparing the average topic weights from each lecture and question set (averaging across time and questions, respectively) reveals a striking correspondence (Supplementary Fig. 2). Specifically, the average topic weights from Lecture 1 are strongly correlated with the average topic weights from questions about Lecture 1 ($r(13) = 0.809$, $p < 0.001$, 95% confidence interval (CI) = [0.633, 0.962]), and the average topic weights from Lecture 2 are strongly correlated with the average topic weights from questions about Lecture 2 ($r(13) = 0.728$, $p = 0.002$, 95% CI = [0.456, 0.920]). At the same time, the average topic weights from the two lectures are negatively correlated with the average topic weights from their non-matching question sets (Lecture 1 video vs. Lecture 2 questions: $r(13) = -0.547$, $p = 0.035$, 95% CI = [−0.812, −0.231]; Lecture 2 video vs. Lecture 1 questions: $r(13) = -0.612$, $p = 0.015$, 95% CI = [−0.874, −0.281]), indicating that the topic model also exhibits some degree of specificity. The full set of pairwise comparisons between average topic weights for the lectures and question sets is reported in Supplementary Fig. 2.

Another, more sensitive, way of summarizing the conceptual content of the lectures and questions is to look at variability in how topics are weighted over time and across different questions (Fig. 3). Intuitively, the variability in the expression of a given topic relates to how much "information"[35] the lecture (or question set) reflects about that topic. For example, suppose a given topic is weighted on heavily throughout a lecture. That topic might be characteristic of some aspect or property of the lecture overall (conceptual or otherwise), but unless the topic's weights change in meaningful ways over time, it

would be a poor indicator of any specific conceptual content in the lecture. We therefore also compared the variances in topic weights (over time and across questions) between the lectures and questions. The variability in topic expression was similar for the Lecture 1 video and questions ($r(13) = 0.824$, $p < 0.001$, 95% CI = [0.696, 0.973]), and for the Lecture 2 video and questions ($r(13) = 0.801$, $p < 0.001$, 95% CI = [0.539, 0.958]). Simultaneously, as reported in Fig. 3B, the variabilities in topic expression across different videos and lecture-specific questions (i.e., Lecture 1 video vs. Lecture 2 questions; Lecture 2 video vs. Lecture 1 questions) were negatively correlated, and neither video's topic variability was reliably correlated with the topic variability across general physics knowledge questions. Taken together, the analyses reported in Fig. 3 and Supplementary Fig. 2 indicate that a topic model fit to the videos' transcripts can also reveal correspondences (at a coarse scale) between the lectures and questions.

An individual lecture may be organized around a single broad theme at a coarse scale, but at a finer scale, each moment of a lecture typically covers a narrower range of content. Given the correspondence we found between the variabilities in topic expression across moments of each lecture and questions from its corresponding set (Fig. 3), we wondered whether the text embedding model might additionally capture these conceptual relationships at a finer scale. For example, if a particular question asks about the content from one small part of a lecture, we wondered whether the text embeddings could be used to automatically identify the matching moment(s) in the lecture. To explore this, we computed the correlation between each question's topic weights and the topic weights for each second of its corresponding lecture, and found that each question appeared to be temporally specific (Fig. 4). In particular, most questions' topic vectors were maximally correlated with a well-defined (and relatively narrow) range of timepoints from their corresponding lectures, outside of which the correlations fell off sharply (Supplementary Figs. 3, 4). We also qualitatively examined the best-matching intervals for each question by comparing the questions' text to the transcribed text from the most-correlated parts of the lectures (Supplementary Table 3). Despite that the questions were excluded from the text embedding model's training set, in general we found (through manual inspection) a close correspondence between the conceptual content that each question probed and the content covered by the best-matching

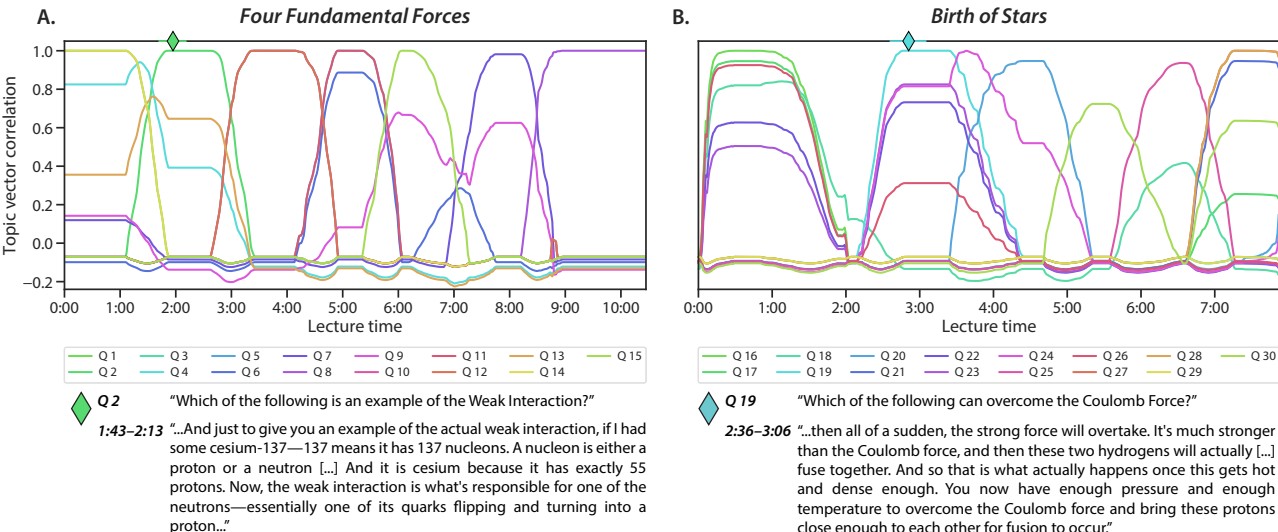

**Q 2**  "Which of the following is an example of the Weak Interaction?"

*1:43–2:13* "...And just to give you an example of the actual weak interaction, if I had some cesium-137—137 means it has 137 nucleons. A nucleon is either a proton or a neutron [...] And it is cesium because it has exactly 55 protons. Now, the weak interaction is what's responsible for one of the neutrons—essentially one of its quarks flipping and turning into a proton..."

**Q 19**  "Which of the following can overcome the Coulomb Force?"

*2:36–3:06* "...then all of a sudden, the strong force will overtake. It's much stronger than the Coulomb force, and then these two hydrogens will actually [...] fuse together. And so that is what actually happens once this gets hot and dense enough. You now have enough pressure and enough temperature to overcome the Coulomb force and bring these protons close enough to each other for fusion to occur."

**Fig. 4 | Which parts of each lecture are captured by each question?** Each panel displays time series plots showing how each question's topic vector correlates with each video timepoint's topic vector (**A**: correlations for the *Four Fundamental Forces* lecture and associated questions; **B**: correlations for the *Birth of Stars* lecture and associated questions). The colors denote question identities. The diamonds in each panel denote the moment of peak correlation between the indicated question and the lecture trajectory. The associated questions' text and snippets of the lectures' transcripts from the surrounding 30 seconds, are displayed at the bottom of the figure.

moments of the lectures. Two representative examples are shown at the bottom of Fig. 4.

The ability to quantify how much each question is asking about the content from each moment of the lectures could enable more detailed insights into participants' knowledge. Traditional approaches to estimating how much a student knows about the content of a given lecture entail administering some form of assessment (e.g., a quiz) and computing the proportion of questions the student answered correctly. But if two students receive identical scores on such an assessment, might our modeling framework help us to gain more nuanced insights into the specific content that each student has mastered (or failed to master)? For example, a student who misses three questions that were all about the same concept (e.g., concept *A*) will have gotten the same proportion of questions correct as another student who missed three questions about three different concepts (e.g., *A*, *B*, and *C*). But if we wanted to help these two students fill in the gaps in their understandings, we might do well to focus specifically on concept *A* for the first student, but to also add in materials pertaining to concepts *B* and *C* for the second student. In other words, raw proportion-correct measures may capture how much a student knows, but not what they know. We wondered whether our modeling framework might enable us to (formally and automatically) infer participants' knowledge at the scale of individual concepts (e.g., as captured by a single moment of a lecture).

We developed a simple formula (Eq. (1)) for using a participant's responses to a small set of multiple-choice questions to estimate how much that participant knows about the concept reflected by any arbitrary coordinate $x$ in text embedding space (e.g., the content reflected by any moment in a lecture they had watched; see *Estimating dynamic knowledge traces*). Essentially, the estimated knowledge at coordinate $x$ is given by the weighted proportion of quiz questions the participant answered correctly, where the weights reflect how much each question is "about" the content at $x$. When we apply this approach to estimate the participant's knowledge about the content presented in each moment of each lecture, we can obtain a detailed time course describing how much knowledge that participant has about the content presented at any part of the lecture. As shown in Fig. 5A, C, we can apply this approach separately for the questions from each quiz participants took throughout the experiment. From just a few questions

per quiz (see *Estimating dynamic knowledge traces*), we obtain a high-resolution snapshot (at the time each quiz was taken) of what participants knew about any moment's content, from either of the two lectures they watched (comprising a total of 1100 samples across the two lectures).

While the time courses in Fig. 5A, C provide detailed estimates about participants' knowlege, these estimates are of course only useful to the extent that they accurately reflect what participants actually know. As one sanity check, we anticipated that the knowledge estimates should reflect a content-specific boost in participants' knowledge after watching each lecture. In other words, if participants learn about each lecture's content upon watching it, the knowledge estimates should capture that. After watching the *Four Fundamental Forces* lecture, participants should exhibit more knowledge for the content of that lecture than they had before, and that knowledge should persist for the remainder of the experiment. Specifically, knowledge about that lecture's content should be relatively low when estimated using Quiz 1 responses, but should increase when estimated using Quiz 2 or 3 responses (Fig. 5B). Indeed, we found that participants' estimated knowledge about the content of *Four Fundamental Forces* was substantially higher on Quiz 2 versus Quiz 1 ($t(49) = 8.764$, $p < 0.001$) and on Quiz 3 versus Quiz 1 ($t(49) = 10.519$, $p < 0.001$). We found no reliable differences in estimated knowledge about that lecture's content on Quiz 2 versus 3 ($t(49) = 0.160$, $p = 0.874$). Similarly, we hypothesized (and subsequently confirmed) that participants should show greater estimated knowledge about the content of the *Birth of Stars* lecture after (versus before) watching it (Fig. 5D). Specifically, since participants watched that lecture after taking Quiz 2 (but before Quiz 3), we hypothesized that their knowledge estimates should be relatively low on Quizzes 1 and 2, but should show a boost on Quiz 3. Consistent with this prediction, we found no reliable differences in estimated knowledge about the *Birth of Stars* lecture content on Quiz 1 versus 2 ($t(49) = 1.013$, $p = 0.316$), but estimated knowledge was substantially higher on Quiz 3 versus 2 ($t(49) = 10.561$, $p < 0.001$) and Quiz 3 versus 1 ($t(49) = 8.969$, $p < 0.001$).

If we are able to accurately estimate a participant's knowledge about the content tested by a given question, our estimates of their knowledge should carry some predictive information about whether

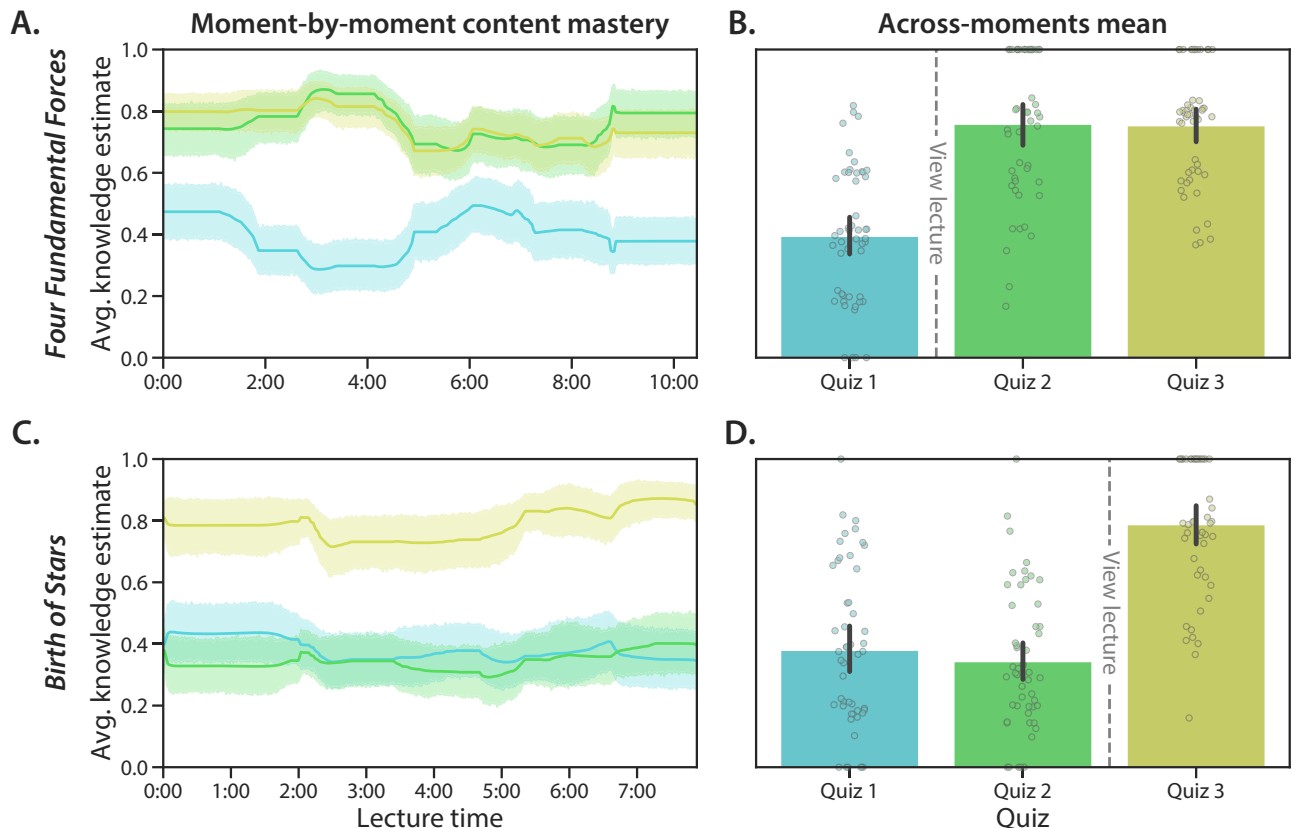

**Fig. 5 | Estimating knowledge about the content presented at each moment of each lecture. A** Knowledge about the time-varying content of *Four Fundamental Forces*. Each trace displays the weighted proportion of correctly answered questions about the content reflected in each moment of the lecture (see *Estimating dynamic knowledge traces*), using responses from a single quiz (color). The traces are averaged across participants. **B Average estimated knowledge about Four Fundamental Forces**. Each bar displays the across-timepoint average knowledge, estimated using the responses to one quiz's questions. **C Knowledge about the time-varying content of Birth of Stars**. The panel is in the same format as (**A**), but here the knowledge estimates are for the moment-by-moment content of the *Birth of Stars* lecture. **D Average estimated knowledge about Birth of Stars**. The panel is in the same format as (**B**), but here the knowledge estimates are for the content of the *Birth of Stars* lecture. **All panels.** Error ribbons and error bars denote 95% confidence intervals of the mean, estimated across participants (*n* = 50).

they are likely to answer that question correctly or incorrectly. We developed a statistical approach to test this claim. For each quiz question a participant answered, in turn, we used Eq. (1) to estimate their knowledge at that question's embedding-space coordinate based on other questions that participant answered on the same quiz. We repeated this for all participants, and for each of the three quizzes. Then, separately for each quiz, we fit a generalized linear mixed model (GLMM) with a logistic link function to explain the probability of correctly answering a question as a function of estimated knowledge at its embedding coordinate, while accounting for varied effects of individual participants and questions (see *Generalized linear mixed models*). To assess the predictive value of the knowledge estimates, we compared each GLMM to an analogous (i.e., nested) null model that assumed these estimates carried no predictive information using parametric bootstrap likelihood-ratio tests.

We carried out three different versions of the analyses described above, wherein we considered different sources of information in our estimates of participants' knowledge for each quiz question. First, we estimated knowledge at each held-out question's embedding coordinate using all other questions answered by the same participant on the same quiz ("All questions"; Fig. 6, top row). This test was intended to assess the overall predictive power of our approach. Second, we estimated knowledge for each question about a given lecture using only the other questions (from the same participant and quiz) about that same lecture ("Within-lecture"; Fig. 6, middle rows). This test was intended to assess the specificity of our approach by asking whether

our predictions could distinguish between questions about different content covered by the same lecture. Third, we estimated knowledge for each question about one lecture using only the questions (from the same participant and quiz) about the other lecture ("Across-lecture"; Fig. 6, bottom rows). This test was intended to assess the generalizability of our approach by asking whether our predictions could extend across the content areas of the two lectures. When estimating participants' knowledge, we used a rebalancing procedure to ensure that (for a given participant and quiz) their knowledge estimates for correctly and incorrectly answered questions were computed from the same underlying proportion of correctly answered questions (see *Generalized linear mixed models*).

When we fit a GLMM to estimates of participants' knowledge for each Quiz 1 question based on all other Quiz 1 questions, we found that higher estimated knowledge for a given question predicted a greater likelihood of answering it correctly (odds ratio (*OR*) = 8.126, 95% CI = [3.116, 20.123], likelihood-ratio test statistic ($\lambda_{LR}$) = 17.002, $p < 0.001$). This relationship held when we repeated this analysis for Quiz 2 (*OR* = 14.902, 95% CI = [4.976, 39.807], $\lambda_{LR}$ = 25.408, $p < 0.001$) and again for Quiz 3 (*OR* = 37.409, 95% CI = [10.425, 107.145], $\lambda_{LR}$ = 40.948, $p < 0.001$). Taken together, these results suggest that our knowledge estimates can reliably predict participants' performance on individual questions when they incorporate information from all (other) quiz content.

We observed a similar set of results when we restricted our estimates of participants' knowledge to consider only their performance

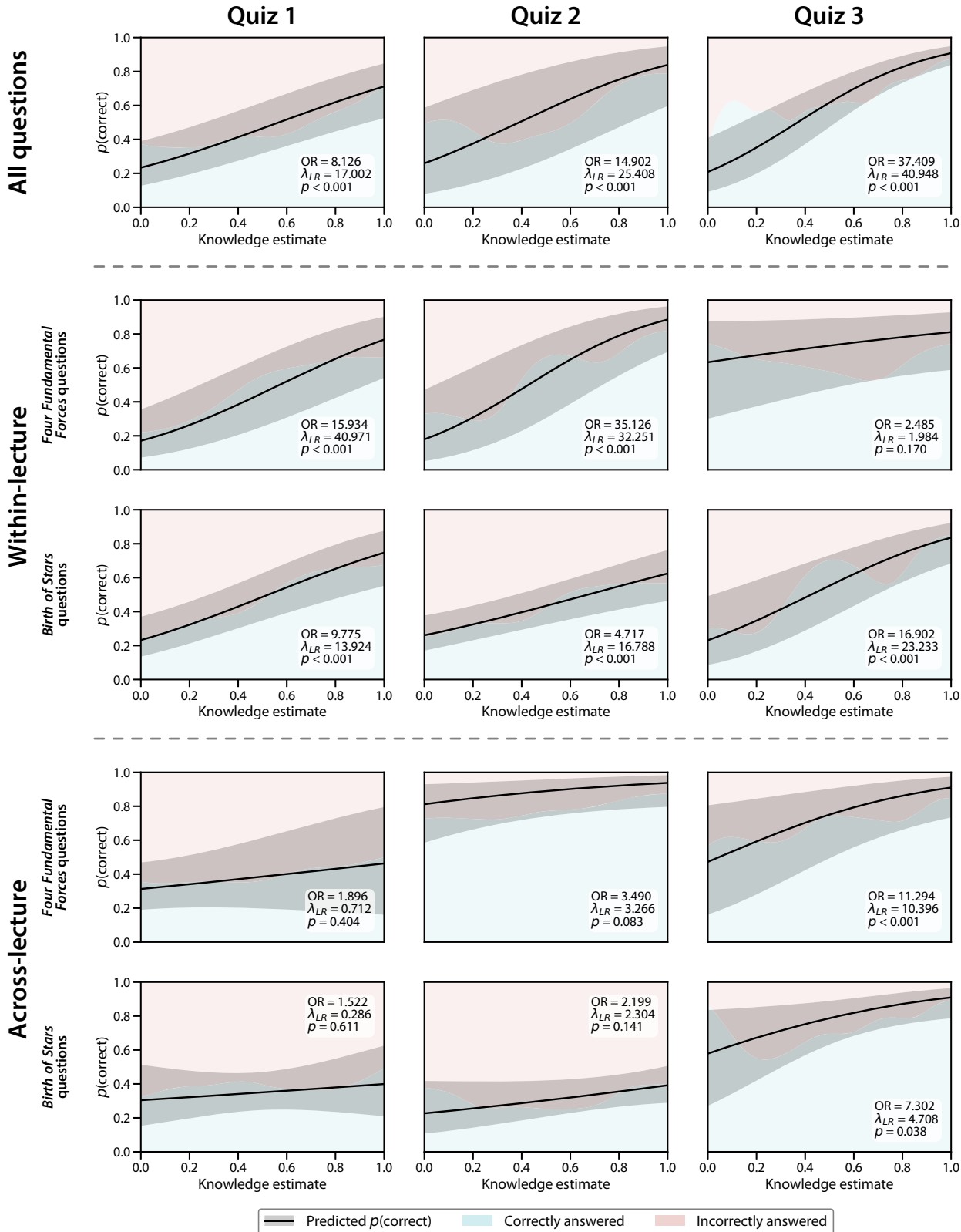

on other questions about the same lecture. Specifically, for Quiz 1, participants' knowledge of *Four Fundamental Forces*-related questions, estimated from their performance on other *Four Fundamental Forces*-related questions, was predictive of their ability to answer those questions correctly ($OR = 15.934$, 95% CI = [5.173, 38.005], $\lambda_{LR} = 40.971$, $p < 0.001$). The same was true of participants' estimated knowledge for *Birth of Stars*-related questions based on their performance on other

*Birth of Stars*-related questions ($OR = 9.775$, 95% CI = [2.930, 25.080], $\lambda_{LR} = 13.924$, $p < 0.001$). These results also held for participants' Quiz 2 responses (*Four Fundamental Forces*: $OR = 35.126$, 95% CI = [5.113, 123.868], $\lambda_{LR} = 32.251$, $p < 0.001$; *Birth of Stars*: $OR = 4.717$, 95% CI = [2.021, 9.844], $\lambda_{LR} = 16.788$, $p < 0.001$) and partially for their Quiz 3 responses (*Birth of Stars*: $OR = 16.902$, 95% CI = [3.353, 53.265], $\lambda_{LR} = 23.233$, $p < 0.001$; *Four Fundamental Forces*: $OR = 2.485$,

**Fig. 6 | Predicting success on held-out questions using estimated knowledge.**
We used generalized linear mixed models (GLMMs) to model the probability of
correctly answering a quiz question as a function of estimated knowledge for its
embedding coordinate (see *Generalized linear mixed models*). Separately for each
quiz (column), we examined this relationship based on three different sets of
knowledge estimates: knowledge for each question based on all other questions the
same participant answered on the same quiz ("All questions"; top row), knowledge
for each question about one lecture based on all other questions (from the same
participant and quiz) about the same lecture ("Within-lecture"; middle rows), and
knowledge for each question about one lecture based on all questions (from the

same participant and quiz) about the other lecture ("Across-lecture"; bottom rows).
The backgrounds in each panel display kernel density estimates of the relative
observed proportions of correctly (blue) versus incorrectly (red) answered ques-
tions, for each level of estimated knowledge along the *x*-axis. The black curves
display the (population-level) GLMM-predicted probabilities of correctly answering
a question as a function of estimated knowledge. Error ribbons denote 95% con-
fidence intervals of the predicted mean probabilities. *OR* denotes the model-
estimated odds ratio. $\lambda_{LR}$ denotes the result of a likelihood-ratio test for the effect of
estimated knowledge. *p*-values were estimated via parametric bootstrapping.

95% CI = [0.724, 8.366], $\lambda_{LR}$ = 1.984, *p* = 0.170). We note that the within-
lecture knowledge estimates are susceptible to ceiling effects in par-
ticipants' quiz performance. For example, on Quiz 3, after viewing both
lectures, no participant answered more than three *Four Fundamental
Forces*-related questions incorrectly, and all but five participants (out
of 50) answered two or fewer incorrectly. (This was the only subset of
questions about either lecture, across all three quizzes, for which this
was true.) Consequently, for 90% of participants, our within-lecture
estimates of their knowledge for *Four Fundamental Forces*-related
questions that they answered incorrectly leveraged information from
at most a single other question they were not able to correctly answer.
This hampered our ability to accurately characterize the specific (and
by the time they took Quiz 3, relatively few) aspects of the lecture
content these participants did not know about, and successfully dis-
tinguish them from the far more numerous aspects of the lecture
content they now did know about. Taken together, these within-
lecture results suggest that our knowledge estimates can reliably dis-
tinguish between questions about different content covered by a sin-
gle lecture, provided there is sufficient diversity in participants' quiz
responses to extract meaningful information about both what they
know and what they do not know.

Finally, we estimated participants' knowledge for each question
about each lecture using only their performance on questions (from
the same quiz) about the other lecture. This is an especially stringent
test of our approach. Our primary assumption in constructing our
knowledge estimates is that knowledge about a given concept is
similar to knowledge about other concepts that are nearby in the
embedding space. However, our analyses in Fig. 3 and Supplementary
Fig. 2 show that the embeddings of content from the two lectures (and
of their associated quiz questions) are largely distinct from each other.
Therefore, any predictive power of these across-lecture knowledge
estimates must overcome large distances in the embedding space. To
put this in concrete terms, this test requires predicting participants'
performance on individual, highly specific questions about the for-
mation of stars from their responses to just five multiple-choice
questions about the fundamental forces of the universe (and
vice versa).

We found that, before viewing either lecture (i.e., on Quiz 1),
participants' abilities to answer *Four Fundamental Forces*-related
questions could not be predicted from their responses to *Birth of Stars*-
related questions (*OR* = 1.896, 95% CI = [0.419, 9.088], $\lambda_{LR}$ = 0.712,
*p* = 0.404), nor could their abilities to answer *Birth of Stars*-related
questions be predicted from their responses to *Four Fundamental
Forces*-related questions (*OR* = 1.522, 95% CI = [0.332, 6.835],
$\lambda_{LR}$ = 0.286, *p* = 0.611). Similarly, we found that participants' perfor-
mance on questions about either lecture could not be predicted given
their responses to questions about the other lecture after viewing *Four
Fundamental Forces* but before viewing *Birth of Stars* (i.e., on Quiz 2;
*Four Fundamental Forces* questions given *Birth of Stars* questions:
*OR* = 3.49, 95% CI = [0.739, 12.849], $\lambda_{LR}$ = 3.266, *p* = 0.083; *Birth of Stars*
questions given *Four Fundamental Forces* questions: *OR* = 2.199,
95% CI = [0.711, 5.623], $\lambda_{LR}$ = 2.304, *p* = 0.141). Only after viewing both
lectures (i.e., on Quiz 3) did these across-lecture knowledge estimates
reliably predict participants' success on individual quiz questions (*Four*

*Fundamental Forces* questions given *Birth of Stars* questions:
*OR* = 11.294, 95% CI = [1.375, 47.744], $\lambda_{LR}$ = 10.396, *p* < 0.001; *Birth of
Stars* questions given *Four Fundamental Forces* questions: *OR* = 7.302,
95% CI = [1.077, 44.879], $\lambda_{LR}$ = 4.708, *p* = 0.038). Taken together, these
results indicate that our ability to form estimates solely across differ-
ent content areas is more limited than our ability to form estimates
that incorporate responses to questions from both content areas (as in
Fig. 6, "All questions") or within a single content area (as in Fig. 6,
"Within-lecture"). However, if participants have recently received some
training on both content areas, the knowledge estimates appear to be
informative even across content areas.

We speculate that these "Across-lecture" results might relate to
some of our earlier work on the nature of semantic representations[36].
In that work, we asked whether semantic similarities could be captured
through behavioral measures, even if participants' true internal
representations differed from the embeddings used to characterize
their behaviors. We found that mismatches between an individual's
internal representation of a set of concepts and the representation
used to characterize their behaviors can lead to underestimates of how
semantically driven those behaviors are. Along similar lines, we sus-
pect that in our current study, participants' conceptual representa-
tions may initially differ from the representations learned by our topic
model. (Although the topic model's representations are still related to
participants' initial internal representations; otherwise we would have
found that knowledge estimates derived from Quizzes 1 and 2 had no
predictive power in the other tests we conducted.) After watching both
lectures, however, participants' internal representations may become
more aligned with the embeddings used to estimate their knowledge
(since those embeddings were trained on the lectures' transcripts).
This could help explain why the knowledge estimates derived from
Quizzes 1 and 2 (before both lectures had been watched) do not reli-
ably predict performance across content areas, whereas estimates
derived from Quiz 3 do.

That the knowledge predictions derived from the text embed-
ding space reliably distinguish between correctly and incorrectly
answered held-out questions (Fig. 6) suggests that geometric rela-
tionships within this space can help explain what participants know.
But how far does this explanatory power extend? For example,
suppose we know that a participant correctly answered a question at
embedding coordinate *x*. As we move farther away from *x* in the
embedding space, how does the likelihood that the participant
knows about the content at a given location fall off with distance?
Conversely, suppose the participant instead answered that same
question incorrectly. Again, as we move farther away from *x* in the
embedding space, how does the likelihood that the participant does
not know about the content at a given coordinate change with dis-
tance? We reasoned that, assuming our embedding space is captur-
ing something about how individuals actually organize their
knowledge, a participant's ability to answer questions embedded
very close to *x* should tend to be similar to their ability to answer the
question embedded *at x*. But once we reach some sufficiently large
distance from *x*, our ability to infer whether or not a participant will
correctly answer a question based on their ability to answer the
question at *x* should be no better than guessing based on their

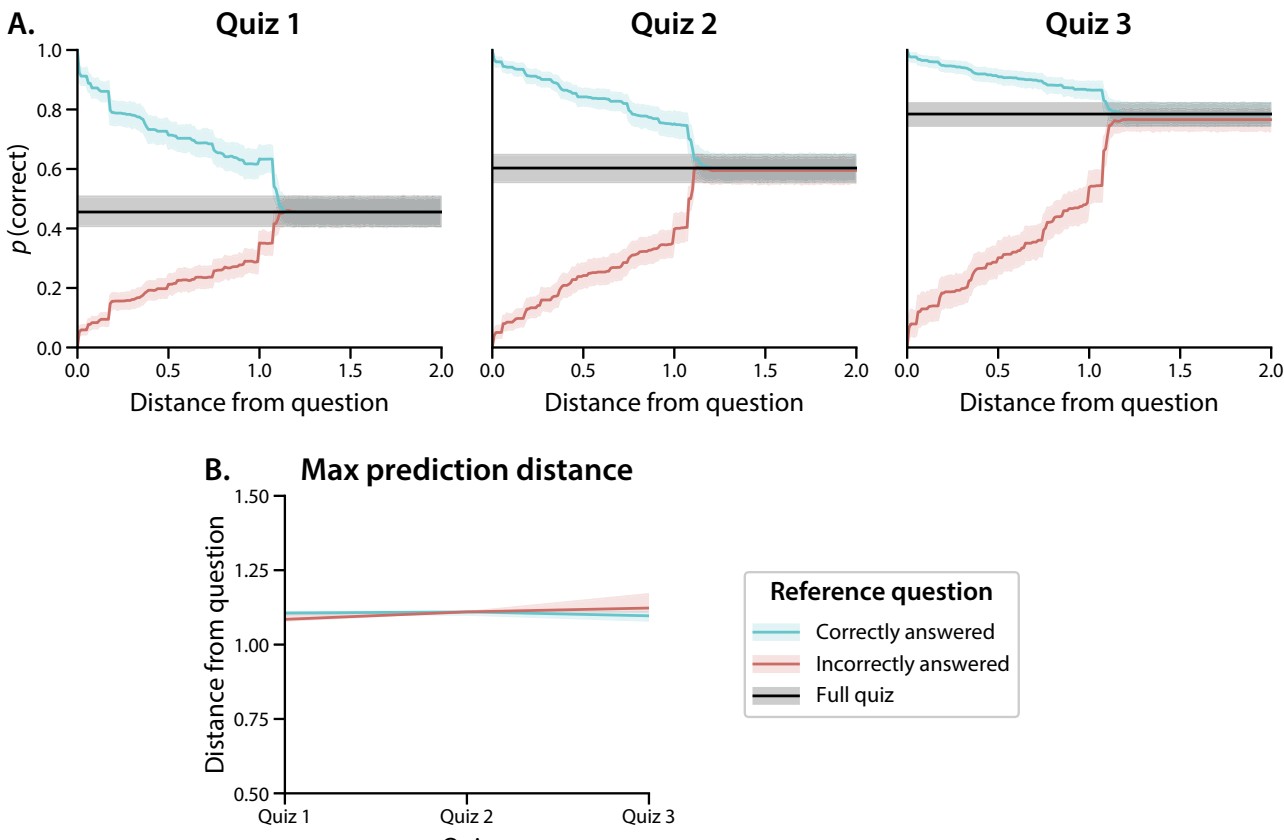

**Fig. 7 | Knowledge falls off gradually in text embedding space. A Performance versus distance.** For each participant, and for each correctly answered question (blue) or incorrectly answered question (red), we computed the proportion of correctly answered questions within a given distance of that question's embedding coordinate. We used these proportions as a proxy for participants' knowledge about the content within that region of the embedding space. We repeated this analysis for all questions and participants, and separately for each quiz (column). The black lines denote the average proportion correct including all questions a participant answered on the given quiz (i.e., not restricted by distance from the reference question). Error ribbons denote bootstrap-estimated 95% confidence intervals of the across-participants mean. **B Maximum distance for which performance is reliably different from the average.** We used a bootstrap procedure (see *Estimating the "smoothness" of knowledge*) to estimate the point at which the blue and red lines in (**A**) reliably diverged from the black line. We repeated this analysis separately for correctly and incorrectly answered questions from each quiz. Error ribbons denote 95% confidence intervals of the mean distance over 10,000 subsamples.

overall proportion of correctly answered questions. In other words, beyond the maximum distance at which a participant's ability to answer the question at $x$ is informative of their ability to answer a second question at location $y$, guessing the outcome at $y$ based on the outcome at $x$ should be no more successful than guessing based on a measure that does not consider embedding-space distance.

With these ideas in mind, we asked: conditioned on a participant's ability to answer a given question correctly, what proportion of all questions within some radius $r$ of its embedding coordinate were they able to answer correctly? We plotted this proportion as a function of $r$ for questions that participants answered correctly, and for questions they answered incorrectly. As shown in Fig. 7, we found that quiz performance falls off smoothly with distance, and the rate at which it falls off does not appear to differ across quizzes, as measured by the distance at which performance becomes statistically indistinguishable from a simple proportion-correct score (see *Estimating the "smoothness" of knowledge*). This suggests that, at least within the region of text embedding space spanned by the questions our study's participants answered (and as characterized using our topic model), the rate at which knowledge changes with distance is relatively constant, even as participants' overall level of knowledge varies across quizzes and regions of the embedding space.

Knowledge estimates need not be limited to the contents of these particular lectures and quizzes. As illustrated in Fig. 8, our general

approach to estimating knowledge from a small number of quiz questions may be extended to any content, given its text embedding coordinate. To visualize how knowledge "spreads" through text embedding space to content beyond the lectures participants watched and the questions they answered, we first fit a new topic model to the lectures' sliding windows with $k = 100$ topics. Conceptually, increasing the number of topics used by the model functions to increase the "resolution" of the embedding space, providing a greater ability to estimate knowledge for content that is highly similar to (but not precisely the same as) that contained in the two lectures used to train the model. Aside from increasing the number of topics from 15 to 100, all other procedures and model parameters were carried over from the preceding analyses. As in our other analyses, we resampled each lecture's topic trajectory to 1 Hz and projected each question into a shared text embedding space.

We projected the resulting 100-dimensional topic vectors (for each second of the lectures and each quiz question) onto a shared 2-dimensional plane (see *Creating knowledge and learning map visualizations*). Next, we sampled points from a 100 × 100 grid of coordinates that evenly tiled a rectangle enclosing the 2D projections of the lectures and questions. We then used Eq. (4) to estimate participants' knowledge at each of these 10,000 sampled locations, and averaged these estimates across participants to obtain an estimated average "knowledge map" (Fig. 8A). Intuitively, the knowledge map

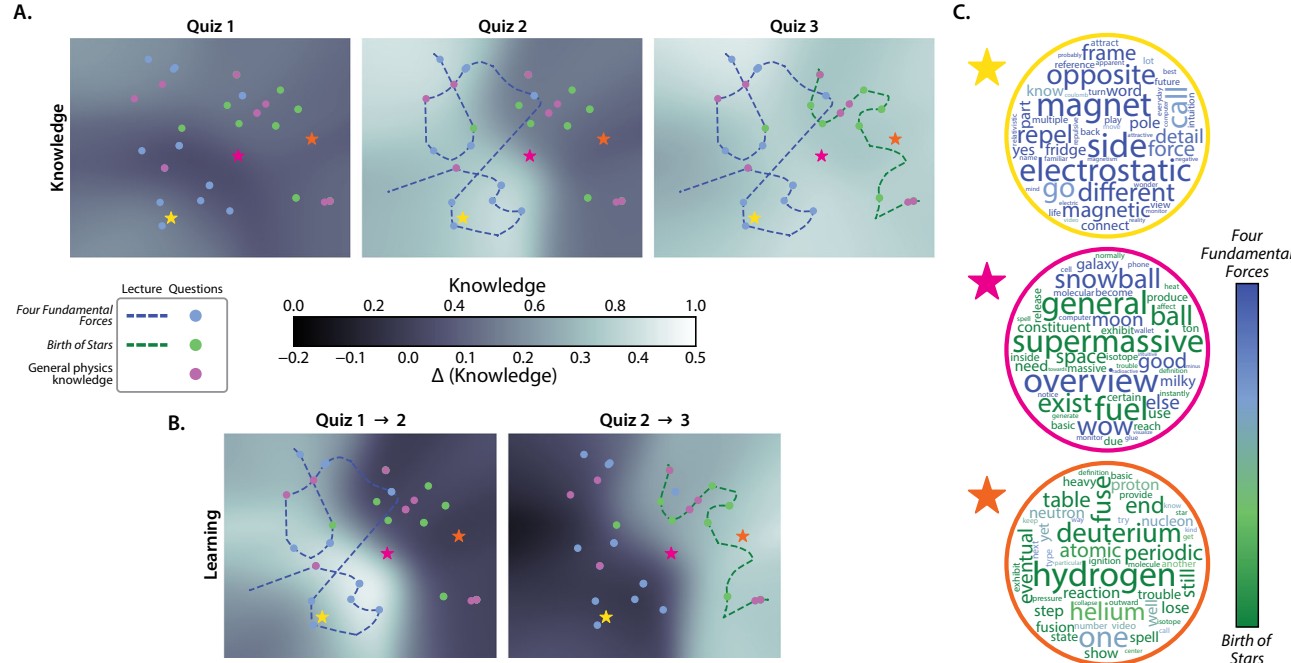

**Fig. 8 | Mapping out the geometry of knowledge and learning. A Average knowledge maps estimated using each quiz.** Each map displays a 2D projection of participants' estimated knowledge about the content reflected by all regions of topic space (see *Creating knowledge and learning map visualizations*). The topic trajectories of the two lectures are indicated by dotted lines (blue: Lecture 1; green: Lecture 2), and the coordinates of each question are indicated by dots (light blue: Lecture 1-related; light green: Lecture 2-related; purple: general physics knowledge). Each map reflects an average across all participants. For individual participants' maps, see Supplementary Figs. 5, 6, and 7. **B Average learning maps estimated between each successive pair of quizzes.** The learning maps follow the same general format as the knowledge maps in (**A**), but here the shading at each

coordinate indicates the difference between the corresponding coordinates in the indicated pair of knowledge maps—i.e., how much the estimated knowledge changed between the two quizzes. Each map reflects an average across all participants. For individual participants' maps, see Supplementary Figs. 8 and 9. **C Word clouds for sampled points in topic space.** Each word cloud displays the weighted blend of words underlying the topic proportions represented at the corresponding colored star's location on the maps. In each word cloud, the words' relative sizes correspond to their relative weights at the starred location, and their colors indicate their relative weights in the *Four Fundamental Forces* (blue) versus *Birth of Stars* (green) lectures, on average, across all timepoints' topic vectors.

constructed from a given quiz's responses provides a visualization of how much participants knew about any content expressible by the fitted text embedding model at the point in time when they completed that quiz. We note that we used these 2D maps solely for visualization; all relevant comparisons, distance computations, and statistical tests we report above were carried out in the original 15-dimensional space, using the 15-topic model.

Several features of the resulting knowledge maps are worth noting. The average knowledge map estimated from Quiz 1 responses (Fig. 8A, leftmost map) shows that participants tended to have relatively little knowledge about any parts of the text embedding space (i.e., the shading is relatively dark everywhere). The knowledge map estimated from Quiz 2 responses shows a marked increase in knowledge on the left side of the map (around roughly the same range of coordinates traversed by the *Four Fundamental Forces* lecture, indicated by the dotted blue line). In other words, participants' estimated increase in knowledge is localized to conceptual content that is nearby (i.e., related to) the content from the lecture they watched prior to taking Quiz 2. This localization is non-trivial: these knowledge estimates are informed only by the embedded coordinates of the quiz questions, not by the embeddings of either lecture (see Eq. (4)). Finally, the knowledge map estimated from Quiz 3 responses shows a second increase in knowledge, localized to the region surrounding the embedding of the *Birth of Stars* lecture participants watched immediately prior to taking Quiz 3.

Another way of visualizing these content-specific increases in knowledge after participants viewed each lecture is displayed in Fig. 8B. Taking the point-by-point difference between the knowledge maps estimated from responses to a successive pair of quizzes yields a

"learning map" that describes the change in estimated knowledge from one quiz to the next. These learning maps highlight that the estimated knowledge increases we observed across maps were specific to the regions around the embeddings of each lecture, in turn.

Because the 2D projection we used to construct the knowledge and learning maps is invertible, we may gain additional insights into these maps' meanings by reconstructing the original high-dimensional topic vector for any location on the map we are interested in. For example, this could serve as a useful tool for an instructor looking to better understand which content areas a student (or a group of students) knows well (or poorly). As a demonstration, we show the top-weighted words from the blends of topics reconstructed from three example locations on the maps (Fig. 8C): one point near the *Four Fundamental Forces* embedding (yellow), a second point near the *Birth of Stars* embedding (orange), and a third point between the two lectures' embeddings (pink). As shown in the word clouds in Panel C, the top-weighted words at the example coordinate near the *Four Fundamental Forces* embedding tended to be weighted more heavily by the topics expressed in that lecture. Similarly, the top-weighted words at the example coordinate near the *Birth of Stars* embedding tended to be weighted more heavily by the topics expressed in that lecture. The top-weighted words at the example coordinate between the two lectures' embeddings show a roughly even mix of words most strongly associated with each lecture.

## Discussion

We developed a computational framework that uses short multiple-choice quizzes to gain nuanced insights into what learners know and how their knowledge changes with training. First, we show that our

approach can automatically match the conceptual knowledge probed by individual quiz questions to the corresponding moments in lecture videos when those concepts were presented (Fig. 4). Next, we demonstrate how we can estimate moment-by-moment "knowledge traces" that reflect the degree of knowledge participants have about each lecture's time-varying content, and capture temporally specific increases in knowledge after viewing each lecture (Fig. 5). We then show that these knowledge estimates can generalize to held-out questions and predict participants' abilities to answer them correctly (Fig. 6). Finally, we use our framework to construct visual maps that provide snapshot estimates of how much participants know about any concept within the scope of our text embedding model, and how much their knowledge of those concepts changes with training (Fig. 8).

Our work makes several contributions to the study of how people acquire conceptual knowledge. First, from a methodological standpoint, our modeling framework provides a systematic means of mapping out and characterizing knowledge in maps that have infinite (arbitrarily many) numbers of coordinates, and of filling out those maps using relatively small numbers of multiple-choice quiz questions. Our experimental finding that we can use these maps to predict success on held-out questions has several psychological implications as well. For example, concepts that are assigned to nearby coordinates by the text embedding model also appear to be known to a similar extent (as reflected by participants' responses to held-out questions; Fig. 6). This suggests that participants also conceptualize similarly the content reflected by nearby embedding coordinates. How participants' knowledge falls off with spatial distance is captured by the knowledge maps we infer from their quiz responses (e.g., Figs. 7, 8). In other words, our study shows that knowledge about a given concept implies knowledge about related concepts, and how far this implication extends in text embedding space.

In our study, we characterize the coordinates of participants' knowledge using a relatively simple "bag-of-words" text embedding model (LDA[18]). More sophisticated text embedding models, such as transformer-based models[23,37–39], can leverage additional textual information such as complex grammatical and semantic relationships between words, higher-order syntactic structures, stylistic features, and more. We considered using transformer-based models in our study, but we found that the text embeddings derived from these models were surprisingly uninformative with respect to differentiating or otherwise characterizing the conceptual content of the lectures and questions we used (see Supplementary discussion). We suspect that this reflects a broader challenge in constructing models that are both high-resolution within a given domain (e.g., the domain of physics lectures and questions) and sufficiently broad as to enable them to cover a wide range of domains. Essentially, "larger" language models learn more complex features of language through training on enormous and diverse text corpora. But as a result, their embedding spaces also "span" an enormous and diverse range of conceptual content, sacrificing a degree of specificity in their capacities to distinguish subtle conceptual differences within a more narrow range of content. In comparing our LDA model (trained specifically on the lectures used in our study) to a larger transformer-based model (BERT), we found that our LDA model provides both coverage of the requisite material and specificity at the level of individual questions, while BERT essentially relegates the contents of both lectures and all quiz questions (which are all broadly about physics) to a tiny region of its embedding space, thereby blurring out meaningful distinctions between different specific concepts covered by the lectures and questions (Supplementary Fig. 10). We note that these are not criticisms of BERT, nor of other large language models trained on large and diverse corpora. Rather, our point is that simpler models trained on relatively small but specialized corpora can outperform much more complex models trained on much larger corpora when we are specifically interested in capturing subtle conceptual differences at the level of a single, narrowly

focused course lecture or quiz question. On the other hand, if our goal had been to choose a model that generalized to many different content areas simultaneously, we would expect our LDA model to perform comparatively poorly to BERT or other much larger general-purpose models. We suggest that bridging this tradeoff between achieving high resolution within a single content area and the ability to generalize to many diverse content areas will be an important challenge for future work.

At the opposite end of the spectrum from large language models, one could also imagine using an even simpler "model" than LDA that relates the contents of course lectures and quiz questions through explicit word-overlap metrics (rather than similarities in the latent topics they exhibit). In a supplementary analysis (Supplementary Fig. 11), we compared the LDA-based question-lecture matches shown in Fig. 4 with analogous matches based on the Jaccard similarity between each question's text and each sliding window from the corresponding lecture's transcript. As for the embeddings derived from BERT, we found that this word-matching approach also blurred meaningful distinctions between concepts presented in different parts of each lecture and tested by different quiz questions. But rather than characterizing their contents at too broad a semantic scale, the lack of specificity in this approach arises from considering too narrow a semantic scale: the sorts of concepts typically conveyed in course lectures and tested by quiz questions are not defined (and meaningful similarities and distinctions between them do not tend to emerge) at the level of individual words.

In other words, while the embedding spaces of more complex large language models afford low resolution at the scale of individual course lectures and questions because they "zoom out" too far, simpler word-matching measures afford low resolution because they "zoom in" too far. In this way, we view our approach as occupying a sort of sweet spot between simpler and more complex alternatives, in that it enables us to characterize the contents of course materials at the appropriate semantic scale where relevant concepts "come into focus." Our approach enables us to accurately and consistently identify each question's content in a way that matches it with specific content from the lectures and distinguishes it from other questions about similar content. In turn, this enables us to construct accurate predictions about participants' knowledge of the conceptual content tested by individual quiz questions (Fig. 6).

Another application for large language models that does not require explicitly modeling the content of individual lectures or questions is to leverage these models' abilities to generate text. For example, generative text models like ChatGPT[38] and LLaMa[39] are already being used to build a new generation of interactive tutoring systems (e.g.,[40]). Unlike the approach we have taken here, these generative text model-based systems do not explicitly model what learners know or how their knowledge changes over time with training. However, also unlike the approach we have taken here, these systems can respond to learners' needs with concise, natural language feedback in real time. One could imagine building a hybrid system that combines the best of both worlds: a smaller model that can infer what learners know and how their knowledge changes over time, and a larger generative model that can translate those insights into concrete, actionable feedback and interventions. Such a hybrid system could potentially be used to build the next generation of interactive tutoring systems that are able to adapt to learners' needs in real time, and provide more nuanced feedback about what learners know and what they do not know.

One limitation of our approach is that topic models contain no explicit internal representations of more complex aspects of knowledge, like knowledge graphs, dependencies or associations between concepts, causality, and so on. These representations might (in principle) be added as extensions to our approach to more accurately and precisely capture, characterize, and track learners' knowledge. However, modeling these aspects of knowledge will likely require substantial additional research effort.

Within the past several years, a global pandemic forced many educators to suddenly adapt to teaching remotely[41–44]. This change in world circumstances is happening alongside (and perhaps accelerating) geometric growth in the availability of high-quality online courses from platforms such as Khan Academy[33], Coursera[45], EdX[46], and others[47]. Continued expansion of the global internet backbone and improvements in computing hardware have also facilitated improvements in video streaming, enabling videos to be easily shared and viewed by increasingly large segments of the world's population. This exciting time for online course instruction provides an opportunity to re-evaluate how we, as a global community, educate ourselves and each other. For example, we can ask: what defines an effective course or training program? Which aspects of teaching might be optimized and/or augmented by automated tools? How and why do learning needs and goals vary across people? How might we lower barriers to receiving a high-quality education?

Alongside these questions, there is a growing desire to extend existing theories beyond the domain of lab testing rooms and into real classrooms[48]. In part, this has led to a recent resurgence of "naturalistic" or "observational" experimental paradigms that attempt to better reflect more ecologically valid phenomena that are more directly relevant to real-world situations and behaviors[49]. In turn, this has brought new challenges in data analysis and interpretation. A key step towards solving these challenges will be to build explicit models of real-world scenarios and how people behave in them (e.g., models of how people learn conceptual content from real-world courses, as in our current study). A second key step will be to understand which sorts of signals derived from behaviors and/or other measurements (e.g., neurophysiological data[50–54]) might help to inform these models. A third major step will be to develop and employ reliable ways of evaluating the complex models and data that are a hallmark of naturalistic paradigms.

Beyond specifically predicting what people know, the fundamental ideas we develop here also relate to the notion of "theory of mind" of other individuals[55–57]. Considering others' unique perspectives, prior experiences, knowledge, goals, etc., can help us to more effectively interact and communicate[58–60]. One could imagine future extensions of our work (e.g., analogous to the knowledge and learning maps shown in Fig. 8), that attempt to characterize how well-aligned different people's knowledge bases or backgrounds are. In turn, this might be used to model how knowledge (or other forms of communicable information) flows not just between teachers and students, but between friends having a conversation, individuals on a first date, participants at a business meeting, doctors and patients, experts and non-experts, political allies or adversaries, and more. For example, the extent to which two people's knowledge maps "match" or "align" in a given region of text embedding space might serve as a predictor of how effectively they will be able to communicate about the corresponding conceptual content.

Ultimately, our work suggests a rich new line of questions about the geometric form of knowledge, how knowledge changes over time, and how we might map out the full space of what an individual knows. Our finding that detailed estimates about knowledge may be obtained from short quizzes shows one way that traditional approaches to evaluation in education may be extended. We hope that these advances might help pave the way for new approaches to teaching or delivering educational content that are tailored to individual students' learning needs and goals.

## Methods
### Participants
We enrolled a total of 50 Dartmouth undergraduate students in our study. Participants received optional course credit for enrolling. No statistical method was used to predetermine sample size and no participants were excluded from the dataset. We asked each participant to complete a demographic survey that included questions about their age, gender, native spoken language, ethnicity, race, hearing, color vision, sleep, coffee consumption, level of alertness, and several aspects of their educational background and prior coursework.

Participants' ages ranged from 18 to 22 years (mean: 19.52 years; standard deviation: 1.09 years). A total of 15 participants reported their gender as male and 35 participants reported their gender as female. As we had no a priori hypotheses regarding sex or gender differences in this study, neither sex nor gender was considered in the study design and we did not perform any sex- or gender-based analyses. A total of 49 participants reported their native language as "English" and 1 reported having another native language. A total of 47 participants reported their ethnicity as "Not Hispanic or Latino" and three reported their ethnicity as "Hispanic or Latino." Participants reported their races as White (32 participants), Asian (14 participants), Black or African American (5 participants), American Indian or Alaska Native (1 participant), and Native Hawaiian or Other Pacific Islander (1 participant). (Note that some participants selected multiple racial categories.)

A total of 49 participants reporting having normal hearing and 1 participant reported having some hearing impairment. A total of 49 participants reported having normal color vision and 1 participant reported being color blind. Participants reported having had, on the night prior to testing, 2–4 h of sleep (1 participant), 4–6 h of sleep (9 participants), 6–8 h of sleep (35 participants), or 8+ h of sleep (5 participants). They reported having consumed, on the same day and leading up to their testing session, 0 cups of coffee (38 participants), 1 cup of coffee (10 participants), 3 cups of coffee (1 participant), or 4+ cups of coffee (1 participant).

No participants reported that their focus was currently impaired (e.g., by drugs or alcohol). Participants reported their current level of alertness, and we converted their responses to numerical scores as follows: "very sluggish" (−2), "a little sluggish" (−1), "neutral" (0), "fairly alert" (1), and "very alert" (2). Across all participants, a range of alertness levels were reported (range: −2–1; mean: −0.10; standard deviation: 0.84).

Participants reported their undergraduate major(s) as "social sciences" (28 participants), "natural sciences" (16 participants), "professional" (e.g., pre-med or pre-law; 8 participants), "mathematics and engineering" (7 participants), "humanities" (4 participants), or "undecided" (3 participants). Note that some participants selected multiple categories for their undergraduate major(s). We also asked participants about the courses they had taken. In total, 45 participants reported having taken at least one Khan Academy course in the past, and 5 reported not having taken any Khan Academy courses. Of those who reported having watched at least one Khan Academy course, 7 participants reported having watched 1–2 courses, 11 reported having watched 3–5 courses, 8 reported having watched 5–10 courses, and 19 reported having watched 10 or more courses. We also asked participants about the specific courses they had watched, categorized under different subject areas. In the "Mathematics" area, participants reported having watched videos on AP Calculus AB (21 participants), Precalculus (17 participants), Algebra 2 (14 participants), AP Calculus BC (12 participants), Trigonometry (11 participants), Algebra 1 (10 participants), Geometry (8 participants), Pre-algebra (7 participants), Multivariable Calculus (5 participants), Differential Equations (5 participants), Statistics and Probability (4 participants), AP Statistics (2 participants), Linear Algebra (2 participants), Early Math (1 participant), Arithmetic (1 participant), and other videos not listed in our survey (5 participants). In the "Science and engineering" area, participants reported having watched videos on Chemistry, AP Chemistry, or Organic Chemistry (21 participants); Physics, AP Physics I, or AP Physics II (18 participants); Biology, AP Biology, or High school Biology (15 participants); Health and Medicine (1 participant); and other videos not listed in our survey (5 participants). We also asked participants whether they had specifically seen the videos used in our experiment. Of

the 45 participants who reported having having taken at least one Khan Academy course in the past, 44 participants reported that they had not watched the *Four Fundamental Forces* video and 1 participant reported that they were not sure whether they had watched it. All participants reported that they had not watched the *Birth of Stars* video. When we asked participants about non-Khan Academy online courses, they reported having watched or taken courses on Mathematics (15 participants), Science and engineering (11 participants), Test preparation (9 participants), Economics and finance (3 participants), Arts and humanities (2 participants), Computing (2 participants), and other categories not listed in our survey (17 participants). Finally, we asked participants about in-person courses they had taken in different subject areas. They reported taking courses in Mathematics (38 participants), Science and engineering (37 participants), Arts and humanities (34 participants), Test preparation (27 participants), Economics and finance (26 participants), Computing (14 participants), College and careers (7 participants), and other courses not listed in our survey (6 participants).

### Experiment

We hand-selected two course videos from the Khan Academy platform: *Four Fundamental Forces* (an introduction to gravity, electromagnetism, the weak nuclear force, and the strong nuclear force; duration: 10 min and 29 s) and *Birth of Stars* (an introduction to how stars are formed; duration: 7 min and 57 s). All participants viewed the videos in the same order (i.e., *Four Fundamental Forces* followed by *Birth of Stars*).

We then hand-created 39 multiple-choice questions: 15 about the conceptual content of *Four Fundamental Forces* (i.e., Lecture 1), 15 about the conceptual content of *Birth of Stars* (i.e., Lecture 2), and 9 questions that tested for general conceptual knowledge about basic physics (covering material that was not presented in either video). To help broaden the set of lecture-specific questions, our team worked through each lecture in small segments to identify what each segment was about conceptually, and then write a question about that concept. The general physics questions were drawn from our team's prior coursework and areas of interest, along with internet searches and brainstorming with the project team and other members of J.R.M.'s lab. Although we attempted to design the questions to test "conceptual knowledge," we note that estimating the specific "amount" of conceptual understanding that each question requires to answer is somewhat subjective, and might even come down to the strategy a given participant used to answer the question at that particular moment. The full set of questions and answer choices may be found in Supplementary Table 1. The final set of questions (and response options) was reviewed and approved by J.R.M. before we collected or analyzed the text or experimental data.

Over the course of the experiment, participants completed three 13-question multiple-choice quizzes: the first before viewing Lecture 1, the second between Lectures 1 and 2, and the third after viewing Lecture 2 (see Fig. 1). The questions appearing on each quiz, for each participant, were randomly chosen from the full set of 39, with the constraints that (a) each quiz contained exactly 5 questions about Lecture 1, 5 questions about Lecture 2, and 3 questions about general physics knowledge, and (b) each question appeared exactly once for each participant. The orders of questions on each quiz, and the orders of answer options for each question, were also randomized. We obtained informed consent from all participants, and our experimental protocol was approved by the Committee for the Protection of Human Subjects at Dartmouth College. We used this experiment to develop and test our computational framework for estimating knowledge and learning.

### Analysis

**Statistics**. All of the statistical tests performed in our study were two-sided. The 95% confidence intervals we report for each correlation were estimated from bootstrap distributions of 10,000 correlation coefficients obtained by sampling (with replacement) from the observed data.

**Constructing text embeddings of multiple lectures and questions.** We adapted an approach we developed in prior work[34] to embed each moment of the two lectures and each question in our pool in a common representational space. Briefly, our approach uses a topic model (Latent Dirichlet Allocation[18]) trained on a set of documents to discover a set of $k$ "topics" or "themes." Formally, each topic is defined as a distribution of weights over words in the model's vocabulary (i.e., the union of all unique words across all documents, excluding stopwords). Conceptually, each topic is intended to give larger weights to words that are semantically related (as inferred from their tendency to co-occur in the same document). After fitting a topic model, each document in the training set, or any new document that contains at least some of the words in the model's vocabulary, may be represented as a $k$-dimensional vector describing how much that document (most probably) reflects each topic. To select an appropriate $k$ for our model, as a starting point, we identified the minimum number of topics that yielded at least one "unused" topic (i.e., in which all words in the vocabulary were assigned uniform weights) after training. This indicated that the number of topics was sufficient to capture the set of latent themes present in the two lectures (from which we constructed our document corpus, as described below). We found this value to be $k = 15$ topics. We found that with a limited number of additional adjustments following ref. 61, such as removing corpus-specific stopwords, the model yielded (subjectively) sensible and coherent topics. The distribution of weights over words in the vocabulary for each discovered topic is shown in Supplementary Fig. 1, and each topic's top-weighted words may be found in Supplementary Table 2.

As illustrated in Fig. 2A, we started by building up a corpus of documents using overlapping sliding windows that spanned each lecture's transcript. Khan Academy provides professionally created, manual transcriptions of all lecture videos for closed captioning. However, such transcripts would not be readily available in all contexts to which our framework could potentially be applied. Khan Academy videos are hosted on the YouTube platform, which additionally provides automated captions. We opted to use these automated transcripts (which, in prior work, we have found to be of sufficiently near-human quality to yield reliable data in behavioral studies[62]) when developing our framework in order to make it more directly extensible and adaptable by others in the future.

We fetched these automated transcripts using the `youtube-transcript-api`[63] Python package. Each transcript consisted of one timestamped line of text for every few seconds (mean: 2.34 s; standard deviation: 0.83 s) of spoken content in the lecture (i.e., corresponding to each individual caption that would appear on-screen if viewing the lecture via YouTube, and when those lines would appear). We defined a sliding window length of (up to) $w = 30$ transcript lines and assigned each window a timestamp corresponding to the midpoint between the timestamps for its first and last lines. This $w$ parameter was chosen to match the same number of words per sliding window (rounded to the nearest whole word, and before preprocessing) as the sliding windows we defined in our prior work (ref. 34; i.e., 185 words per sliding window).

These sliding windows ramped up and down in length at the beginning and end of each transcript, respectively. In other words, each transcript's first sliding window covered only its first line, the second sliding window covered the first two lines, and so on. This ensured that each line from the transcripts appeared in the same number ($w$) of sliding windows. We next performed a series of standard text preprocessing steps: normalizing case, lemmatizing, removing punctuation and removing stop-words. We constructed our corpus of stop-words by augmenting the Natural Language Toolkit

(NLTK[64]) English stop-word list with the following additional words, selected using one of the approaches suggested by ref. [61]: "actual," "actually," "also," "bit," "could," "e," "even," "first," "follow," "following," "four," "let," "like," "mc," "really," "saw," "see," "seen," "thing," and "two." This yielded sliding windows containing an average of 73.8 remaining words, and spanning an average of 62.22 seconds. We treated the text from each sliding window as a single document and combined these documents across the two lectures' windows to create a single training corpus for the topic model.

After fitting the topic model to the two lectures' transcripts, we could use the trained model to transform arbitrary (potentially new) documents into $k$-dimensional topic vectors. A convenient property of these topic vectors is that documents that reflect similar blends of topics (i.e., documents that reflect similar themes, according to the model) will yield similar coordinates (in terms of correlation, cosine similarity, Kullback-Leibler divergence, Euclidean distance, or other geometric measures). In general, the similarity between different documents' topic vectors may be used to characterize the similarity in conceptual content between the documents.

We transformed each sliding window's text into a topic vector, and then used linear interpolation (independently for each topic dimension) to resample the resulting time series to one vector per second. We also used the fitted model to obtain topic vectors for each quiz question in our pool (see Supplementary Table 1). Taken together, we obtained a trajectory for each lecture video describing its path through topic space, and a single coordinate for each question (Fig. 2C). Embedding both lectures and all of the questions using a common model enables us to compare the content from different moments of the lectures, compare the content across lectures, and estimate potential associations between specific questions and specific moments of lecture content.

**Estimating dynamic knowledge traces.** We used the following equation to estimate each participant's knowledge about timepoint $t$ of a given lecture, $\hat{k}(t)$:

$$\hat{k}(f(t,L)) = \frac{\sum_{i \in \text{correct}} \text{ncorr}(f(t,L), f(i,Q))}{\sum_{j=1}^{N} \text{ncorr}(f(t,L), f(j,Q))}, \quad (1)$$

where

$$\text{ncorr}(x,y) = \frac{\text{corr}(x,y) - \text{mincorr}}{\text{maxcorr} - \text{mincorr}}, \quad (2)$$

and where mincorr and maxcorr are the minimum and maximum correlations between the topic vectors for any lecture timepoint and quiz question, taken over all timepoints in the given lecture and all questions about that lecture appearing on the given quiz. We also define $f(s, \Omega)$ as the $s^{\text{th}}$ topic vector from the set of topic vectors $\Omega$. Here $t$ indexes the time series of lecture topic vectors $L$, and $i$ and $j$ index the topic vectors of questions $Q$ used to estimate the participant's knowledge. Note that "correct" denotes the set of indices of the questions the participant answered correctly on the given quiz.

Intuitively, ncorr($x$, $y$) is the correlation between two topic vectors (e.g., the topic vector $x$ for one timepoint in a lecture and the topic vector $y$ for one question on a quiz), normalized by the minimum and maximum correlations (across all timepoints $t$ and questions $j$) to range between 0 and 1, inclusive. Equation (1) then computes the weighted average proportion of correctly answered questions about the content presented at timepoint $t$, where the weights are given by the normalized correlations between timepoint $t$'s topic vector and the topic vectors for each question. The normalization step (i.e., using ncorr instead of the raw correlations) ensures that every question contributes some non-negative amount to the knowledge estimate.

**Generalized linear mixed models.** In the set of analyses reported in Fig. 6, we assessed whether estimates of participants' knowledge at the embedding coordinates of individual quiz questions could be used to reliably predict their abilities to correctly answer those questions. In essence, we treated each question a given participant answered on a given quiz as a lecture consisting of a single timepoint, and used Eq. (1) to estimate the participant's knowledge for its embedding coordinate based on their performance on all other questions they answered on that same quiz ("All questions"; Fig. 6, top row). Additionally, for each lecture-related question (i.e., excluding questions about general physics knowledge), we computed analogous knowledge estimates based on two different subsets of questions the participant answered on the same quiz: (1) all other questions about the same lecture as the target question ("Within-lecture"; Fig. 6, middle rows), and (2) all questions about the other of the two lectures ("Across-lecture"; Fig. 6, bottom rows).

In performing these analyses, our null hypothesis is that the knowledge estimates we compute based on the quiz questions' embedding coordinates do not provide useful information about participants' abilities to correctly answer those questions—in other words, that there is no meaningful difference (on average) between the knowledge estimates we compute for questions participants answered correctly versus incorrectly. Specifically, since we estimate knowledge for a given embedding coordinate as a weighted proportion-correct score (where each question's weight reflects its embedding-space distance from the target coordinate; see Eq. (1)), if these weights are uninformative (e.g., randomly distributed), then our estimates of participants' knowledge should be equivalent (on average) to the unweighted proportion of correctly answered questions used to compute them. In general, for a given participant and quiz, this expected null value (i.e., that participant's proportion-correct score on that quiz) is the same for any coordinate in the embedding space (e.g., any lecture timepoint, quiz question, etc.). However, in the "All questions" and "Within-lecture" versions of the analyses shown in Fig. 6, we estimate each participant's knowledge for each target question using all other questions (or all other questions about the same lecture) they answered on the same quiz. This introduces a systematic dependency between a participant's success on a target question and their proportion-correct score on the remaining questions available to estimate their knowledge for it. For example, suppose a participant correctly answered $n$ out of $q$ questions on a given quiz. If we hold out a single correctly answered question as the target, the proportion of remaining questions answered correctly would be $\frac{n-1}{q-1}$, whereas if we hold out a single incorrectly answered question, the proportion of remaining questions answered correctly would be $\frac{n}{q-1}$. Thus, the proportion of correctly answered remaining questions (and therefore the null-hypothesized value of a knowledge estimate computed from them) is always lower for target questions a participant answered correctly than for those they answered incorrectly.

To correct for this baseline difference under our null hypothesis, we used a rebalancing procedure that ensured our knowledge estimates for questions each participant answered correctly and incorrectly were computed from the same proportion of correctly answered questions. For each target question on a given participant's quiz, we first identified all remaining questions with the opposite "correctness" label (i.e., if the target question was answered correctly, we identified all remaining incorrectly answered questions, and vice versa). We then held out each of these opposite-label questions, in turn, along with the target question, and estimated the participant's knowledge for the target question using all other remaining questions. Since each of these subsets of remaining questions was constructed by holding out one correctly answered question and one incorrectly answered question from the participant's quiz responses, if the participant correctly answered $n$ out of $q$ questions total, then their proportion-correct score on each subset of questions used to estimate their knowledge

would be $\frac{n-1}{q-2}$, regardless of whether they answered the target question correctly or incorrectly. Finally, we averaged over these per-subset knowledge estimates to obtain a rebalanced estimate of the participant's knowledge for the target question that leveraged information from all remaining questions' embedding coordinates, but whose expected value under our null hypothesis was the same as that of each individual subset ($\frac{n-1}{q-2}$). By equalizing the null-hypothesized values of knowledge estimates for correctly and incorrectly answered questions, this procedure ensures that any meaningful relationships we observe between participants' estimated knowledge for individual quiz questions and their abilities to correctly answer them reflect the predictive power of the embedding-space distances we use to weight questions' contributions to the knowledge estimates, rather than an artifact of our testing procedure. Note that if a participant answered all or no questions on a given quiz correctly, their responses contained no opposite-label questions with which to perform this rebalancing, and we therefore excluded their data from our analyses for that quiz. We used this rebalancing procedure when constructing knowledge estimates for the "All questions" and "Within-lecture" versions of the analyses shown in Fig. 6, but not for the "Across-lecture" analyses as, in this case, the target questions and the questions used to estimate participants' knowledge for them were drawn from different subsets of quiz questions (those about one lecture, and those about the other), and were therefore independent.

In each version of this analysis (i.e., row in Fig. 6), and separately for each of the three quizzes (i.e., column in Fig. 6), we then fit a generalized linear mixed model (GLMM) with a logistic link function to the set of knowledge estimates for all questions (or all questions about a particular lecture) that participants answered on the given quiz. We implemented these models in R using the lme4 package[65] and fit them according to the guidance in refs. 66,67. Specifically, we initially fit each model with the maximal random effects structure afforded by our design, which we identified as:

$$accuracy \sim knowledge + (knowledge | participant) + (knowledge | question)$$

where accuracy is a binary value indicating whether each target question was answered correctly or incorrectly, knowledge is estimated knowledge at each target question's embedding coordinate, participant is a unique identifier assigned to each participant, and question is a unique identifier assigned to each quiz question. For models we fit using knowledge estimates for target questions about multiple content areas (i.e., in the "All questions" version of the analysis), we also included an additional random effect term, (knowledge | lecture), where lecture is a categorical value denoting whether the target question was about *Four Fundamental Forces*, *Birth of Stars*, or general physics knowledge. Note that with our coding scheme, identifiers for each question are implicitly nested within levels of lecture and so do not require explicit nesting in our model formula. We then iteratively removed random effects from the maximal model until it successfully converged with a full-rank random effects variance-covariance matrix. We obtained the odds ratios reported in Fig. 6 by exponentiating the estimated coefficient for knowledge from each fitted model. Conceptually, these odds ratios represent how many times greater the odds are that a given participant will answer a given question correctly if their estimated knowledge for its embedding coordinate is 1, compared to if it is 0. We estimated 95% confidence intervals for each odds ratio by generating 10,000 random subsamples (of full size, with replacement) from the data used to fit each model, and refitting the models to each subsample to obtain bootstrap distributions of 10,000 odds ratios.

To assess the predictive value of our knowledge estimates, we compared each GLMM's ability to explain participants' success on individual quiz questions to that of an analogous model which assumed (as we assume under our null hypothesis) that knowledge estimates for correctly and incorrectly answered questions did not systematically differ, on average. Specifically, we used the same sets of observations to which we fit each "full" model to fit a second "null" model with the same random effects structure, but with the coefficient for the fixed effect of knowledge constrained to zero (i.e., we removed this term from the null model). We then compared each full model to its reduced (null) equivalent using a likelihood-ratio test (LRT). Because the standard asymptotic $\chi_d^2$ approximation of the null distribution for the LRT statistic ($\lambda_{LR}$) can be anti-conservative for finite sample sizes[68–70], we computed *p*-values for these tests using a parametric bootstrap procedure[71,72]. For each of 10,000 bootstraps, we used the fitted null model to simulate a sample of observations of equal size to our original sample. We then re-fit both the null and full models to this simulated sample and compared them via an LRT. This yielded a distribution of $\lambda_{LR}$ statistics we may expect to observe given data that conforms to our null hypothesis. We computed a corrected *p*-value for our observed $\lambda_{LR}$ as $\frac{r+1}{n+1}$, where *r* is the number of simulated model comparisons that yielded a $\lambda_{LR}$ greater than our observed value and *n* is the number of simulations we ran (10,000).

**Estimating the "smoothness" of knowledge.** In the analysis reported in Fig. 7A, we show how participants' ability to correctly answer quiz questions changes as a function of distance from a given correctly or incorrectly answered reference question. We used a bootstrap-based approach to estimate the maximum distances over which these proportions of correctly answered questions could be reliably distinguished from participants' overall average proportion of correctly answered questions.

For each of 10,000 iterations, we drew a random subsample (with replacement) of 50 participants from our dataset. Within each iteration, we first computed the 95% confidence interval of the across-subsample-participants mean proportion correct on each of the three quizzes, separately. To compute this interval for each quiz, we repeatedly (1000 times) subsampled participants (with replacement, from the outer subsample for the current iteration) and computed the mean proportion correct of each of these inner subsamples. We then identified the 2.5th and 97.5th percentiles of the resulting distributions of 1,000 means. These three intervals (one for each quiz) served as our thresholds for confidence that the proportion correct within a given distance from a reference question was reliably different (at the $p < 0.05$ significance level) from the average proportion correct across all questions on the given quiz.

Next, for each participant in the current subsample, and for each of the three quizzes they completed (separately), we iteratively treated each of the 15 questions appearing on the given quiz as the reference question. We constructed a series of concentric 15-dimensional spheres centered on the reference question's embedding-space coordinate, where each successive sphere's radius increased by 0.01 (correlation distance) between 0 and 2, inclusive (i.e., tiling the range of possible correlation distances with 201 spheres in total). We then computed the proportion of questions enclosed within each sphere that the participant answered correctly, and averaged these per-radius proportion-correct scores across reference questions that were answered correctly, and those that were answered incorrectly. This resulted in two number-of-spheres sequences of proportion-correct scores for each subsample participant and quiz: one derived from correctly answered reference questions, and one derived from incorrectly answered reference questions.

We computed the across-subsample-participants mean proportion correct for each radius value (i.e., sphere) and "correctness" of reference question. This yielded two sequences of proportion-correct scores for each quiz, analogous to the blue and red lines displayed in

Fig. 7A, but for the present subsample. For each quiz, we then found the minimum distance from the reference question (i.e., sphere radius) at which each of these two sequences of per-radius proportion-correct scores intersected the 95% confidence interval for the overall proportion correct (i.e., analogous to the black error bands in Fig. 7A).

This resulted in two intersection distances for each quiz (for correctly answered and incorrectly answered reference questions). Repeating this full process for each of the 10,000 bootstrap iterations output two distributions of intersection distances for each of the three quizzes. The means and 95% confidence intervals for these distributions are plotted in Fig. 7B.

**Creating knowledge and learning map visualizations.** An important feature of our approach is that, given a trained text embedding model and participants' performance on each quiz question, we can estimate their knowledge about any content expressible by the embedding model—not solely the content explicitly probed by the quiz questions, or even appearing in the lectures. To visualize these estimates (Fig. 8, Supplementary Figs. 5, 6, 7, 8, and 9), we used Uniform Manifold Approximation and Projection (UMAP[73,74]) to construct a 2D projection of the text embedding space. Whereas our main analyses used a 15-topic embedding space, we used a 100-topic embedding space for these visualizations. This change in the number of topics overcame an undesirable behavior in the UMAP embedding procedure, whereby embedding coordinates for the 15-topic model tended to be grouped into separated clusters, rather than forming a smooth trajectory through the 2D space. When we increased the number of topics to 100, the embedding coordinates in the 2D space formed a smooth trajectory through the space, with substantially less aggressive grouping (Fig. 8). Creating a "map" by sampling this 100-dimensional space at high resolution to obtain an adequate set of topic vectors spanning the embedding space would be computationally intractable. However, sampling a 2D grid is trivial.

At a high level, the UMAP algorithm obtains low-dimensional embeddings by minimizing the cross-entropy between the pairwise (clustered) distances between the observations in their original (e.g., 100-dimensional) space and the pairwise (clustered) distances in the low-dimensional embedding space (in our approach, the embedding space is 2D). In our implementation, pairwise distances in the original high-dimensional space were defined as 1 minus the correlation between each pair of coordinates, and pairwise distances in the low-dimensional embedding space were defined as the Euclidean distance between each pair of coordinates.

In our application, all of the coordinates we embedded were topic vectors, whose elements are always non-negative and sum to one. Although UMAP is an invertible transformation at the embedding locations of the original data, other locations in the embedding space will not necessarily follow the same implicit rules as the original high-dimensional data. For example, inverting an arbitrary coordinate in the embedding space might result in negative-valued vectors, which are incompatible with the topic modeling framework. To protect against this issue, we log-transformed the topic vectors prior to embedding them in the 2D space. When we inverted the embedded vectors (e.g., to estimate topic vectors for word clouds, as in Fig. 8C), we passed the inverted (log-transformed) values through the exponential function to obtain a vector of non-negative values, and normalized them to sum to one.

After embedding both lectures' topic trajectories and the topic vectors of every question, we defined a rectangle enclosing the 2D projections of the lectures' and quizzes' embeddings. We then sampled points from a regular $100 \times 100$ grid of coordinates that evenly tiled this enclosing rectangle. We sought to estimate participants' knowledge (and learning; i.e., changes in knowledge) at each of the resulting 10,000 coordinates.

To generate our estimates, we placed a set of 39 radial basis functions (RBFs) throughout the embedding space, centered on the 2D projections for each question (i.e., we included one RBF for each question). At coordinate $x$, the value of an RBF centered on a question's coordinate $\mu$ is given by:

$$\mathrm{RBF}(x, \mu, \lambda) = \exp\left\{-\frac{||x - \mu||^2}{\lambda}\right\}. \tag{3}$$

The $\lambda$ term in the RBF equation controls the smoothness of the function, where larger values of $\lambda$ result in smoother maps. In our implementation, we used $\lambda = 50$. Next, we estimated the "knowledge" at each coordinate $x$, using:

$$\hat{k}(x) = \frac{\sum_{i \in \mathrm{correct}} \mathrm{RBF}(x, q_i, \lambda)}{\sum_{j=1}^{N} \mathrm{RBF}(x, q_j, \lambda)}. \tag{4}$$

Equation (4) computes the weighted proportion of correctly answered questions, where the weights are given by how nearby (in the 2D space) each question is to $x$. We also defined "learning maps" as the coordinate-by-coordinate differences between any pair of knowledge maps. Intuitively, learning maps reflect the change in knowledge between two maps.

### Reporting summary
Further information on research design is available in the Nature Portfolio Reporting Summary linked to this article.

## Data availability
All of the data analyzed in this manuscript may be found at https://github.com/ContextLab/efficient-learning-khan and is archived on Zenodo at https://doi.org/10.5281/zenodo.17783435[75]. All Khan Academy content is available for free at https://www.khanacademy.org.

## Code availability
All of the code for running our experiment and carrying out the analyses may be found at https://github.com/ContextLab/efficient-learning-khan and is archived on Zenodo at https://doi.org/10.5281/zenodo.17783435[75].

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

## Acknowledgements

We acknowledge useful discussions, assistance in setting up an earlier (unpublished) version of this study, and assistance with data collection efforts from Will Baxley, Max Bluestone, Daniel Carstensen, Kunal Jha, Caroline Lee, Lucy Owen, Xinming Xu, and Kirsten Ziman. Our work was supported in part by NSF CAREER Award Number 2145172 to J.R.M. The content is solely the responsibility of the authors and does not necessarily represent the official views of our supporting organizations. The funders had no role in study design, data collection and analysis, decision to publish, or preparation of the manuscript.

## Author contributions

Conceptualization: P.C.F., A.C.H., and J.R.M. Methodology: P.C.F., A.C.H., and J.R.M. Software: P.C.F. Validation: P.C.F. Formal analysis: P.C.F. Resources: P.C.F., A.C.H., and J.R.M. Data curation: P.C.F. Writing (original draft): J.R.M. Writing (review and editing): P.C.F., A.C.H., and J.R.M. Visualization: P.C.F. and J.R.M. Supervision: J.R.M. Project administration: P.C.F. Funding acquisition: J.R.M.

## Competing interests

The authors declare no competing interests.
