## [Transparent Peer Review file · Nature Communications]

Text embedding models yield detailed conceptual knowledge maps derived from short multiple-choice quizzes

Corresponding Author: Mr Paxton Fitzpatrick

Version 0:

Reviewer comments:

Reviewer #1

(Remarks to the Author)

Fitzpatrick and colleagues present a neat method for quantifying the content of real-world educational lecture videos and show that the resulting embedding space can be used to better understand and visualize quiz results assessing what knowledge students have learned. This manuscript is well-written and the figures are well-designed; I'm enthusiastic about the approach and the results are compelling. I have a couple clarifying questions about the methods, but my main concerns revolve around making sure the interpretations match the scope of the present data.

1. I have two related high-level comments, both of which I'm having hard time articulating. First, I think the authors should be careful about the interpretative language they use throughout. Sometimes the language feels a bit lofty or overreaching. Let me try to provide an example: I'm not sure I fully buy the rhetorical parallel between "memorization" and "understanding." I don't think anyone in educational contexts really thinks of rote memorization as memorizing character strings the way a language model might. On the other hand, I don't really see how the topic model used in the present study (or even much more sophisticated language models) are doing anything above and beyond memorizing statistical (i.e. co-occurrence) structure (albeit at multiple scales). Sure, the model is not a simple, discrete lookup table, but I'm not sure we want to make any strong claim that the model captures "understanding" of physics either. If I understand correctly, the main advance from "memorization" toward "understanding" is made by embedding words in a continuous vector space, granting Shepard-(1987) style(?) generalization (i.e. the authors commitment that knowledge is fundamentally "smooth"). But the analyses simply demonstrate that the embeddings for a test set of words (i.e. quiz questions) can be interpolated from the embedding space learned from a training set of words (i.e. the video transcripts). Is this kind of interpolation fundamentally different from "memorization"? (A more radical claim here might be that much of what is seemingly "understanding" can be reduced to this kind of interpolation—which I generally agree with, cf. our Hasson et al., 2020 paper—but I doubt the authors really want to make that claim either.)

2a. Second, I think the authors should be very clear about the limits/scope of their approach, particularly based on the dataset used herein. For instance, in the Introduction, you use aspirational language about "the sum total of a person's knowledge" and "the nearly infinite set of possible things a person could know"—but the actual data/model used here is relatively small-scale (and there's no explicit evidence for scalability). I can't imagine that the co-occurrence statistics across only <20 minutes of video lectures are particularly dense or rich. Just to confirm, the model is not pretrained in any way and does not incorporate any conceptual information from outside the lecture videos, right? For example, the authors say (line 492) "Conceptually, each topic is intended to give larger weights to words that are semantically related *or* tend to co-occur in the same documents" (my emphasis on "or"). Does the model know that words are semantically related above and beyond the co-occurrence statistics? My understanding was that "semantically related" is derived entirely from the co-occurrences statistics. In another example, you say "Knowledge estimates need not be limited to the content of the lectures." But the embedding space is strictly limited to the co-occurrence structure extracted from these particular lectures—right?

2b. Put another way, there's a tension in the writing between the potential generality of the method and the targeted nature of

the embedding space (i.e. tuned to the content of the two lectures) used herein. Here's a specific example: In my understanding, the negative correlations between mismatching lecture and video topic weights (line 176) are an indicator that the embedding space reflects an "either this or that" structure specific to the two lectures. I have the intuition that near-zero correlation here would provide stronger evidence for "specificity" (line 179) than anticorrelation (as we see with the general physics questions). I wonder if this trivializes the dimensionality reduction results in Figure 7—i.e. could it be any other way? (Maybe this is fine!)

2c. Another specific example: The authors want to make a strong claim that the model doesn't rely on surface-level similarity and that the "specific words" used in the quiz questions have "very little overlap" with the lectures? But there must be some overlap (i.e. co-occurrence), right? In my understanding, a completely disjoint set of words, none of which occurred in these video transcripts, would not map onto this embedding space in any meaningful way. To be clear, I don't think this is really a problem or a weakness of the method!—I'm just trying to articulate some discomfort when reading.

3. How exactly were the questions constructed for each lecture? For example, Figure 2 suggests that the questions follow the trajectories of the lectures, at least to some extent; obviously this is sensible and "by design." It's great that the authors provide the full set of questions/answers (Table S1)! However, I think it's important that the authors describe their thought process in designing the questions a bit more explicitly. For example, how exactly did they minimize "surface-level" similarity while ensuring enough conceptual overlap that the questions map onto the embedding space in a meaningful way?

4. The authors correlated the topic weights for each question with the topic weights at each time point in the lecture, and in Figure 4 show that "each question appeared to be temporally specific" (line 208). Intuitively, this seems reasonable... But couldn't this result also be interpreted as creating tension with the idea that the questions aren't simply keying into "surface-level" similarities? Put another way, if I thought this approach captured some kind of holistic "understanding", I might hypothesize that good questions would integrate multiple concepts introduced across different parts of the lecture. (I might be conflating the overall validity of the approach for capturing "understanding" with the quality of the questions here.)

5. The authors used windows comprising 30 lines(?) of the transcript to define documents for input to the topic model. This seems like an important parameter as it impacts the co-occurrence structure observed by the topic model, right? How was this number 30 chosen? I assume it represents some happy medium between providing a wide enough window to yield rich co-occurrence structure while retaining some temporal specificity. Transcript "lines" seems like a somewhat arbitrary unit; I assume it relates to how YouTube renders the transcript? How many words tend to occur in each of these windows? What's the average duration of these windows in seconds?

6. I don't fully understand the motivation for choosing 15 topics for most of the analyses, or why you increase to 100 topics for Figure 7. I don't think there's anything wrong with picking a somewhat arbitrary round number in these cases; just curious if there's any particular reasoning behind the present choices.

Minor comments:

Regarding my previous comments on scalability, do the authors think this kind of approach could be extended from topic models to the contextual embeddings learned by large language models? Could be worth mentioning if this is a valid future direction, given the current popularity of these models. My guess is that if we want a method that can capture "the sum total of a person's knowledge" and "the nearly infinite set of possible things a person could know," large language models are currently the best approach.

In Figures 2 and 7, you use PCA to visualize two- and three-dimensional projections of the embedding space. What proportion of cumulative variance do these first two and three PCs account for?

All participants saw the Four Fundamental Forces video first, followed by the Birth of a Star video, right? Should we be worried about some order effect?

Line 283: These t-values have 600+ degrees of freedom; I assume this is determined by the total number of questions across all participants. Should we be worried that this kind of statistical test doesn't take into account subject-level variability?

Line 492: Do you use a specific set of stop words? (e.g. the NLTK package has a standard list of stop words)

Line 557: "insure" > "ensure"

References:

Hasson, U., Nastase, S. A., & Goldstein, A. (2020). Direct fit to nature: an evolutionary perspective on biological and artificial neural networks. *Neuron*, 105(3), 416-434. <https://doi.org/10.1016/j.neuron.2019.12.002>

Shepard, R. N. (1987). Toward a universal law of generalization for psychological science. *Science*, 237(4820), 1317-1323. <https://doi.org/10.1126/science.3629243>

Reviewer #2

(Remarks to the Author)

The authors develop a framework for characterizing the acquisition of knowledge across time, specifically in the context of educational lectures. Participants watched two physics-related lectures on the Khan Academy platform, and completed quizzes before the first lecture, in between lectures, and after the second lecture. Questions on all quizzes tapped into knowledge from the two distinct lectures (as well as general physics knowledge). The information contained in the lectures and associated quiz questions was embedded in a topic space derived from word co-occurrences contained therein. This approach enabled the authors to (1) define each of the two lectures as a trajectory through this embedding space, (2) quantitatively determine the differences in conceptual content between the two lectures (by assessing topic weight and variability), (3) match quiz questions to specific moments in the lecture with relatively high resolution, (4) determine how much knowledge participants had about different points in the lecture, (5) predict whether a participant got a question correct or incorrect based on their estimated knowledge at that relevant coordinate in the embedding space, and (5) visualize how knowledge spreads through the embedding space as a result of watching the lectures. I am convinced that this framework yields more informative feedback on students' quiz results, relative to calculating percent correct. I find the authors' approach to be an exciting one, and that it provides methods others can use to quantify and visualize the specific content of one's knowledge and how it might change over time. The methodology is sound, and the introduction and discussion are written well. This work primarily proposes a methodological framework rather than presenting new empirical results. Despite my overall enthusiasm, some other comments/suggestions are summarized below.

1. It is mentioned multiple times that correspondences between quiz questions and lectures (e.g., localizing each quiz question to a specific point along a lecture's trajectory) are not superficial, because "the lectures and questions used different words." As far as I can see, there is no quantification of word overlap between lectures and quizzes, and we are supposed to take the authors' word on this. I find it difficult to believe that there is no word overlap between lectures and quiz questions, and would like to see some quantified measure of word overlap. The degree to which correspondences in the embedding space are "superficial" or "deep" is unclear and as it stands, I am not fully convinced that the current analyses tap into deep conceptual knowledge.
2. I would like to see the "Estimating Knowledge" analyses to go a step farther. The authors estimate participants' knowledge at the position of each quiz question's coordinate in a leave-one-out fashion, and show that these estimates reliably track whether the question was answered correctly or incorrectly. However, this analysis currently includes within-lecture prediction e.g., Lecture 1 quiz question performance to estimate a left-out Lecture 1 question. This analysis would be much more convincing and empirically interesting if a cross-lecture approach was used. That is, if for Quiz 2, performance on Lecture 1 questions could be used to predict accuracy on (unwatched) Lecture 2 quiz questions. This would be more convincing evidence that the embedding space is capturing deep conceptual meaning.
3. In the current "Estimating Knowledge" analyses, I'm having a hard time understanding what is happening in Quiz 3. It makes sense that there's no difference in the correct/incorrect distributions in Quiz 1, and it's great that it emerges in Quiz 2, but I would have also expected to see the effect for Quiz 3 (and indeed a weak effect was found). The authors explain this as a "ceiling effect", but it doesn't look like the knowledge estimates are at ceiling in Fig6-Quiz3. Further, if the knowledge estimates based on the embedding space are not sufficiently predicting quiz question accuracy, then what does? In other words, if the knowledge estimates for all regions of the space were high, then why are the participants' responses wrong? Is the model not capturing other pertinent knowledge, or do the authors think participants were choosing answers randomly?
4. I don't find the "Moment-by-moment" plots in Figure 5 to be useful, and in a sense they are misleading. With time on the x-axis, the implication is that the plot will be showing learning over time, as the lecture unfolds. This leads to the initial question of why the plots don't show an increase over time. However, this analysis does not reveal increasing knowledge over time, but rather the degree of knowledge relating to each moment in the lecture, assessed after the lecture was completed. What extra information are we getting from these moment-to-moment plots, that we're not getting from the averages presented in Fig 5B and 5D? Especially since there's no indication of how topics are changing from moment to moment, these plots seem superfluous. An alternate possibility is to show what topics participants know in each of the quizzes.
5. The Topic Weight Variability analysis (and corresponding Fig 3A) feels a bit lackluster. The authors have set up this rich, high-dimensional space in which to perform their other analyses, so it would be more satisfying if this analysis was also done using a geometric approach. For example, could variance be captured by the distance spanned by a lecture's trajectory in each of the 15 dimensions? Could variability be visualized in 3D space? I also think the decision to visualize the topic variance, rather than topic weights, could be better justified—it's not clear in the manuscript's current form why we should care more about variance than weights, which is more intuitively relevant.
6. This a relatively minor qualm, but referring to a coordinate in the embedding space as "a single concept" or "individual concept" doesn't seem quite right (e.g., "We wondered whether our modeling framework might enable us to...infer participants' knowledge at the scale of individual concepts (e.g., as captured by a single moment of a lecture). The notion of a single concept implies that concepts are units of some kind, and capturing conceptual knowledge in a high-dimensional space runs counter to that view. Further, what does a "single concept" mean if the authors count "fish", "water" and "fish swim in water" to each be a concept (p. 3)? In this case, a full sentence would count as a concept, or even the information contained in a whole document. Maybe the authors take this view, but it feels particularly odd given the initial use of "fish" and "water" as initial examples.

Reviewer #3

(Remarks to the Author)
See attached file.

Version 1:

Reviewer comments:

Reviewer #1

(Remarks to the Author)
The authors have adequately addressed my comments. I appreciate the very thorough response and I endorse the manuscript for acceptance. The comparisons to BERT and word-level Jaccard similarity, and the accompanying explanations, were also very helpful for understanding what insights their approach provides.

Reviewer #2

(Remarks to the Author)
I am happy with the responses and edits from the authors in the updated version of the manuscript. In particular, I find the across-lecture knowledge prediction and the "smoothness of knowledge" analyses to be important and interesting, and make a satisfying contribution to the full set of empirical results.

There are a few new sentences that are hard to parse, along with some typos, so the authors should take care to make those fixes. I don't need to see an updated version, but if the authors find it useful I can give a few examples:

p. 14 - line 339 - "and and"

p. 23 - line 470 - "The 'spatial smoothness'... is being captured by the knowledge maps we are inferring from their quiz questions."

p. 36 - line 803 - "...from our dataset full dataset."

I should also note that new edits on lines 72 and 738 are cut off on the provided pdf and so I cannot speak to their content.

Reviewer #3

(Remarks to the Author)
See attached PDF.

Version 2:

Reviewer comments:

Reviewer #2

(Remarks to the Author)
The authors clearly put in a lot of work to address Reviewer 3's concerns. My judgment of the manuscript remains similar to what it was during the last submission, though I do have some comments relating to the new content/analyses and how it is described and interpreted.

1. The "predicting success on held-out questions using estimated knowledge" section is the one that went through the largest overhaul. I still find the within-lecture and across-lecture analyses to be important, though it is unfortunate that with the new methods the results are harder to interpret. The authors do a good job of trying to explain the pattern of results (sometimes knowledge estimates do predict quiz question accuracy, and other times they do not), though it is not a strong set of results with clear empirical insights. However, since in my view the value of this work is in its methodological rather than empirical contribution, I do not see this as a reason to not approve the manuscript. That being said, I do have a (a) point of confusion, and (b) question/suggestion regarding the analysis.

a. In the rebuttal document, the authors explain an issue regarding the models incorporating the percent-correct measure, in which using percent-correct alone results in a negative relationship between estimated knowledge and probability of answering a question correctly (rebuttal starting at page #3). Given this strange relationship, the authors say that they have decided not to include these results in their paper. However, in the paper, the authors do include analyses that relate to this negative null result (starting manuscript page #16). I assume that the authors are referring to slightly different analyses in

their manuscript vs. rebuttal, but if they decided to include analyses with this negative relationship in the manuscript, why did they leave out other ones? The fact that I am not clear why certain analyses were included or excluded makes me feel like I am missing an important aspect of these analyses, but perhaps this is a fault of my own.

b. In any event, the situation in which the null result is a negative relationship causes confusions and complications (regarding the results displayed in Fig. 6). If the null result is negative (and not zero), then shouldn't the significance of the results be assessed relative to that negative value (rather than zero)? For example, might there be some observed relationship that may not itself be significantly positive, but still different from the negative null? If so, and if I understand the author's analysis correctly, then might they not be using a significance test that is actually too conservative? This might be a situation in which generating a null distribution through permutation testing might be the best solution; then the significance of the observed relationship can be compared to the null distribution to determine significance.

2. The authors discuss their LDA embedding space in relation to more complex (BERT) and simpler (word-matching) models. I have (a) a comment on manuscript organization, and (b) a question about what this means in practice.

a. It seems strange to take up so much space with the comparisons between LDA and BERT while the BERT results are relegated to the supplementary materials. As a style suggestion, it would seem cleaner/more efficient to either move some of these discussions to the supplementary material alongside those results, or to condense the current material and keep in the discussion.

b. The authors claim that their LDA embedding space is a "sweet spot" in between LLMs like BERT and simpler word-matching models. How specific is this to the particular lectures and quizzes used in the current study? Might e.g. LLMs work better than LDA for other researchers wanting to use these methods? If this is the case, then how should other researchers using their model determine which kind of embedding space to use? What kind of test can be run to determine the correct level of granularity in the embedding space? The proposed model will not be that useful of a tool if it takes thorough testing to determine the appropriate kind of embedding space before any actual knowledge analysis is done.

3. In their explanation of their Fig. 6 results, the authors suggest that one possibility is that participants forgot some material: "If this forgetting happens in a relatively "random" way (with respect to spatial distance within the embedding space), then it could explain why some held-out questions about Four Fundamental Forces were answered incorrectly, even if questions at nearby coordinates (i.e., about similar content) were answered correctly." (p. 18).

a. This seems to contradict their theory/finding regarding the "smoothness" of knowledge space. If forgetting happens randomly relative to the spatial location of knowledge, this evokes a kind of "Swiss cheese" knowledge space with holes randomly scattered about—in this case, knowledge would NOT be smooth, as the authors seem to claim. I can't reconcile these two claims. Perhaps the smoothness visualized in Fig. 7 is a function of averaging/combining across participants, but then making the general smoothness claim would not be so warranted.

(Remarks on code availability)

Version 3:

Reviewer comments:

Reviewer #4

(Remarks to the Author)

The manuscript by Fitzpatrick and colleagues describes an application of text embedding models to measure and "map" conceptual knowledge during and after learning about some topics in physics (watching videos and then taking quizzes). I did not review prior versions of this manuscript but have been asked to assess the responses of the authors to prior rounds of feedback. Given this context, below I provide my own evaluation of the manuscript (mostly focused on the responses to Reviewer 3's concerns), but I do not specifically suggest or request any further edits to the manuscript.

First, it is clear that the authors have put a tremendous amount of work into responding to prior comments from reviewers. I think they have been about as responsive as possible and the paper is very well written. In many respects it is a beautiful paper. Reviewers 1 and 2 seem generally satisfied with the revisions. Reviewer 3, however, raised a number of concerns related to details of the statistical analyses, the specific models used, and the overall significance of the findings. My impression is that the authors have satisfactorily addressed Reviewer 3's comments related to the statistical analyses and have appropriately contextualized the choice of model (with helpful comments about other models). However, I do share some of Reviewer 3's concerns about the significance of the work—and these concerns are, I think, less addressable unless the scope of the paper was substantially expanded (which I think is infeasible). Thus, I have a mixed impression: the authors have done some very commendable work on this paper and there is certainly a wow factor to the methods and visualizations (which may be enough for the paper to have impact), but I do have concerns about the actual significance of the work.

Comments

1. My overarching reaction to the manuscript is that it is more impressive than it is significant. Reviewer 3 shared a similar reaction: "the authors' algorithm is cool, but does it sufficiently advance us beyond the capabilities that already exist in large

language models, which are available to students and instructors.” Reviewer 3 was perhaps more narrowly focused on the advance of the current methods relative to other large language models or to ChatGPT, but I think the point applies more generally. I might differ from Reviewers 1 and 2, but, to my mind, the empirical results are more of a sanity check than meaningful results on their own. That is, I don’t think these results speak to a theory or debate in the field of cognitive science. Rather, as the authors acknowledge, this is more of a methods paper and I think the empirical results mostly demonstrate that the method is doing what it “should do.” For example, showing that subjects know more about a topic after watching the video or that knowledge about a topic “predicts” performance for similar questions is something that most people would take at face value. Likewise, the fact that questions mapped to fairly specific time points in the lectures is something that almost needs to be true. If the questions are drawn from the lecture content (which they are), then it would be a red flag if the method couldn’t map questions to timepoints in the lecture. Thus, for each of these results, if the model failed to show these effects, it would be a reason to tweak the model, not a meaningful result. (And I think the authors would mostly agree with all of this). In fact, there are quite a lot of degrees of freedom that went into getting results that “make sense.” This is, of course, more evident in the rebuttal than in the actual manuscript.

2. I think the question is whether the paper does enough, as a methods paper. (Again, I think this is where Reviewer 3’s concerns were, as well). As I note above, I think the authors have done enough on the statistics/analyses front to make a compelling case that the method “works.” That said, the paper does not introduce a new large language model, nor does it even use the latest models. Thus, I think the value as a methods paper comes down to the specific way in which existing models are leveraged to ‘map’ knowledge. Ultimately, I am not particularly convinced that there is much practical value in this method. The authors motivate the ideas in terms of teachers that might want to map what students know. Putting aside concerns about the actual feasibility of a teacher using this, the immediate question that came to my mind was: what is the advantage (gain) of this method relative to much simpler approaches. The authors compare their method against a strawman of a single test grade (e.g., 85%). The argument is that a single test grade makes it impossible to assess which specific topics or concepts a student has learned. That’s true, of course. But, for most of us that teach classes, it is very common (and almost a necessity) that we structure exams so that questions are organized around sub-topics. For example, we might deliberately include a certain number of questions per subject area. It is therefore quite trivial to have sub-scores by topic area. While I genuinely loved the visualizations that were presented in the current paper (again, it is beautiful work), I really struggled to see how or whether these visualizations provide more useful information than sub-scores on an exam. Even for the analyses that seek to predict performance on held-out questions, I was left wondering whether this method is any more sensitive than simple conditional probabilities based on sub-scores. For example, if a student answered the first 4 questions on a sub-topic correctly, what is the probability of answering question 5 correctly? What if they only answered 2 of the first 4 questions correctly? For a methods paper, I think that there is somewhat of a burden to demonstrate that the new method being presented outperforms existing methods. And, ultimately, I was not convinced that the current paper does this. Yes, the high-resolution coordinate spaces are impressive and they certainly give the impression of being much more nuanced than a sub-score, but it is harder to see the actual application of that. Can higher-resolution feedback be provided to a student or teacher? And would this higher-resolution feedback make a meaningful difference to learning outcomes? In these respects, there is no clear evidence or argument that this new method is superior to much, much simpler alternatives.

3. One way to demonstrate the value of the method would be to actually run a classroom experiment showing that students and/or teachers benefitted from the high-resolution maps compared to some simpler alternative. I realize the authors would argue that this is beyond the scope of the current paper (and I agree), but I think if the scope of the current paper is simply to demonstrate the method without demonstrating the actual value of the method, then it is necessarily a less significant contribution.

4. Relatedly, Reviewer 3 raises the point that it’s not clear the current model would outperform ChatGPT in the core function of assessing conceptual knowledge. I tend to agree. I do get the authors’ points that their method can track and map learning in a way that ChatGPT isn’t capable of (I agree with this), but, to me, it really comes down to the potential value of the method. This gets back to my question of whether the high resolution maps can actually be translated to some meaningful feedback or intervention. But if the goal is simply to have a user-friendly, powerful method for assessing conceptual knowledge, I think ChatGPT is probably the more appealing option.

(Remarks on code availability)

Reviewer #1 (Remarks to the Author):

Fitzpatrick and colleagues present a neat method for quantifying the content of real-world educational lecture videos and show that the resulting embedding space can be used to better understand and visualize quiz results assessing what knowledge students have learned. This manuscript is well-written and the figures are well-designed; I'm enthusiastic about the approach and the results are compelling. I have a couple clarifying questions about the methods, but my main concerns revolve around making sure the interpretations match the scope of the present data.

We thank the reviewer for their positive assessment!

1. I have two related high-level comments, both of which I'm having hard time articulating. First, I think the authors should be careful about the interpretative language they use throughout. Sometimes the language feels a bit lofty or overreaching. Let me try to provide an example: I'm not sure I fully buy the rhetorical parallel between "memorization" and "understanding." I don't think anyone in educational contexts really thinks of rote memorization as memorizing character strings the way a language model might. On the other hand, I don't really see how the topic model used in the present study (or even much more sophisticated language models) are doing anything above and beyond memorizing statistical (i.e. co-occurrence) structure (albeit at multiple scales). Sure, the model is not a simple, discrete lookup table, but I'm not sure we want to make any strong claim that the model captures "understanding" of physics either. If I understand correctly, the main advance from "memorization" toward "understanding" is made by embedding words in a continuous vector space, granting Shepard- (1987) style(?) generalization (i.e. the authors commitment that knowledge is fundamentally "smooth"). But the analyses simply demonstrate that the embeddings for a test set of words (i.e. quiz questions) can be interpolated from the embedding space learned from a training set of words (i.e. the video transcripts). Is this kind of interpolation fundamentally different from "memorization"? (A more radical claim here might be that much of what is seemingly "understanding" can be reduced to this kind of interpolation—which I generally agree with, cf. our Hasson et al., 2020 paper—but I doubt the authors really want to make that claim either.)

This is an important point, and one also raised by the other reviewers: to what extent are we capturing "understanding" as opposed to "memorization" with our approach? As an aside, this distinction is actually not the main claim we are trying to make in our paper—our comment about capturing understanding versus memorization was intended as an example of how these models might be *useful* (to educators, and so on). Nonetheless, there are a number of important details to unpack here.

First, we agree that the reviewer's example of memorizing character strings (like a language model might) is quite different from what we generally think of as "rote memorization" in educational settings. For example, memorizing a textbook to the point that a student could "regurgitate" the information on an exam still requires some basic *semantic* understanding of the material, as opposed to doing anything like learning co-occurrence statistics of the textbook's words.

As we clarify on page 3, we are not trying to claim that the topic models (or even other more sophisticated language models we could have used instead, like ChatGPT) are themselves “understanding” the material in any colloquial sense of the word. (We view those models more along the lines of the reviewer’s description—essentially learning co-occurrence statistics of the text.) Rather, our approach is to use the *embedding spaces* that these models define as a way of constructing *maps* of what people might know (or even think about). Specifically, the embedding spaces these models learn have the property that nearby coordinates (e.g., in Euclidean distance) tend to reflect related concepts. In that sense, the spaces these models learn are “smooth” (differentiable everywhere) representations that we can use to efficiently describe (and infer) what *people* know or understand. The property that ends up being most useful is smoothness. Because the embedding spaces (learned by text embedding models) have the property that nearby coordinates reflect similar concepts, we reason that *knowledge* or *understanding* about the concept reflected by a given coordinate might imply knowledge or understanding about related concepts—which will tend to be at nearby coordinates in the embedding space.

We view the question of whether the embedding space captured by our (or other) text embedding model(s) is a “useful” representation of someone’s knowledge as an empirical one. In other words, these maps are “useful” to the extent that they capture something predictive or informative about what people actually know. We show that we can predict individual participants’ answers to held-out questions (Fig. 6), so this tells us that something about what the model “considers” to be conceptually related is also predictive of how participants’ knowledge “spreads” across different concepts. We characterize this spread more directly in our revised manuscript (Fig. 7) by showing the distance (in topic space) over which the knowledge representations we derive using our approach provide predictive power over and above simply computing the proportions of correctly answered questions.

The deeper question of whether the knowledge maps we construct from participants’ quiz performance capture true understanding versus something closer to memorization is trickier to get at. We think this comes down to the specific *questions* that we asked participants, as opposed to something about the embedding spaces. If the questions themselves require deep understanding (e.g., answering them correctly requires something beyond what can be achieved through memorization alone), then the maps we learn from participants’ responses will also reflect deep understanding. If the questions themselves can be answered by memorization, then our maps will also reflect “memorized” concepts as opposed to true “understanding.”

The questions we asked participants in our experiment vary with respect to how much we think they might capture memorization versus deeper understanding. For example, we suspect that questions like “*In the famous equation attributed to Albert Einstein, $E = mc^2$, what does the letter ‘m’ represent?*” could be answered by something close to pure memorization. But other questions, like “*In your body, there are a tremendous amount of negatively-charged electrons. Your computer also contains a huge number of negatively-charged electrons. We know that like charges repel, but you and your computer are not repelled apart. Which of the following explains why?*” might require deeper understanding. The

“amount of understanding” a given question “requires” to answer is somewhat subjective, and might even come down to the “strategy” a given participant is using to answer that question at that particular moment. We have added a note to this effect on page 31 of our revised manuscript.

From a methodological standpoint, what we see as the major contributions of our approach are (1) a systematic means of mapping out and characterizing knowledge in maps that have infinite (arbitrary) numbers of coordinates, and (2) ways of filling those maps using relatively small numbers of multiple choice quiz questions. Our experimental finding that we can use these maps to predict responses to held-out questions has several psychological implications as well. For example, concepts that are assigned to nearby coordinates by the text embedding model also appear to be “known to a similar extent” (as reflected by participants’ responses to held-out questions; Fig. 6). This suggests that participants also conceptualize similarly the content reflected by nearby embedding coordinates. The “spatial smoothness” of participants’ knowledge (as estimated using quiz performance) in text embedding space is being captured by the knowledge maps we are inferring from their quiz responses (Fig. 7). In other words, our study shows that knowledge about a given concept implies knowledge about related concepts, and we also show how estimated knowledge falls off with distance in text embedding space.

To address these issues, we have added clarifying text throughout our revised manuscript. In addition, on page 8 we clarify how we interpret and use the text embeddings from the topic models, and on page 23 we clarify what we see as the major contributions of our approach.

*2a. Second, I think the authors should be very clear about the limits/scope of their approach, particularly based on the dataset used herein. For instance, in the Introduction, you use aspirational language about “the sum total of a person’s knowledge” and “the nearly infinite set of possible things a person could know”—but the actual data/model used here is relatively small-scale (and there’s no explicit evidence for scalability). I can’t imagine that the co-occurrence statistics across only <20 minutes of video lectures are particularly dense or rich. Just to confirm, the model is not pretrained in any way and does not incorporate any conceptual information from outside the lecture videos, right? For example, the authors say (line 492) “Conceptually, each topic is intended to give larger weights to words that are semantically related *or* tend to co-occur in the same documents” (my emphasis on “or”). Does the model know that words are semantically related above and beyond the co-occurrence statistics? My understanding was that “semantically related” is derived entirely from the co-occurrences statistics. In another example, you say “Knowledge estimates need not be limited to the content of the lectures.” But the embedding space is strictly limited to the co-occurrence structure extracted from these particular lectures—right?*

We have modified our text throughout our paper (including the statements the reviewer mentioned in the introduction) to clarify what we see as the scope of our work. As the reviewer notes, our current study does not show that we can map out *everything* someone knows; rather, we are focused only on a small “region” of what people know. To be clear, there *are* infinitely many coordinates in that small region of space (i.e., the space is continuous and differentiable everywhere within the convex hull formed by the union of all lecture timepoints’ and quiz questions’ embeddings). But of course the

space of concepts we focus on in our work does not (nearly) span everything someone could know, learn, or think about since, as the reviewer notes, we train the model using a relatively small number of documents that are (presumably) about a relatively small set of “discretizable” learning objectives.

We confirm that the embedding models are not pre-trained on data outside of the lecture transcripts and quiz questions. At a high level, the reviewer is correct that our topic modeling approach (Latent Dirichlet Allocation, or LDA) is driven by co-occurrence statistics. However, LDA does impose some additional structure on the embeddings beyond *pure* co-occurrence statistics (unlike, for example, Latent Semantic Analysis, or LSA, which is essentially like carrying out PCA on the dataset’s word counts matrix). LDA attempts to learn the themes or “topics” that *underlie* the observed text, and each document (observation) is cast as a weighted blend of topics (which are in turn defined as distributions of weights over words in the vocabulary). The key difference between LDA and “pure” co-occurrence based approaches is that LDA can do things like teasing apart different uses of the same word. For example, suppose a word like “bat” appears in a document. In the absence of other information, we could guess that the word might refer to a winged mammal, a piece of sporting equipment, or something one can do with eyelashes. Models like LSA would infer that themes or “factors” related to each of these themes were present in the given document. LDA, by contrast, uses other words in the document to parcel out which specific use of “bat” most likely accounts for our observation.

This ability of LDA to identify latent themes enables the model to generalize (better than models like LSA) to *new* texts, as long as those new texts include a set of words that overlap with the trained model’s vocabulary. In our implementation, although individual quiz questions might use different words to describe a given concept from a lecture, our model still “matches” questions with reasonable-seeming segments of video, when we compare the model’s embeddings of the videos and questions (Fig. 4). In response to one of the reviewer’s related comments (2c; see below), we have also run a new analysis showing that these LDA embeddings exhibit more temporally specific matches to the videos than obtained through simpler word count overlap.

Finally, the reviewer points out an ambiguity in our wording regarding semantic relatedness versus co-occurrence. We have clarified that sentence to read (emphasis added): “Conceptually, each topic is intended to give larger weights to words that are semantically related (*as inferred from their tendency to co-occur in the same document*).”

2b. Put another way, there’s a tension in the writing between the potential generality of the method and the targeted nature of the embedding space (i.e. tuned to the content of the two lectures) used herein. Here’s a specific example: In my understanding, the negative correlations between mismatching lecture and video topic weights (line 176) are an indicator that the embedding space reflects an “either this or that” structure specific to the two lectures. I have the intuition that near-zero correlation here would provide stronger evidence for “specificity” (line 179) than anticorrelation (as we see with the general physics questions). I wonder if this trivializes the dimensionality reduction results in Figure 7—i.e. could it be any other way? (Maybe this is fine!)

In general, we agree that there is some degree of “either this or that” structure to the embeddings. For example, when we look at the embeddings of the lectures and questions (Figs. 3, S2), several topic dimensions appear to be “specific” to *Four Fundamental Forces* (topics 2, 3, 8, 12, 13, and 15), whereas others appear to be specific to *Birth of Stars* (topics 1, 4, 5, 6, 7). There are also some topics that appear in the average embeddings of both lectures and their associated questions (e.g., topics 11 and 14). This also explains the “negative correlations” the reviewer is referring to. From our prior work, and that of other groups (e.g., Boyd-Graber et al.’s 2014 chapter on “Care and feeding of topic models”), this follows from a fundamental tradeoff in text embedding models, between generalizability and specificity.

To illustrate, let’s consider two approaches to fitting a given model. In both cases, we’ll imagine using the same model architecture, the same number of parameters, the same data set sizes, and the same inference procedure. The difference is that in hypothetical approach A, we’ll choose our training corpus to span a wide variety of content from a diverse set of sources. By contrast, in hypothetical approach B, we’ll choose our training corpus to contain documents that cover a relatively narrow scope. At a high level, since both models had the same number of observations and both models are of the same “complexity” (i.e., matching architectures, numbers of parameters, goodness of fit to their respective training datasets, and so on), these could be considered “equivalent” models. Another way to think about these models, though, is about the tradeoffs each makes between *generalizability* (i.e., something like the chances the model would be able to generate a “reasonable” embedding of some randomly selected held-out text) and *specificity* (i.e., something akin to the degree to which the embeddings are able to reliably differentiate between subtly different texts or concepts). Essentially, a model’s design, parameters (e.g., number of embedding dimensions), and the amount and quality of training data collectively determine the effective “resolution” of the model—i.e., related to the fidelity of the overall embedding space. But within those constraints, the choice of training corpus determines whether the embedding space spans a wide range of content at a relatively lower resolution (as in the “approach A” example above), or whether the embedding space “zooms in” on a narrower range of content at a comparatively higher resolution (as in the “approach B” example above). In other words, the particular set of documents we use to train an embedding model ends up determining the breadth and depth of the embedding space.

One way this can play out is in the sort of “either this or that” representations the reviewer is noticing in our model, as well as the tension they’re identifying between “generality of the method versus the targeted nature of the embedding space.” Since we trained our text embedding model on documents that were primarily “about” two similar but partially separable sets of concepts, the subtle conceptual variations within a video are each “assigned” several topic dimensions that enable the model to represent those variations within each individual video. That’s the aspect of the model’s features that looks like “switching” or like an “either this or that” representation. If we had used a greater diversity of documents in our training corpus, the within-video resolution would have suffered, and that “switching” behavior would have been more subtle.

More broadly, there are real challenges to building embedding models that are *both* sufficiently high-resolution within a given topic *and* sufficiently broad so as to cover a wide range of topics. For example, embeddings derived from even large modern models like BERT, GPT-{2,3,4}, LLaMa, and others, that are also trained on enormous text corpora, end up yielding surprisingly poor resolution within the content space spanned by a single course video. For example, here are some comparisons between basic visualizations from our paper derived from our topic modeling approach (LDA) versus an example large language model (BERT):

We highlight three general differences in the above figure. In the top row (“Trajectories”) we show a 3D projection (using PCA) of the content trajectories for each lecture (blue and green lines) and each question (blue, green, and purple dots). The LDA embeddings exhibit several desirable properties compared with the BERT embeddings:

- The LDA embeddings of the lectures and questions are “near” each other– e.g., the convex hull enclosing the two lectures’ trajectories is highly overlapping with the convex hull enclosing the questions’ embeddings. The BERT embeddings of the lectures and questions are instead largely distinct.
- The LDA embeddings of the questions for each lecture and the corresponding lecture’s trajectory are similar. E.g., the LDA embeddings for lecture 1 questions (blue dots) tend to appear closer to the lecture 1 trajectory (blue line), and the LDA embeddings for lecture 2 questions (green dots) tend to appear closer to the lecture 2 trajectory (green line). The BERT embeddings do not show this property.

In the middle row (“Per-question matches”), we display the correlations between each question’s embedding coordinate and the embedding of each moment of the corresponding lecture. Again, the LDA embeddings show several desirable properties compared with the BERT embeddings:

- The time series plot of individual questions’ correlations are different from each other when computed using LDA (e.g., the traces can be clearly visually separated), whereas the correlations computed from BERT embeddings of different questions all look very similar. This tells us that LDA is capturing some differences in content between the questions, whereas BERT is not.
- The time series plots of individual questions’ correlations have clear “peaks” when computed using LDA, but not when computed using BERT. This tells us that LDA is capturing a “match” between the content of each question and a relatively well-defined time window of the corresponding lectures. The BERT embeddings appear to blur together the content of the questions versus specific moments of each lecture.

In the bottom row (“Cross-lecture correlation matrices”) we compare the pairwise correlations between embeddings of lectures and questions within versus across content areas (i.e., content covered by the individual lectures, lecture-specific questions, and by the “general physics knowledge” questions). The LDA embeddings show a strong contrast between same-content embeddings versus across-content embeddings. In other words, the embeddings of questions about the lecture 1 (“Four fundamental forces”) material are highly correlated with the embeddings of lecture 1, but *not* with the embeddings of lecture 2, questions about lecture 2, or general physics knowledge questions. We see a similar pattern with the LDA embeddings of the lecture 2 (“Birth of stars”) questions. In contrast, the BERT embeddings are *all* highly correlated with each other.

Taken together, these comparisons illustrate how LDA (trained on the specific content in question) provides both coverage of the requisite material and specificity at the level of the content covered by individual questions. BERT, on the other hand, essentially assigns both lectures and all of the questions (which are all broadly about “physics”) into a tiny region of its embedding space, thereby blurring out meaningful distinctions between different *specific* concepts covered by the lectures and questions.

We note that these are not criticisms of BERT (or other large language models trained on large and diverse corpora). Rather, our point is that simple fine-tuned models trained on a relatively small but specialized corpus can outperform much more complicated models trained on much larger corpora, when we are specifically interested in capturing subtle conceptual differences at the level of a single course lecture or question. Of course if our goal had been to find a model that *generalized* to simultaneously capturing many different content areas, we would expect our approach to perform comparatively poorly relative to BERT or other much larger models. We suggest that bridging the tradeoff between high resolution within each content area versus the ability to generalize across many different content areas using a single set of model weights will be an important challenge for future work in this domain.

We have added a discussion of these issues to pages 23–25 of our revised manuscript, and have also added the figure above to our supplementary materials (Supp. Fig. 6).

2c. Another specific example: The authors want to make a strong claim that the model doesn't rely on surface-level similarity and that the “specific words” used in the quiz questions have “very little overlap” with the lectures? But there must be some overlap (i.e. co-occurrence), right? In my understanding, a completely disjoint set of words, none of which occurred in these video transcripts, would not map onto this embedding space in any meaningful way. To be clear, I don't think this is really a problem or a weakness of the method! —I'm just trying to articulate some discomfort when reading.

The reviewer is correct that there often is *some* overlap between the specific words used in the quiz questions and the corresponding parts of the lecture that discuss the “matching” concepts. What we meant (and have now attempted to clarify throughout our revised manuscript) is that the topic models provide additional characterization of the conceptual content of the lectures and questions *over and above* what we see using simple word matching alone.

To compare topic-based matching (correlation between topic vectors) with word-based matching (e.g., overlap between the words in a question vs. in a sliding window of a lecture's transcript), we implemented a simple measure of word overlap using Jaccard similarity. (The Jaccard similarity between text A and text B is computed as the number of unique words in the intersection of words from A and B, divided by the number of unique words in the union of words from A and B.) To carry out a fair comparison with our topic modeling approach, we ran the text through the same preprocessing steps we used prior to fitting the topic models, including removing stop words and

punctuation, converting digits to word-form, and so on. We can then compare the time series of correlation-based similarities (using text embeddings) versus the time series of word-based (Jaccard) similarities for each question and lecture. In the figure below, we show the time series plots for two example questions (rows; question text is displayed above the corresponding panels), each computed using either topic correlations (left panels) or Jaccard similarities (right panels):

The question in the top row (Question 5 about Lecture 1) is about comparing gravity and the electromagnetic force. When we use topic correlations to match up the question’s embedding with Lecture 1 timepoints (left column), we can see a clearly defined “peak” where the embeddings show strong correlations. When we examine the Lecture 1 transcript during that interval, we can see that the instructor is describing how those two forces relate (emphasis added):

“...disappears as an actual force as an actual interaction now the next the next force up the hierarchy which is one that we are more familiar with it is something it's what actually dominates most of the chemistry that we deal with and electromagnetism that we deal with and *that's the electro magnetic force we write it in magenta electro magnetic magnetic force and just to give a sense this is this is 10 to the 36 times the strength of gravity 10 to 36 times the strength of gravity* so it kind of puts the weak force in its place it's 10 to the 12 times stronger than the weak force so these are huge numbers that we're talking about either this relative that or even this relative to gravity and so you might be saying well you know *the electromagnetic force that's unbelievably strong why doesn't that apply over over over these these kind of macro*

scales like gravity let me write there macro scales macro scales why doesn't it apply to macro scales and actually there's nothing about the electromagnetic force why it can't it or it actually does apply over large distances the reality though is *you don't have these huge concentrations of either electric coulomb charges or magnetism the way you do mass so the mass that you have such huge concentrations it can operate over huge huge distances even though it's way way way weaker than the electromagnetic force the electromagnetic force what happens is because it's both attractive and repulsive* it tends to kind of sort itself out so you don't have these huge huge huge concentrations of charge now the other thing you might be wondering about is you know why is it called the electromagnetic force in our everyday life there's things like there's things like the Coulomb force that or the electrostatic force which we're familiar with positive charges or like charges want to repel if both of these were negative the same thing would be..."

In contrast, simple word matching (right column) identifies moments when any of the *words* in the question are used during the lecture (but not necessarily all together)-- e.g., mentions of "following," "difference," "gravity," "electromagnetic," and "force" at other moments in the lecture also evoke increases in Jaccard similarity.

Similarly, the question in the bottom row (Question 15 about Lecture 1) is about the Coulomb Force. Again, our topic correlations approach (left column) yields a well-defined match interval that corresponds to an interval in the lecture when the instructor is describing the Coulomb Force (emphasis added):

"...electromagnetic force in our everyday life there's things like there's things like the Coulomb force that or the electrostatic force which we're familiar with positive charges or like charges want to repel if both of these were negative the same thing would be happening and different charges like to attract we've seen this multiple times this is the Coulomb force or the electrostatic force electro static and then on the other side of the word yes you have the magnetic part and magnets you know you have you've played with magnets on your fridge they what's what you know if they're the same side of the magnet they're going to repel each other if they..."

On the other hand, simple word matching (right column) shows non-specific matches throughout the lecture to moments when the transcript includes "coulomb," "force," or "refer," but not necessarily all together (e.g., other uses of "force" will evoke an increase in Jaccard similarity, whether or not the lecture at those moments is specifically referring to the *Coulomb* Force).

We see analogous patterns for each question. Although the topic correlations are not always perfect (for example, our topic model appears to occasionally confuse forces that are strong with the Strong Force), using an explicit model of topics that uses co-occurrence patterns in the text to identify latent themes provides our analyses with structure that enables us to match up questions and lecture timepoints well in practice:

In the above figure, each dot denotes a single comparison between a lecture timepoint (top: Four Fundamental Forces; bottom: Birth of Stars) and a single question. The blue dots denote the correlations between topic vectors and the orange dots denote Jaccard similarities. In each case, the topic correlations exhibit a clear “peak” correlation at just a few well-defined moments in the lectures, and most of the remaining timepoints have correlations near 0. The Jaccard similarities, however, often exhibit much broader ranges of values, reflecting that Jaccard similarities yield matches that are less temporally specific (i.e., more diffuse) than those computed using topic correlations.

We have added a brief discussion of the benefits of using a topic model (versus simple word matching) on pages 25–26, along with three new supplemental figures (combining elements of the figures pasted into our response above; Supp. Figs. 3, 4, and 5) and a new supplemental table (Supp. Tab. 3) showing lecture text from the best-matching segments of the lectures, for each question.

3. How exactly were the questions constructed for each lecture? For example, Figure 2 suggests that the questions follow the trajectories of the lectures, at least to some extent; obviously this is sensible and “by design.” It’s great that the authors provide the full set of questions/answers (Table S1)! However, I think it’s important that the authors describe their thought process in designing the questions a bit more explicitly. For example, how exactly did they minimize “surface-level” similarity while ensuring enough conceptual overlap that the questions map onto the embedding space in a meaningful way?

One of our lab's undergraduate research assistants worked alongside a Masters student who was rotating in the lab for a month to develop a list of questions (these researchers are acknowledged in the paper for their valuable contributions to the work, but they did not meet the criteria for authorship discussed with all team members at the start of the project, as determined by JRM). The senior author (JRM) tasked the pair of researchers with coming up with "15 conceptual questions about each lecture, along with 9 additional questions about general physics knowledge." To help broaden the set of lecture-specific questions, the researchers were further instructed to work through each lecture in small segments, identify what each segment was "about" conceptually, and then write a question about that concept. The general physics questions were drawn from the researchers' coursework along with internet searches and brainstorming with the project team and other members of JRM's lab. The final set of questions (and response options) was reviewed and approved by JRM before collecting or analyzing the text or experimental data.

We have added a note to this effect on pages 30–31 of our revised manuscript. We have also toned down our language describing our question pool as reflecting "surface-level" versus "conceptual" details where appropriate, as we acknowledge that we did not have an explicit process for optimizing or designing questions to be "conceptual" beyond our own intuitions.

4. The authors correlated the topic weights for each question with the topic weights at each time point in the lecture, and in Figure 4 show that "each question appeared to be temporally specific" (line 208). Intuitively, this seems reasonable... But couldn't this result also be interpreted as creating tension with the idea that the questions aren't simply keying into "surface-level" similarities? Put another way, if I thought this approach captured some kind of holistic "understanding", I might hypothesize that good questions would integrate multiple concepts introduced across different parts of the lecture. (I might be conflating the overall validity of the approach for capturing "understanding" with the quality of the questions here.)

The reviewer raises a number of interesting questions here about whether the questions we asked participants are specifically capturing deeper understanding as opposed to solely capturing surface-level details. As we note above, since we don't have an explicit definition of what specifically differentiates "conceptual" versus "non-conceptual" questions, to some extent we end up relying on our subjective judgements in these matters. Our sense is that, across the set of questions we asked participants, there is substantial variance in how "deeply" each question probes conceptual understanding versus something more akin to rote memorization. We also discussed this issue, along with some example questions that we see as varying in "depth," in our response to the reviewer's comment 1. In summary, we acknowledge that at least some of our questions likely do *not* capture what we mean by deep understanding, and for other questions it is difficult to objectively say how deep an understanding is required. To address this point, we have toned down how we describe our experiment's questions throughout our revised manuscript, and we have also added a note to this effect on page 31.

The above said, we *can* say *some* things about what our approach is versus isn't capturing. For example, consider an embedding space that fails to capture any deeper semantic meaning (e.g., as in a "random" embedding space for which Euclidean or correlation-based distance was fully uncoupled from semantic meaning). In such a space, there would be no predictive value of the embeddings. In other words, knowing that a participant answered a given question correctly would provide no additional information (on average) about the participant's knowledge about the content covered by questions that were nearby in the embedding space as compared to questions that were far away. On the other hand, if an embedding space *does* capture similarities in semantic meaning in a way that follows similarities in participants' knowledge, then participants should have similar "levels of knowledge" about questions whose embeddings were nearby in the space.

Of course, knowledge itself cannot be directly measured; it can only be inferred "indirectly" through behaviors or other observations. But to the extent that participants' knowledge affects their chances of answering questions correctly, the embedding spaces learned by our topic modeling approach, and our framework for estimating what we call "knowledge," seem to have some explanatory power (in terms of predicting responses to held-out questions).

We also lack any direct means of defining or determining the "depth" of someone's knowledge. We found that the embedding coordinates of a held-out question, along with participants' performance on other questions at nearby coordinates, reflect how the participant will perform on the held-out question (Fig. 6). We also found that explanatory power falls off smoothly with distance in the text embedding space (Fig. 7). This suggests that we are capturing something more like "actual meaning" (as reflected in semantic relationships between different questions), over and above exact text matches (arguably the most basic framing of "surface level details").

With respect to the reviewer's point about how concepts may integrate across different timepoints, we do not see temporal specificity as necessarily in tension with how "deep" or "complex" (or even "holistic") a particular concept is. We see this as depending on several factors including what the concept itself *is*, the particulars of the lecture in which the concept is explained or described, the instruction style, and so on. Presumably the temporal specificity also depends on the text embedding model's sensitivity to distinct aspects of the same overarching concept.

Another way of considering the reviewer's comment about temporal specificity versus depth may be captured in part by our response to their comment 2c, above. In our response to that comment, we report on an analysis specifically aimed at comparing the temporal specificity of question-lecture matches computed using topic models versus simpler measures of word overlap. We found that our topic modeling approach, which inherently captures deeper structure and co-occurrence patterns than word overlap alone, results in more temporally specificity. This tells us that, at least for those two example approaches, greater temporal specificity is consistent with a model that captures *more* semantic detail (which one could interpret as a "deeper" semantic representation, or as reflecting greater nuances in "similarity").

5. *The authors used windows comprising 30 lines(?) of the transcript to define documents for input to the topic model. This seems like an important parameter as it impacts the co-occurrence structure observed by the topic model, right? How was this number 30 chosen? I assume it represents some happy medium between providing a wide enough window to yield rich co-occurrence structure while retaining some temporal specificity. Transcript "lines" seems like a somewhat arbitrary unit; I assume it relates to how YouTube renders the transcript? How many words tend to occur in each of these windows? What's the average duration of these windows in seconds?*

The reviewer is raising a number of important questions about how we chose the "parameters" of our general approach. In our prior work (Heusser et al., 2021, Nature Human Behaviour), where we developed a similar "topic modeling of sliding text windows" approach, we were computing text embeddings of human-generated annotations. We reasoned in that work that each sentence of the annotations could be treated roughly as a "unit of thought" for the purposes of our analyses. Essentially choosing the level of resolution of the analysis (letters, words, sentences, paragraphs, entire documents, etc.) determines the "resolution" at which differences between successive observations can be detected or characterized. Sentences seemed like a reasonable choice in that prior work.

In our present study, we tried to follow our prior approach as closely as we could, but due to differences in the text and requirements of our approach, some adaptations were needed. For example, in our present work we used automatically generated (via speech to text algorithms) transcripts that did not include punctuation. Further, Khan Academy lecture videos are often narrated in a sort of "stream of consciousness" style that often doesn't map cleanly onto sentences per se. Instead, we treated each "line" of the automatically generated transcript as a stand in for a "sentence." To give a sense, each line of the automatically generated lecture transcripts corresponds to a single block of closed captioning text that is displayed on-screen during a given part of the lecture. We selected a window length of 30 "lines" in our current paper to match the same average number of words per sliding window (before preprocessing, rounded to the nearest word) as in the sentence-based transcripts we used in our prior work (Heusser et al., 2021). The sliding windows had an average of 73.8 words and lasted for an average of 62.22 seconds.

We have added some additional explanation of how we chose the sliding window lengths in our study, along with the average word count and duration per window, to page 33 of our revised manuscript.

6. *I don't fully understand the motivation for choosing 15 topics for most of the analyses, or why you increase to 100 topics for Figure 7. I don't think there's anything wrong with picking a somewhat arbitrary round number in these cases; just curious if there's any particular reasoning behind the present choices.*

This choice was motivated by some observations about how UMAP works and some choices that affect how accurately the embeddings of the data in the low dimensional space reflect the original

high-dimensional data. To give some context, the UMAP algorithm is what we use to visualize the 2D “maps” by converting correlations in the original (high-dimensional) topic space onto Euclidean distances in the 2D map space. This property makes it possible to intuitively visualize the similarity relations (correlations between topic vectors) we think are most reflective of actual semantic similarities. However, in “solving” for the mapping between the high and low dimensional spaces, UMAP ends up distorting the data in a few different ways. Some of these distortions, such as information loss when we move from high-dimensional to low-dimensional representations, are unavoidable. But others appear to depend in part on properties of the original high-dimensional data, including the number of dimensions, number of observations, etc., and on the number of dimensions in the target low-dimensional space. We noted two types of distortions in particular. First, UMAP has a tendency to “snap” nearby points in the original high-dimensional feature space onto the same (or very nearby) coordinates in the low-dimensional projection. Second, because the low-dimensional embedding is computed using clusters of nearby observations in the original (high-dimensional) space, observations in the low-dimensional space can be “clumped” so that they appear closer together in the low-dimensional projection than they are in the original space. We can see how these two types of distortions appear in the 2D projections of 15 versus 100 dimensional text embeddings of the same lectures and questions:

When we “connect the dots” between successive observations in the lectures (left column), both sets of trajectories look “reasonable” (e.g., they’re relatively smooth, the two trajectories are partially overlapping but mostly distinct, the questions “about” each lecture correctly map onto the appropriate parts of the 2D space, and so on). But when we examine the embeddings of the individual timepoints of the lectures (right column), it’s clear that there are more “gaps” in the 15-topic version. Since we know that both trajectories are smooth in the original high-dimensional feature space (because we computed them using overlapping sliding windows and resampled each trajectory to have a constant sampling rate), it’s clear that the 100-topic versions more accurately reflect how the original high-dimensional data behaves. We thought fewer gaps would also be desirable when visualizing the 2D knowledge maps, which require interpolating from nearby observations (so gaps in the data

require more interpolation and less reliance on actual observations). To select an appropriate number of topics (k) for our model, as a starting point, we identified the minimum number of topics that yielded at least one “unused” topic (i.e., in which all words in the vocabulary were assigned uniform weights) after training. This suggested that the number of topics might be sufficient to capture the set of latent themes present in the two lectures. We found this value to be $k = 15$ topics. We found that with a limited number of additional adjustments following Boyd-Graber et al. (2014; e.g., removing corpus-specific stop-words), the model yielded (subjectively) sensible and coherent topics.

Although the above reasoning suggests that the 100-topic embeddings are better suited for creating the 2D knowledge map visualizations, we also identified several important differences in the topic activation patterns when we used 15 vs. 100 topics:

- In the 15-topic model, topic activations persist for longer durations than in the 100-topic model. We thought that these longer durations might boost the sensitivity of our approach by integrating over longer spans of time (e.g., under the assumption that the estimated topic weights are “noisy”).
- In the 15-topic model, topic activations often overlap (e.g., in a given timepoint, several topics often have large weights), whereas this happens only rarely in the 100-topic model. This property of the 15-topic model seemed potentially useful for capturing interesting co-variation across topics.
- The 15-topic model (roughly) “segments” the lectures into on the order of 5-10 chunks (roughly one per minute). When we examine the temporal correlation matrix, we can see roughly 5-10 square “blocks” of strong correlations along the diagonal. The 100-topic model segments the lectures much more finely (approaching around 10 seconds per “block”). Intuitively, we wanted to capture broad concepts that we thought might take on the order of a minute to “explain,” as opposed to much lower-level concepts that might be mentioned in a few seconds. This again led us to prefer the 15-topic model in our main analyses.

These patterns may be seen in the figure below (left: 15-topic model; right: 100-topic model; first and third columns show topic activations over time; second and fourth columns show the temporal correlation matrices; top row corresponds to the *Four fundamental forces* lecture and the bottom row corresponds to the *Birth of stars* lecture):

We briefly discuss some of this logic on pages 18–19. We note that all of our *analyses* rely on the 15-topic model, and we carry out the relevant comparisons, distance computations, and statistical tests in the original 15-dimensional space for that model. We used the 100-dimensional model solely for visualization, when we created the knowledge maps (Fig. 8).

Minor comments:

Regarding my previous comments on scalability, do the authors think this kind of approach could be extended from topic models to the contextual embeddings learned by large language models? Could be worth mentioning if this is a valid future direction, given the current popularity of these models. My guess is that if we want a method that can capture “the sum total of a person’s knowledge” and “the nearly infinite set of possible things a person could know,” large language models are currently the best approach.

We share the reviewer’s intuition that large language models may be the most promising way to scale up our approach to a broader range of concepts. As we noted in our response to the reviewer’s comment 2b above, simply swapping in a large language model trained on a much larger and more diverse corpus does not seem to “just work.” Rather, as we discuss above, we’ve found an overarching tradeoff: when we include a larger and more diverse set of content in our training corpus, or when we used substantially larger models (e.g., BERT), we found that the resulting models are more generalizable to broad ranges of content, but less able to pick up on low-level details (e.g., at the level of individual questions or small segments of individual lectures). We’ve been starting to think through how one might construct models that work well at a variety of conceptual scales. One idea we’ve had is to potentially combine multiple models into a single “hybrid multi-scale model.”

Another is to re-think how the embeddings themselves are used internally by the models, or how we use those embeddings to construct maps. While we're excited about these possibilities, we also feel that they're beyond the scope of our present study.

In Figures 2 and 7, you use PCA to visualize two- and three-dimensional projections of the embedding space. What proportion of cumulative variance do these first two and three PCs account for?

In Figure 2C, we use PCA to visualize the 3D projections of the embedding space. Those first 3 components explain 36.2% of the variance in the original embeddings of the lecture videos and questions. In Figure 7 we use UMAP (not PCA) to visualize the embedding spaces as 2D maps. Unlike with PCA, there's no straightforward way of estimating variance explained by UMAP, since it is a form of nonlinear dimensionality reduction. (Leland McInnes, the lead author of the UMAP algorithm, explains this issue further here: <https://github.com/lmcinnes/umap/issues/122>).

All participants saw the Four Fundamental Forces video first, followed by the Birth of a Star video, right? Should we be worried about some order effect?

All participants in our study viewed the two videos in the same order. Therefore it's of course possible that there might be order effects that we are not accounting for (i.e., we did not counterbalance the viewing order). That said, we are not aware of any reason why viewing order should specifically influence or affect our main findings.

If the two lectures had a dependency relationship (e.g., understanding one lecture required first having learned the content of the other) then this might introduce more serious order effects. But we specifically selected the two lectures in our study to be self-contained (and therefore conceptually independent of each other). We mention this as one of the criteria for choosing these two particular videos on page 6.

Line 283: These t-values have 600+ degrees of freedom; I assume this is determined by the total number of questions across all participants. Should we be worried that this kind of statistical test doesn't take into account subject-level variability?

We generate the knowledge estimates for each held-out question using only the same participant's responses to other questions on the same quiz (Fig. 6). So there is no "leakage" across participants in our predictions (e.g., the fact that other participants tended to get a particular question wrong will not inform our predictions of how the *current* participant answered that question). Our statistical tests are carried out on the full distribution of knowledge estimates; specifically, we compare the estimated knowledge of held-out questions that were actually answered correctly vs. held-out questions that were actually answered incorrectly. Given that our overall approach is already relatively complicated, we opted for a simple and easy-to-interpret statistical test, rather than attempting to model participant-level effects or other factors (e.g., of question identity, content area, etc.). While an

imperfect solution, we made an incremental improvement to our statistical framework by using non-parametric Mann-Whitney U-tests instead of t-tests.

Line 492: Do you use a specific set of stop words? (e.g. the NLTK package has a standard list of stop words)

We use the NLTK package's "English" stop words list, augmented with the following additional list of words (identified following Boyd-Grayber et al., 2014): ['actual', 'actually', 'also', 'bit', 'could', 'e', 'even', 'first', 'follow', 'following', 'four', 'let', 'like', 'mc', 'really', 'saw', 'see', 'seen', 'thing', 'things', 'two']. We clarify this point on page 33.

Line 557: "insure" > "ensure"

Fixed!

Reviewer #2 (Remarks to the Author):

The authors develop a framework for characterizing the acquisition of knowledge across time, specifically in the context of educational lectures. Participants watched two physics-related lectures on the Khan Academy platform, and completed quizzes before the first lecture, in between lectures, and after the second lecture. Questions on all quizzes tapped into knowledge from the two distinct lectures (as well as general physics knowledge). The information contained in the lectures and associated quiz questions was embedded in a topic space derived from word co-occurrences contained therein. This approach enabled the authors to (1) define each of the two lectures as a trajectory through this embedding space, (2) quantitatively determine the differences in conceptual content between the two lectures (by assessing topic weight and variability), (3) match quiz questions to specific moments in the lecture with relatively high resolution, (4) determine how much knowledge participants had about different points in the lecture, (5) predict whether a participant got a question correct or incorrect based on their estimated knowledge at that relevant coordinate in the embedding space, and (5) visualize how knowledge spreads through the embedding space as a result of watching the lectures. I am convinced that this framework yields more informative feedback on students' quiz results, relative to calculating percent correct. I find the authors' approach to be an exciting one, and that it provides methods others can use to quantify and visualize the specific content of one's knowledge and how it might change over time. The methodology is sound, and the introduction and discussion are written well. This work primarily proposes a methodological framework rather than presenting new empirical results. Despite my overall enthusiasm, some other comments/suggestions are summarized below.

We appreciate the reviewer's summary and positive overall assessment.

1. It is mentioned multiple times that correspondences between quiz questions and lectures (e.g., localizing each quiz question to a specific point along a lecture's trajectory) are not superficial, because "the lectures and questions used different words." As far as I can see, there is no quantification of word overlap between lectures and quizzes, and we are supposed to take the authors' word on this. I find it difficult to believe that there is no word overlap between lectures and quiz questions, and would like to see some quantified measure of word overlap. The degree to which

correspondences in the embedding space are “superficial” or “deep” is unclear and as it stands, I am not fully convinced that the current analyses tap into deep conceptual knowledge.

Reviewer 1 raised a similar point about comparing word overlap in topic correlations in their comment 2c, above. In brief, to explore this we ran a new analysis to explicitly compare word overlap with topic vector correlations. We used Jaccard similarity (number of unique words in the intersection divided by number of unique words in the union) to quantify word overlap. As shown in this figure (copied below for convenience), the topic vectors for different questions often match up with a temporally well-defined segment of the corresponding lecture, whereas word matches alone are more diffuse:

Essentially, this is because simple word matching (right column) identifies moments when any of the *individual* words in the question are used during the lecture, whereas topic vector correlation identifies moments when the *underlying themes* (most probably) expressed by the question’s specific combination of words were also expressed in the lecture. For example, in the first question (top row), mentions of “following,” “difference,” “gravity,” “electromagnetic,” and “force” at other moments in the lecture also evoke increases in Jaccard similarity.

Reviewer 1 also raised a similar point about whether our approach is capturing “superficial” or “deep” knowledge (comment 1, above). For convenience, we’ve copied the relevant portion of our response to that comment below:

The deeper question of whether the knowledge maps we construct from participants' quiz performance capture true understanding versus something closer to memorization is trickier to get at. We think this comes down to the specific questions that we asked participants, as opposed to something about the embedding spaces. If the questions themselves require deep understanding (e.g., answering them correctly requires something beyond what can be achieved through memorization alone), then the maps we learn from participants' responses will also reflect deep understanding. If the questions themselves can be answered by memorization, then our maps will also reflect "memorized" concepts as opposed to true "understanding."

The questions we asked participants in our experiment vary with respect to how much we think they might capture memorization versus deeper understanding. For example, we suspect that questions like "In the famous equation attributed to Albert Einstein, $E = mc^2$, what does the letter 'm' represent?" could be answered by something close to pure memorization. But other questions, like "In your body, there are a tremendous amount of negatively-charged electrons. Your computer also contains a huge number of negatively-charged electrons. We know that like charges repel, but you and your computer are not repelled apart. Which of the following explains why?" might require deeper understanding. The "amount of understanding" a given question "requires" to answer is somewhat subjective, and might even come down to the "strategy" a given participant is using to answer that question at that particular moment. We have added a note to this effect on page 31 of our revised manuscript.

2. I would like to see the "Estimating Knowledge" analyses to go a step farther. The authors estimate participants' knowledge at the position of each quiz question's coordinate in a leave-one-out fashion, and show that these estimates reliably track whether the question was answered correctly or incorrectly. However, this analysis currently includes within-lecture prediction e.g., Lecture 1 quiz question performance to estimate a left-out Lecture 1 question. This analysis would be much more convincing and empirically interesting if a cross-lecture approach was used. That is, if for Quiz 2, performance on Lecture 1 questions could be used to predict accuracy on (unwatched) Lecture 2 quiz questions. This would be more convincing evidence that the embedding space is capturing deep conceptual meaning.

We've carried out the analysis the reviewer suggested (see Fig. 6, also copied below). In the previous version of our analysis (corresponding to the top row of the figure, labeled "All questions"), we used all (held-in) questions (about lecture 1, lecture 2, and general knowledge) from each quiz to predict knowledge on a single held-out question.

The reviewer’s suggested analysis appears as a panel in the “Across-lecture” part of the figure (middle rows). Specifically, they suggest using Quiz 2 responses to Lecture 1 questions to predict accuracy on Lecture 2 questions from the same quiz (i.e., before Lecture 2 had been watched). This corresponds to the second row and middle column of the “Across-lecture” part of the figure. Indeed, we see that the predicted knowledge for held-out Lecture 2 questions that participants answered correctly is significantly higher than for held-out Lecture 2 questions that participants answered incorrectly. We also carried out other variants of this analysis, whereby we predicted knowledge for Lecture 1 questions using responses to Lecture 2 questions, and we also carried out the analysis separately for responses to each of the three quizzes. As the reviewer suggests, we see this as showing how the predictions obtained using our model can generalize across different content areas.

For completeness, we also carried out a third general class of test (“Within-lecture”), whereby we used the responses to Lecture 1 questions to predict knowledge for a held-out Lecture 1 question from the same quiz. We repeated this procedure using Lecture 2 questions to predict knowledge for a held-out Lecture 2 question, and for each of the three quizzes. This latter analysis was suggested by Reviewer 3, and helps to demonstrate the *specificity* of the predictions— i.e., the ability to generate predictions about conceptual knowledge at the resolution of different concepts covered within the *same* lecture: In addition to the analysis suggested by the reviewer, their comment led us to consider the broader question of “how far in the text embedding space knowledge estimates extend.” For example, suppose we know that a participant answers a question (at embedding coordinate X) correctly. As we move away from X in the embedding space, how does knowledge (as estimated by quiz performance) “fall off” with distance? Or, suppose the participant instead answered that same question *incorrectly*. Again, as we move away from X in the embedding space, how does what the participant *doesn't* know about the content change with distance? We reasoned that, assuming our space is capturing something about how participants actually organize their knowledge, conceptual knowledge right around X should be similar to the participant’s knowledge of the content at X . And at another extreme, at some distance (after moving sufficiently far away from X), our guesses about what participants know (based on their response to the question at location X) should be no better than guessing based on their overall proportion of correctly answered questions—i.e., if Y is very far away from X , all we can do with the participant’s response to X is guess that “their performance on quiz questions about Y is about equal to their average performance on quiz questions about *any* material.” With these ideas in mind, we asked: conditioned on answering a question correctly, what proportion of *all* questions (within some radius r of that question’s embedding coordinate) were answered correctly? We could then plot the proportion as a function of r . Similarly, we could ask, conditioned on answering a question *incorrectly*, how the proportion of correct responses changed with r . We found that quiz performance falls off smoothly with distance, and the “rate” of the falloff doesn’t appear to change across the different quizzes, as measured by the distance at which performance becomes statistically indistinguishable from a simple proportion-correct score (Fig. 7):

Taken together, this new analysis suggests that participants' knowledge (as reflected by their quiz performance on specific questions) changes relatively smoothly and gradually across our text embedding space.

3. In the current "Estimating Knowledge" analyses, I'm having a hard time understanding what is happening in Quiz 3. It makes sense that there's no difference in the correct/incorrect distributions in Quiz 1, and it's great that it emerges in Quiz 2, but I would have also expected to see the effect for Quiz 3 (and indeed a weak effect was found). The authors explain this as a "ceiling effect", but it doesn't look like the knowledge estimates are at ceiling in Fig6-Quiz3. Further, if the knowledge estimates based on the embedding space are not sufficiently predicting quiz question accuracy, then what does? In other words, if the knowledge estimates for all regions of the space were high, then why are the participants' responses wrong? Is the model not capturing other pertinent knowledge, or do the authors think participants were choosing answers randomly?

There are a few things going on here. First, as the reviewer notes, we do see a reliable effect for Quiz 3 (the test the reviewer is referring to is in the upper right panel of the figure we pasted in with our response to the previous comment).

In our prior submission, we speculated that the reason Quiz 2 responses were "easier" to predict than Quiz 3 responses had to do with the "contrast" between the highest and lowest knowledge within the embedding space. When participants take Quiz 2, they tend (on average) to have relatively high knowledge for Lecture 1 content and relatively low knowledge for Lecture 2 content. On the other hand, when participants take Quiz 3, they tend (again, on average) to have relatively high knowledge *everywhere* in the embedding space. Our "ceiling effect" comment was meant to reflect that pattern, but we can see how the wording was confusing. We ended up entirely re-writing that part of our results section (pages 14–17).

Regarding the reviewer's question about why some participants' responses are still wrong on Quiz 3, this is simply a reflection of the difference between knowledge on *average* versus knowledge for an *individual participant*. As the reviewer points out, and as shown in Figure 8, the average "Quiz 3" knowledge map shows relatively high levels of knowledge throughout the embedding space. Further, there is relatively little contrast between the most and least known content in the average map for Quiz 3 (e.g., as compared with the Quiz 2 map). However, when we examine the Quiz 3 knowledge maps for individual participants (Supp. Fig. 9), we can see that there is substantial variability:

Knowledge: Quiz 3

As shown in the individual maps, not all individual participants are at “ceiling” even after watching both lectures. That we can predict knowledge for individual held-out questions shows that (at least to

some extent), participants are *not* simply responding at random. Rather, their responses appear to track with our predictions about what they do versus don't know.

4. I don't find the "Moment-by-moment" plots in Figure 5 to be useful, and in a sense they are misleading. With time on the x-axis, the implication is that the plot will be showing learning over time, as the lecture unfolds. This leads to the initial question of *why* the plots don't show an increase over time. However, this analysis does not reveal increasing knowledge over time, but rather the degree of knowledge relating to each moment in the lecture, assessed after the lecture was completed. What extra information are we getting from these moment-to-moment plots, that we're not getting from the averages presented in Fig 5B and 5D? Especially since there's no indication of how topics are changing from moment to moment, these plots seem superfluous. An alternate possibility is to show what topics participants know in each of the quizzes.

As the reviewer notes, the Figure 5 time course plots (Figs. 5A and 5C) show "*the weighted proportion of correctly answered questions about the content reflected in each moment of the lectures*" (Fig. 5 caption). The "averages" presented in Figures 5B and 5D show a sort of corrected version of an overall "proportion-correct" score for each of the three quizzes (the correction accounts for the fact that the questions do not tile the lecture content perfectly evenly). The time courses in Figures 5A and 5C show how much participants "know" about different *parts* of the lectures at the time they took each quiz. This can reveal interesting differences across the different content presented in the lectures. For example, before watching Lecture 1, participants tend to be relatively unknowledgeable about the content presented around the 3–4 minute range of Lecture 1 (as compared with, say, the content presented around the 0–2 minute range, or the 6–7 minute range). *After* watching Lecture 1, however (when participants take Quizzes 2 and 3), we see two main effects. First, participants' knowledge about every part of the Lecture 1 material is (on average) greater, relative to before they watched Lecture 1. Second, not all of their knowledge increases by the same amount. In particular, on average, participants' knowledge about the content from minutes 3–4 of Lecture 1 ends up being even *greater* than their knowledge about the content from 0–2 or 6–7 minutes. This suggests that some aspects of knowledge might be more "malleable" than others. This could reflect something about how the content is presented, an interaction with other aspects of participants' knowledge, or some other factor.

We see this ability to estimate participants' knowledge for individual moments of the lectures as an important advance of our method, enabled by our ability to map each quiz question onto the specific parts of the lectures that conveyed information relevant to correctly answering it (Fig. 4). For example, one could imagine these time-resolved knowledge estimates as the basis for an automated tool that helps students identify specific sections of course content to review, based on their performance on simple quizzes similar to those used in our study. Real-world instructors might also find value in identifying systematic differences in how successfully a class learns different sections of a lesson's content. In terms of the information that the time courses shown in Figure 5A and 5C provide over and above the averages shown in panels B and D, one could imagine two different series of moment-by-moment knowledge estimates: one in which estimated knowledge is generally low for

moments from the first half of the lecture and high for moments from the second half, and another in which estimated knowledge is generally high for the first half and low for the second half. Averaging both of these time courses across moments as we do in 5B and 5D would result in mean knowledge estimates that are similar or identical, despite the time courses themselves being meaningfully different. Importantly, if these two time courses of knowledge estimates were derived from different individuals (or groups of individuals), both the contents of those individuals' knowledge and what an instructor could do to most effectively help them "fill in gaps" in their knowledge would also be meaningfully different.

Although we continue to see the Figure 5A and 5C time course plots as interesting and informative, we do take the reviewer's point that they can be misinterpreted. To help clarify, we have updated the figure's caption to make it more explicit that we are estimating knowledge about the *content* presented at each moment of the lectures, as opposed to estimating "how much is known *overall*" at each moment of training.

5. The Topic Weight Variability analysis (and corresponding Fig 3A) feels a bit lackluster. The authors have set up this rich, high-dimensional space in which to perform their other analyses, so it would be more satisfying if this analysis was also done using a geometric approach. For example, could variance be captured by the distance spanned by a lecture's trajectory in each of the 15 dimensions? Could variability be visualized in 3D space? I also think the decision to visualize the topic variance, rather than topic weights, could be better justified – it's not clear in the manuscript's current form why we should care more about variance than weights, which is more intuitively relevant.

We appreciate the reviewer's feedback and we share their enthusiasm for an aesthetically pleasing presentation. We have explored several ways of displaying the content of Figure 3A. We can provide some additional background to clarify our process, and our decision to ultimately maintain the prior presentation format.

Geometrically, topic overlap simply looks like "points that are nearby." We show in Fig. 2C how the lectures' and (individual) questions' embeddings relate when rendered in 3D. The challenge is in capturing the full embedding space, since it's difficult to visualize more than 3 dimensions. Our "bar graph" approach (while, we admit, is less "flashy") lets us zoom in on each individual topic dimension, which shows more completely the specific dimensions along which lectures and questions are similar or different. This is in contrast to projecting the lecture trajectories or questions into a lower-dimensional (e.g., 3D) space, where information is necessarily lost.

We also appreciate the reviewer's comment about topic variability versus topic weights. To clarify, both approaches yield similar looking visualizations (we provide the "topic weights" version of Fig. 3 in Supp. Fig. 2). That said, we think that the "variability" version is closer to what we *mean* when we think of two texts being "about" the same thing. One reason why weights alone can be misleading is somewhat technical and specific to constraints placed on LDA's topic vectors. Because topic vectors

must sum to 1, the number of topics “present” in any given document will influence the weights of the other topics. Therefore the raw (absolute) weights can conflate the presence of a given “theme” with the *absence* of other themes.

Even beyond the constraints that are specific to LDA, if a given topic (or a feature dimension of some other embedding model) is consistently set to the same (large) value across all timepoints in a lecture, it can’t express any specific concepts in the lecture that span less than its entire duration. But if a given topic shows up during a specific interval in one lecture, that tells us that topic has something to do with whatever content was covered during that interval. We want to be able to pick out those sorts of “transient” patterns that reflect concepts at a lower level than “what the entire lecture is about on average.” So we think that the notion of identifying topics for which some moments (or questions) are weighted strongly and others are weighted *weakly* is more indicative of those topics being important. In other words, that contrast between strong and weak weights that depend on the specific content being covered is what we see as diagnostic of the topic being “important.” In practice, the two measures (variability vs. means) end up being more similar than we had anticipated (e.g., compare Fig 3 vs. Supp. Fig 2).

6. *This a relatively minor qualm, but referring to a coordinate in the embedding space as “a single concept” or “individual concept” doesn’t seem quite right (e.g., “We wondered whether our modeling framework might enable us to...infer participants’ knowledge at the scale of individual concepts (e.g., as captured by a single moment of a lecture). The notion of a single concept implies that concepts are units of some kind, and capturing conceptual knowledge in a high-dimensional space runs counter to that view. Further, what does a “single concept” mean if the authors count “fish”, “water” and “fish swim in water” to each be a concept (p. 3)? In this case, a full sentence would count as a concept, or even the information contained in a whole document. Maybe the authors take this view, but it feels particularly odd given the initial use of “fish” and “water” as initial examples.*

This question of what a concept “is” is an important one. There are many potential ways to define what a concept is, and to construct hierarchies of subordinate and superordinate concepts. We view this as an ontological question far beyond the scope of the current manuscript, so in our manuscript we chose to treat a “concept” as equivalent to a text embedding coordinate. Since any amount of text can be embedded, our notion of a “concept” as framed in our paper is consistent with all of the examples the reviewer proposes—a single word, a phrase, a sentence, a paragraph, an entire document, or even a collection of documents. One way to “think about” what a concept is, in this framing, is to read word clouds of the sort presented in Figure 7C. If each individual word evokes some mental representation, then a “concept” is the weighted sum of the representations of all the words in the cloud (where the weights are given by the words’ sizes). Under this framing, most concepts are not easily namable (at least not as single words or even short sentences). Still, this framing *is* still compatible with *some* concepts being nameable. For example, an “edge case” word cloud might contain a single word (e.g., “fish”), in which case the “meaning of that single word” is a “concept” by our definition. So in that sense, our “fish” and “water” examples aren’t incompatible with our overall view of what concepts are. Nonetheless, we take the reviewer’s broader point that some clarity is

needed. We have added a note about what we mean by “concept” (within the context of our paper) on page 3:

“...‘concepts’ are defined implicitly by the model’s geometry (e.g., how the embedding coordinate of a given word or document relates to the coordinates of other text embeddings; Piantadosi and Hill, 2022)”

Reviewer #3 (Remarks to the Author):

The authors apply a computational method called “topic modeling” to text transcribed from instructional videos (on two topics in physics). The result is a quantitative representation of the content of each moment of the video in the form of vectors embedded in a multidimensional space. They then represent the content of questions about these lectures in the same space. They report that this data-driven modeling approach (1) can match questions to their appropriate lectures, and even to the specific section of the lecture which covers the relevant content; (2) can capture how the knowledge of human learners evolves from pre-lecture to post-lecture; (3) estimate whether a learner will answer a question correctly or not based on their answers to other, related questions; (4) estimate the learner’s knowledge of parts of the embedding space that were not explicitly covered by the lectures (or questions).

Whereas I find the overall approach interesting, I do not believe this work, in its current form, merits publication in this journal. The main reason is that I think there is a discrepancy between the claims that the authors make and the data they present. Specifically:

1. Most generally, throughout the paper different research goals are mentioned, and I found it somewhat difficult to discern what exactly the authors were aiming to do. The paper would benefit from clarifying the research goals. Of course, a single paper can achieve multiple goals, but in such a case it would be useful to state all of them together.

Our primary research goal is to advance our understanding of what it means to acquire deep, real-world conceptual knowledge (page 4). We break this down into two sub-goals (page 5): first, we want to gain detailed insights into what learners know at different points in their training. Second, we want our approach to be potentially scalable to large numbers of diverse concepts, courses, and students.

To state these goals differently, we want to map and track the acquisition of complex conceptual knowledge, like what students might learn in a course. Accomplishing this goal required developing an approach to *representing* knowledge (when the “number” of unique concepts one might want to consider is nearly infinite). We also wanted our approach to easily transfer to other sorts of courses, e.g., as opposed to being “hard coded” or as opposed to requiring human-generated labels.

1.1. The authors might be trying to come up with a new tool that would be applicable to educators. If application is one of the goals, I am not sure why the authors are not using available technology that can be more quickly adopted, such as large language models (LLMs) like GPT-2. Activity vectors from hidden layers in such models are available, so text excerpts from lectures and questions could be all embedded in the space of hidden activity, and it’s plausible that those embeddings would be just as good as those obtained with the topic model. Moreover, such a model would not have to be re-trained for every new domain (whereas the author’s model would likely have to be re-trained on

topics that are far from physics, such as literature, history, political science, computer science, etc.). One might argue that LLM representations are uninterpretable, in contrast to the “topics” (dimensions) of a topic model, but the current paper seems to have no interest in interpreting those “topics” anyway (they are not used in any analysis), so that argument is irrelevant. In fact, one might even imagine doing away with hidden activations altogether, and just giving ChatGPT a lecture text and a question, asking it to identify which part of the lecture is referred to by a question, and using the number of utterances between this part of the lecture and any other part to conduct similar analyses to those proposed by the authors (e.g., constructing knowledge maps). If the goal is to propose scalable tools for wide adoption, it appears that testing such models to see whether they could actually work is an important first step before proposing methods like those advanced by the authors.

We appreciate the reviewer’s suggestions here. There are a few points we want to unpack.

First, regarding our choice of model, one benefit of using simple topic models (LDA) over more sophisticated models like GPT-2 is that training is actually relatively inexpensive (with respect to time, compute power, and memory). For example, on the consumer-grade laptop we used to write and perform our analyses, training the model and transforming our complete document corpus takes 803 ms (+/- 5.23 ms; M +/- SD over 1,000 iterations). Therefore re-training our topic models on new domains is “scalable” in a way that would not be feasible for more sophisticated models or much larger corpora. Topic models can even be fit or updated to accommodate streaming data, often in near-real time (e.g., Hoffman, Bach, and Blei, 2010, NeurIPS).

That said, we’ve certainly considered using LLMs like GPT-2, ChatGPT, etc. In fact, in our other ongoing work (e.g., <https://github.com/ContextLab/chatify>) we’ve been developing LLM-based “tutors” to help students learn course materials. We agree that incorporating LLMs into education technologies to help teachers and students presents an exciting opportunity!

In our current study, our goal is to *map and track* what our participants know, and how they learn with training. This requires maintaining a formal representation of what material is being learned, what is known, and so on. For example, suppose one were to ask ChatGPT to match up each question with some part of a lecture. What would one “do” with that response? Ultimately, in our view, we would need to turn those “match labels” into numbers of some sort. And that would seem to be best accomplished by looking directly at the text embeddings (e.g., internal to ChatGPT), since ChatGPT’s outputs can be unreliable, heavily dependent on the specific prompt used, and so on. Ignoring ChatGPT’s reliability, though, ChatGPT also has no built-in mechanism for keeping track of what the student knows, or how that knowledge might relate to the content of a course the student is learning from. Building in those mechanisms would appear to require something like what we built in our current paper.

Another aspect of the reviewer’s comment touches on which model might be “best” for constructing or learning text embeddings of lecture content. This idea relates to a comment also raised by Reviewer 1 in their comment 2b, about a “tension” between the potential generality of our approach versus the targeted nature of our particular embedding space (trained to the content of these two lectures). Although LLMs like BERT, GPT-{2, 3, 4}, ChatGPT, LLaMA, and so on are incredibly flexible,

generalizable, and powerful, they are surprisingly *bad* at distinguishing between subtle conceptual differences at the scale of parts of a single lecture video. As a quick example (and reproducing some of our response to Reviewer 1’s comment 2b for convenience), we used BERT (a transformer-based LLM trained on a large corpus, with 768 feature dimensions) to carry out some of the core analyses from our main paper (i.e., swapping out BERT for the topic modeling approach we reported on):

Quoting from our previous response:

We highlight three general differences in the above figure. In the top row (“Trajectories”) we show a 3D projection (using PCA) of the content trajectories for each lecture (blue and green lines) and each question (blue, green, and purple dots). The LDA embeddings exhibit several desirable properties compared with the BERT embeddings:

The LDA embeddings of the lectures and questions are “near” each other– e.g., the convex hull enclosing the two lectures’ trajectories is highly overlapping with the convex hull enclosing the questions’ embeddings. The BERT embeddings of the lectures and questions are instead largely distinct.

The LDA embeddings of the questions for each lecture and the corresponding lecture’s trajectory are similar. E.g., the LDA embeddings for lecture 1 questions (blue dots) tend to appear closer to the lecture 1 trajectory (blue line), and the LDA embeddings for lecture 2 questions (green dots) tend to appear closer to the lecture 2 trajectory (green line). The BERT embeddings do not show this property.

In the middle row (“Per-question matches”), we display the correlations between each question’s embedding coordinate and the embedding of each moment of the corresponding lecture. Again, the LDA embeddings show several desirable properties compared with the BERT embeddings:

The time series plot of individual questions’ correlations are different from each other when computed using LDA (e.g., the traces can be clearly visually separated), whereas the correlations computed from BERT embeddings of different questions all look very similar. This tells us that LDA is capturing some differences in content between the questions, whereas BERT is not.

The time series plots of individual questions’ correlations have clear “peaks” when computed using LDA, but not when computed using BERT. This tells us that LDA is capturing a “match” between the content of each question and a relatively well-defined time window of the corresponding lectures. The BERT embeddings appear to blur together the content of the questions versus specific moments of each lecture.

In the bottom row (“Cross-lecture correlation matrices”) we compare the pairwise correlations between embeddings of lectures and questions within versus across content areas (i.e., content covered by the individual lectures, lecture-specific questions, and by the “general physics knowledge” questions). The LDA embeddings show a strong contrast between same-content embeddings versus across-content embeddings. In other words, the embeddings of questions about the lecture 1 (“Four fundamental forces”) material are highly correlated with the embeddings of lecture 1, but not with the embeddings of lecture 2, questions about lecture 2, or general physics knowledge questions. We see a similar pattern with the LDA embeddings of the lecture 2 (“Birth of stars”) questions. In contrast, the BERT embeddings are all highly correlated with each other.

Taken together, these comparisons illustrate how LDA (trained on the specific content in question) provides both coverage of the requisite material and specificity at the level of the content covered by individual questions. BERT, on the other hand, essentially assigns both lectures and all of the questions (which are all broadly about “physics”) into a tiny region of

its embedding space, thereby blurring out meaningful distinctions between different specific concepts covered by the lectures and questions.

We note that these are not criticisms of BERT (or other large language models trained on large and diverse corpora). Rather, our point is that simple fine-tuned models trained on a relatively small but specialized corpus can outperform much more complicated models trained on much larger corpora, when we are specifically interested in capturing subtle conceptual differences at the level of a single course lecture or question. Of course if our goal had been to find a model that generalized to simultaneously capturing many different content areas, we would expect our approach to perform comparatively poorly relative to BERT or other much larger models. We suggest that bridging the tradeoff between high resolution within each content area versus the ability to generalize across many different content areas using a single set of model weights will be an important challenge for future work in this domain.

We have added a discussion of these issues to pages 23–25 of our revised manuscript, and have also added the figure above to our supplementary materials (Supp. Fig. 6).

1.2. To follow up on the issue of technology: ways to evaluate the specific knowledge of students already exist. For instance, some instructors give specific feedback on specific questions in online quizzes; I personally weigh different questions based on whether they require knowing things that were explicitly stated in class vs. near-transfer to new examples vs. far-transfer that also requires combination of knowledge across different lectures; and, more generally, there is a vast literature spanning 2-3 decades on the importance of specific feedback to students. The question is why these methods are not more wildly adopted. If the reason is related to scalability / time / effort, it is currently unclear whether the authors' method is better suited for wide adoption compared to using something like ChatGPT (I am by no means a fan of ChatGPT, but the fact remains that it is overall surprisingly good at tasks that are not dissimilar to what the authors are trying to do).

Personalized human instruction is hard to beat; we're not aware of any modern approaches, including ChatGPT, that approach the level of an experienced and expert human instructor. The reviewer's examples of providing personalized feedback, manually tagging (in advance) which questions are associated with which content areas, considering a variety of questions that test understanding at different levels, and so on are all excellent ways to effectively teach. One could imagine that an experienced one-on-one tutor could employ all of these techniques, perhaps combined with creative ways of teaching the material, to serve as a "gold standard" of sorts, with respect to what one might hope to achieve with an automated tutoring approach if all of the appropriated pieces were in place.

What might be required to build such a system? One consideration is that effective tutors might hold in mind a representation of what their student *already* knows, what they are ready to learn, what the learning objectives are, what previous concepts they learned quickly versus struggled with, and so on. For a given content area or skill (e.g., "computing derivatives of trigonometric functions") there will be a set of relevant concepts the student will need to grasp in order to have mastered the material. Our current framework for mapping out participants' knowledge (and how their knowledge changes over

time with training) serves as a “prototype” of what an automated tutor might incorporate as its internal representation of “what the student knows” at a given moment. A desirable aspect of our approach is that our mapping procedure can generalize to any content area (i.e., there is nothing inherent to the method that makes it better suited to the particular content we happened to select; we could have just as easily chosen courses in art history instead of physics, and one could reasonably expect similar results and performance). Of course we have not explicitly tested every possible content area, but we see our work as a “first step” in the direction of a generalized mapping approach.

Another consideration is how the learner’s knowledge might be estimated. If one has the benefit of a human tutor, the tutor can (as the reviewer suggests) manually “match up” each question with core concepts that their student might need to know. But *without* a human tutor, could this matching process be carried out automatically? Again, we show a proof of concept demonstration that we can *automatically* match up specific questions with specific moments in existing lectures (Fig. 4, Supp. Tab. 3). Further, we provide a formal approach to estimating knowledge using the learner’s responses to a relatively small number of questions. And by showing that we can accurately predict knowledge for held-out questions (Fig. 6), we show that our approach can generalize to concepts outside of the immediate training set used to construct the knowledge estimates. This is a critical requirement for building scalable tutoring systems that can work across a variety of content areas.

As we noted above, while we (like the reviewer) see some limitations of tools like ChatGPT, we also agree with the reviewer that ChatGPT can serve as a useful component of automated tutoring systems. But ChatGPT has no internal machinery for representing or tracking the learner’s knowledge, nor does it maintain a “theory of mind” of the learner, nor does it (in and of itself, to the best of our own understanding) have any deep understanding of the material itself. So neither ChatGPT nor other related tools or systems we are aware of can solve the problem of mapping out or tracking knowledge in the scalable (i.e., automated) and generalizable way we present in our paper.

To address this issue, we have added a comment to the discussion section comparing our approach to other “alternatives” like ChatGPT, and clarifying what we see as some benefits and differences of our approach (page 25).

1.3. The authors might also be interested in providing a computational model that tells us something about human psychology (e.g., learning processes). In this case, the representational assumptions that the model makes require some discussion. On the one hand, it would be important to compare the model to “baseline” or “control” models in order to claim that the current model is useful (compared to, e.g., averaging word2vec embeddings of words in each lecture sentence or question and using representations; this might be what the authors mean by “superficial matching”?); on the other hand, it would be important to discuss claims that embeddings models are a non-starter for complex knowledge representation because they do not have any explicit notion of causal schemas like “knowledge graphs” / “concept maps” / “theory theory”, etc. In a sense, the authors’ model might occupy an “uncanny valley”, being too complicated compared to simpler models that can achieve the same empirical results without claiming any psychological reality, but not structured enough to be a quantitative theory of human learning.

The reviewer brings up several points here. We'll address each in turn.

First, the reviewer proposes comparing our approach (using topic models) to some alternatives. In our response to this and Reviewer 1's comment 2c, we compare our topic modeling approach to a simpler "word matching" approach (i.e., what we meant by "superficial matching") within the context of "matching up" individual questions with specific moments of each lecture. Our topic models tend to identify well-defined time intervals in the lecture that match up well conceptually with the questions (e.g., Fig. 4 shows some example time courses and question/lecture matches, and we have added Supp. Figs. 3 and 4 and Supp. Tab. 3 to show these time courses and text matches for all of the questions). On the other hand, simpler word matching does *not* behave nearly as well. For convenience we have pasted in the figure we included in response to Reviewer 1's comment; the figure shows some example time courses for questions, as computed using topic models (left) and word matching (right):

As we also note above (in response to Reviewer 2's similar comment):

[T]his is because simple word matching (right column) identifies moments when any of the individual words in the question are used during the lecture, whereas topic vector correlation identifies moments when the underlying themes (most probably) expressed by the question's specific combination of words were also expressed in the lecture. For example, in the first question (top row), mentions of "following," "difference," "gravity," "electromagnetic," and "force" at other moments in the lecture also evoke increases in Jaccard similarity.

At the other end of the “methods spectrum,” one could imagine replacing our topic modeling approach with “fancier” (and orders of magnitude larger, both with respect to number of parameters and training corpus size) transformer-based models like BERT, GPT-{2, 3, 4}, LLaMa, etc. This doesn’t work particularly well either, as we report in response to the reviewer’s comment 1.1, above.

In this sense, we would argue that our approach actually occupies the *opposite* of an “uncanny valley” —rather, we think our approach occupies a sort of “sweet spot” that enables us to capture the relevant content at the appropriate semantic scale. Our approach enables us to accurately and consistently identify each question’s content in a way that also matches up with what is presented in the lectures. In turn, this enables us to construct accurate predictions about participants’ knowledge of the conceptual content tested by held-out questions (Fig. 6).

We agree with the reviewer that topic models contain no explicit internal representations of more complex aspects of “knowledge” like knowledge graphs, dependencies or associations between concepts, and so on. We have added a note (page 26) to clarify that we are not claiming that our approach incorporates these sorts of representations, but that such features might be added as extensions to our approach to more accurately and precisely capture, characterize, and track knowledge.

1.4. The authors might be interested in providing a quantitative measure of how knowledge “behaves”. In this case, I think the authors’ claims are over stated, especially with regard to “high-resolution insights”, as I discuss in (2) below.

We have added several new tests and analyses to bolster our “high-resolution insights” claim. We describe several of them below in response to the reviewer’s specific comments. We also ran an analysis specifically related to how knowledge “behaves” as one moves through the text embedding space. We described it above in Reviewer 2’s comment 2, but we have copied the text and figure below for convenience:

In addition to the analysis suggested by the reviewer, their comment led us to consider the broader question of “how far in the text embedding space knowledge estimates extend.” For example, suppose we know that a participant answers a question (at embedding coordinate X) correctly. As we move away from X in the embedding space, how does knowledge (as estimated by quiz performance) “fall off” with distance? Or, suppose the participant instead answered that same question incorrectly. Again, as we move away from X in the embedding space, how does what the participant doesn’t know about the content change with distance? We reasoned that, assuming our space is capturing something about how participants actually organize their knowledge, conceptual knowledge right around X should be similar to the participant’s knowledge of the content at X. And at another extreme, at some distance (after moving sufficiently far away from X), our guesses about what participants know (based on their response to the question at location X) should be no better than guessing based on their overall proportion of correctly answered questions—i.e., if Y is very far away from X, all we

can do with the participant’s response to X is guess that “their performance on quiz questions about Y is about equal to their average performance on quiz questions about any material.” With these ideas in mind, we asked: conditioned on answering a question correctly, what proportion of all questions (within some radius r of that question’s embedding coordinate) were answered correctly? We could then plot the proportion as a function of r . Similarly, we could ask, conditioned on answering a question incorrectly, how the proportion of correct responses changed with r . We found that quiz performance falls off smoothly with distance, and the “rate” of the falloff doesn’t appear to change across the different quizzes, as measured by the distance at which performance becomes statistically indistinguishable from a simple proportion-correct score (Fig. 7):

2. The authors claim that they provide “nuanced insights into what learners know and how their knowledge changes with training”. I do not believe this is the case.

We have attempted to clarify this claim in our revised manuscript. The “nuanced insights” phrasing is meant to refer to predictions and maps at the conceptual level of individual moments of the lectures (Fig. 5) and individual quiz questions (Fig. 6). Our “changes with training” phrasing simply refers to our updated estimates using each set of quizzes in turn.

We also created detailed knowledge and learning maps before/after watching each lecture (visualized in Fig. 8). This gives us another high-resolution view of participants’ knowledge, and those maps change over time as the participants learn. We also test the accuracy of the maps by asking whether

the knowledge predicted at the embedding coordinates of held-out questions tracks with participants' performance on those questions (Fig. 6), and we characterize how "knowledge" (as estimated using quiz performance) changes with distance in the embedding space (Fig. 7).

2.1. *Nearly all analyses yield the same type of knowledge that would be obtained from calculating "proportion correct" separately for each subset of questions (Birth of Stars, Four Fundamental Forces, general physics). Namely, students do not know any topics before a lecture; after lecture #1, they learn the content of that lecture but not unrelated content from lecture #2 which they haven't yet watched; and after viewing lecture #2, they also learn its content. Despite the authors' claims for higher-resolution quantification of how knowledge evolves throughout a lecture, their analyses in Figures 3, 5, 6, and 7 only provide evidence for this general pattern. For instance, we can compute % correct on 4 questions about Birth of Stars to predict the correctness of the 5th question; would this analysis be significantly inferior to the one proposed by the authors? Overall, my impression is that the authors do not quantitatively evaluate any pattern that is more nuanced than "knowledge of lecture #1 and knowledge of lecture #2 can be separated".*

Our approach and results yield a number of insights into participants' knowledge and learning that simpler "proportion correct" measures would not (directly) support. First, in panels A and C of Figure 5, we show estimates of participants' knowledge about each moment of each of the lectures, which we obtain by leveraging our ability to map quiz questions they answered correctly and incorrectly onto specific relevant time periods of the lectures' contents. While participants do indeed tend to know more about each lecture's content overall after viewing it than they did before (as one might expect), these temporally specific ("high-resolution") knowledge estimates provide additional insights into which *parts* of each lecture participants know comparatively more or less about, as well as which parts they learn more or less successfully through viewing it. The analyses shown in Figure 6 show that these insights generalize to the level of individual quiz questions. The analyses in Figures 7 and 8 show how quiz question performance (our proxy for "knowledge") changes across locations in the text embedding space (i.e., across different "concepts"). Based on the reviewer's comment regarding our approach quantifying "*how knowledge evolves throughout a lecture*", as well as their comment 2.3 below and a related comment by reviewer 2 (comment 4), we realize that these plots can be misinterpreted as displaying how much participants know *overall* at each moment of the lectures, rather than characterizing knowledge *about the content* at each timepoint (of each lecture). We have amended the caption for Figure 5 to clarify what we are trying to show in panels A and C.

As shown in Figure 7 (also included in our response to the reviewer's comment 1.4), we show in our revised manuscript that our predictions are substantially more precise ("high-resolution") than simple proportion-correct measures, provided that we are attempting to estimate knowledge within a correlation distance of (roughly) 1 from the embedding coordinates of a question they answered. Beyond a correlation distance of ~ 1 , our predictions approach what could be obtained through simple proportion-correct measures (Fig. 7B).

We also specifically tested the reviewer's concern about whether the knowledge patterns we identify are more nuanced than the level of different lectures (this has been incorporated into Figure 6 in our

revised manuscript). In an excerpt from this figure below, we show knowledge predictions obtained using only the other questions about the same lecture’s material (top row: lecture 1; bottom row: lecture 2; left column: Quiz 1; middle column: Quiz 2; right column: Quiz 3):

We show that we can (often, though not always) accurately predict participants’ knowledge of a held-out question, using only the other four questions about the same lecture, from the same quiz. We expected these predictions to be quite noisy, considering each knowledge estimate is derived from just four other responses to multiple questions, but we nonetheless show that knowledge at the coordinates of held-out questions participants answered correctly is often predicted to be higher than at the coordinates of held-out questions participants answered incorrectly.

2.2. The authors claim: “we show that our approach can automatically match the conceptual knowledge probed by individual quiz questions to the corresponding moments in lecture videos when those concepts were presented”. This is inaccurate. First, the authors show that their method can match quiz questions to the overall lecture they correspond to (Figure 3B), but this is only a sanity check (as the authors acknowledged), not a technological advancement – instructors who write questions can tag them for which content they are testing, and we hardly need a computational tool to achieve this goal; in fact, this appears to be what instructors mentally do when trying to come up with questions to test different areas of content, and what the authors of this manuscript themselves did when creating questions for lecture #1 vs. #2. Second, the authors report qualitative impressions that each question can be further matched to a particular timeframe within a lecture (Figure 4), but I do not believe they have provided any test of this impression (my apologies if I missed it!). For instance, if instructors are given the questions and are asked to match them to specific timeframes within the lecture – do they provide the same estimates? (Or, if they are

given 3 timeframes identified by the model, can they tell which one corresponds to a given question?). If a student missed a question and is referred back to a particular timeframe of the lecture to re-watch, do they do better than a student referred to a different part of the lecture? Here, too, I wonder: how useful is the “high resolution” match between question and content, beyond what instructors do anyway when compose questions? (e.g., I know exactly which slides are pertinent to each question on my exams, and it’s possible ChatGPT would be able to match questions to the relevant part of a lecture).

We’d like to push back on the reviewer’s claim that we are matching up questions and lectures only at the level of the entire lecture, as opposed to what we describe as “corresponding moments” in individual lectures. In Figure 4, for example, we show that each question’s embedding displays a time course of correlations with the moment-by-moment embeddings of the lecture that tends to “peak” at a relatively well-defined time interval within the lecture. Further, when we manually examine the question text versus the lecture transcripts at those best-matching moments, they nearly always “align” with respect to which concepts the question is asking about versus what the lecturer is discussing at those moments. This alignment can be assessed by manually comparing the lectures’ and questions’ text. In addition to the specific examples we provide in Figure 4, we have added two new supplemental figures showing these “peaks” for each question (which we detect automatically; Supp. Figs. 3 and 4), as well as a new supplemental table showing the question and lecture text for *all* lecture 1 and 2 questions; Supp. Tab. 3).

We also wish to clarify that while we agree that a human instructor could manually match up quiz questions with moments in the lectures, the goal in our manuscript is to do this matching *without* human-generated tags. As we mention in response to the reviewer’s comment 1.2 above, we see this move from manually generated to automatically generated tags as a necessary part of scaling up automated teaching tools beyond what could be accomplished by human instructors alone.

Finally, we appreciate the reviewer’s suggestion that ChatGPT might be able to match up questions with the relevant parts of each lecture. The challenge (as we also described in response to the reviewer’s comment 1.2) is that it’s not clear what one would then “do” with those ChatGPT-derived labels. Constructing explicit knowledge estimates using our framework requires knowing the specific embedding coordinates of each question. That also lets us say, not only that a given question is mostly “about” the content covered by some part of a lecture, but it also lets us quantitatively compare the similarity of content covered by any (arbitrary) pair of questions. In turn, we use those similarity values to construct weighted averages that form our knowledge estimates.

2.3. The authors claim: “we demonstrate how we can estimate moment-by-moment “knowledge traces” that reflect the degree of knowledge participants have about each video’s time-varying content, and capture temporally specific increases in knowledge after viewing each lecture”: this claim has the same problems detailed in 2.2; namely, all quantitative analyses in the paper focus on differences in knowledge between pre- to post-lecture, not on demonstrating that the time-varying trace itself captures anything meaningful.

We show (e.g., in Fig. 6) that we can predict knowledge about the content captured by individual questions (on individual quizzes), which in turn localize to specific well-defined windows in the

lectures (Fig. 4). Our pre- versus post-lecture tests (e.g., Fig. 5B, 5D) are *not* intended to be temporally specific, since those tests average across all timepoints in the relevant lecture(s). As Reviewer 2 pointed out (comment 4), and as mentioned in our response to comment 2.1, our previous description of what is shown in Figures 5A and 5C required some clarification, which we have made by editing the Figure 5 caption. In those “traces” we are characterizing knowledge about the *content* at each timepoint (of each lecture), not “what or how much is known at each moment of training.”

2.4. *The authors claim: “these knowledge estimates can generalize to held-out questions”: there is not test demonstrating that this analysis (Figure 6) does anything more than capturing low-resolution differences between entire topics. Intriguingly, the results of Quiz 3 suggests that this might be the case: contra to the authors’ claim, Quiz 2 is not the most sensitive test, because passing it merely requires distinguishing between questions about lecture #1 from questions about lecture #2, which is a very low-resolution test (students haven’t learned content that they haven’t been exposed to); Quiz 3 is the one that requires the highest sensitivity, assuming that students know some parts of each lecture better than other parts of that same lecture, and that there are individual differences in which parts are more vs. less understood. Such a case would allow to test the authors’ claim that knowing which questions in a given lecture a student has answered correctly (rather than an overall estimate of a student’s score on that lecture) can predict which other questions from that lecture they will answer correctly. If the assumption of variations in knowledge throughout a lecture and across students does not hold and, instead, there is a “ceiling effect” as the authors claim (all students know all sections of both lectures very well), then the design of the study is inappropriate for testing high-resolution knowledge.*

Reviewer 2 had a somewhat different take on this idea (comment 2). They argue that it is specifically *across-lecture* predictions that show that we are capturing deep conceptual meaning, as opposed to something coarser. They suggested using lecture 1 questions to predict knowledge for held-out lecture 2 questions (and vice versa) from the same quiz. We show in our revised Figure 6 that estimated knowledge for held-out other-lecture questions that participants answered correctly is higher than for held-out other-lecture questions that they answered incorrectly. This holds for Quizzes 2 and 3 (as do our predictions using *all* questions from each quiz to predict a single held-out question).

We also ran the “within-lecture” variant of this analysis that the reviewer is proposing here, in response to the reviewer’s previous comment (2.1); we pasted in the relevant results in response to that comment (above; also see Fig. 6, “within-lecture” plots). In summary, after participants watch lecture 1 (i.e., on Quiz 2), we are able to reliably predict knowledge for held-out lecture 1 questions using only other lecture 1 questions from the same quiz. Similarly, after participants watch lecture 2 (i.e., on Quiz 3), we are able to reliably predict knowledge for held-out lecture 2 questions using only other lecture 2 questions from the same quiz. (We also ran other variants of these tests; the full set of tests is reported in Fig. 6.)

2.5. *The authors claim: “visual maps that provide snapshot estimates of how much participants know about any concept within the scope of our text embedding model, and how much their knowledge of those concepts changes*

with training”: again, their analyses seem to capture differences between two distinct lectures, not across moments / topics within a lecture.

Our new “knowledge falloff” analysis (Fig. 7; also pasted into our response for the reviewer’s comment 1.4) shows that the accuracy of our knowledge estimates falls off gradually and smoothly with increasing distance in the text embedding space. This goes beyond simply distinguishing between questions from one lecture versus the other.

As we mention in our response to the reviewer’s comment 2.4, our new “within-lecture” knowledge prediction analysis (Fig. 6, “within-lecture” panels) also shows that we can capture knowledge at the sub-lecture resolution (i.e., knowledge about a single held-out question, given other questions from the same lecture and from the same quiz).

2.6. The authors claim: “our work suggests a rich new line of questions about the geometric “form” of knowledge”: again, the low-resolution nature of the authors’ results seem quite far from studying the geometric form of knowledge, compared to existing lines of work (both recent ones, and ones that date back 20 years). For instance, some studies have focused on the intrinsic dimensionality of content manifolds (e.g., “Intrinsic Dimension Estimation for Robust Detection of AI-Generated Texts”), or on the linear geometry of word embeddings and how it reflects structured world knowledge (e.g., “How do blind people know that blue is cold? Distributional semantics encode color-adjective associations”). Other models have explored how the geometry of knowledge evolves over time (e.g., Jay McClelland’s early work on connectionist models of semantic knowledge; or papers like “Structured Semantic Knowledge Can Emerge Automatically from Predicting Word Sequences in Child-Directed Speech”).

By “geometric,” we are referring to geometric (e.g., shape or distance-based) relations between different concepts. Our embedding spaces define geometries that describe how concepts relate: difference in “meaning” is reflected by Euclidean distance in those spaces. In general, that we can explain participants’ behaviors using these representations suggests that the text embedding spaces we use to “map out” knowledge are capturing at least some aspects of how knowledge varies across different content for our participants. Our new “knowledge falloff” analysis (Fig. 7; also pasted into our response for the reviewer’s comment 1.4) also shows how, given that you know “something” (e.g., at the coordinate of some question) that knowledge (or lack thereof) changes as a function of distance in the embedding space.

We appreciate the reviewer’s pointers to other related work, and we have added in citations of these papers to our revised manuscript.

2.7. One way to reframe what the authors are doing is to treat the two lectures as smaller sections within a long lecture in an academic course (e.g., a 90 minute class). In this case, distinguishing between the two lectures is akin to breaking an entire lecture into smaller (“higher-resolution”) parts. However, I do not believe this is how the authors conceive of their work, because if that was the case, they would not have visualizations like those in Figure 4 or 5A, which are about second-to-second changes within a 10-minute lecture.

First, if we're interpreting the reviewer's comment as intended, our understanding is that the reviewer is following up on their prior suggestions that we are distinguishing concepts only at the level of entire "lectures" as opposed to at the sub-lecture resolution (e.g., smaller concepts mentioned within a given lecture video). In our responses above (and with the new analyses we've added), we've attempted to clarify our approach and findings. For example, in Figure 6 we show that we can estimate "knowledge" (as reflected by quiz performance) at the level of individual questions, which in turn tend to map onto relatively well-defined intervals within the lectures (Fig. 4, Supp. Tab. 3).

Regarding the reviewer's comment about how we conceive of our work, we can also unpack how we are thinking about "concepts" and "lectures." As we note on page 3, we treat concepts as equivalent to embedding coordinates; we see them as defined implicitly by the embedding model's geometry. Within the individual lectures, concepts build on each other: if one were to chop up a single lecture into "conceptual chunks" and watch the different segments "out of order," the material would be very difficult to follow. In contrast, across the lectures we showed participants in our study, there are no dependencies (as noted on page 6 of our manuscript, we intentionally chose introductory videos from two different lecture series). In other words, watching lecture 1 before lecture 2 is no more difficult to follow than watching lecture 2 before lecture 1.

Regarding Figures 4 and 5A, our general goal is to show that the conceptual content of a single lecture is not a uniform thing, but rather it varies from moment to moment as different material is discussed or presented. This would be true whether we consider what we call "lectures 1 and 2" to be parts of a longer "unified" lecture, or whether we (as in our current framing) consider lectures 1 and 2 to be distinct "units" for the purposes of our analysis.

3. Some minor comments are below. Some of these are suggestions, whereas others are thoughts that I jotted down as I was reading the manuscript and would perhaps be interesting or useful to the authors, but are equally likely to be irrelevant musings.

We appreciate the author's comments and suggestions, and for taking the time to write these thoughts out!

3.1. Introduction, line 60: "These models consider not only the co-occurrences of those elements within and across documents, but also patterns in how those elements appear across different scales (e.g., sentences, paragraphs, chapters, etc.), the temporal and grammatical properties of the elements, and other high-level characteristics of how they are used". I think this claim could benefit from some clarifications. For instance, doesn't LDA treat documents as "bags of words", looking only at co-occurrences? (The researcher can cut down a document, as is done here, but this is not a feature of the text embedding model – the model treats whatever "unit" of text it receives as a bag). In static word embeddings, co-occurrences are limited to local ones (e.g., a context window of 10 words), so there is not notion of paragraph / chapter, so there is more or less a single temporal scale being modeled. And in contemporary LLMs, some of the more "complex" features, such as grammatical features, are implicitly inferred from co-occurrences (there is no separate mechanism for learning these properties; a simplified form of them even arises in GloVe or word2vec).

Great point. We have added some text to the discussion to clarify that our framework is based around a “bag of words” model, and we also provide some additional rationale for why we chose to use LDA, e.g., as opposed to more “modern” LLMs (pages 23–25, Supp. Fig. 6).

3.2. Line 63: *“A model that succeeds at capturing an analogue of “understanding” is able to assign nearby feature vectors to two conceptually related documents, even when the specific words contained in those documents have very little overlap.” It seems to me that this task is very easy for LLMs (they can learn which words mean similar things based on distributional patterns). The harder thing is to separate documents with very similar words that talk about conceptually very different things (e.g., distinguishing between statements about A causing B vs. A correlating with B due to a shared cause vs. B causing A...).*

We agree, and we have added a note to the discussion to this effect (page 26).

3.3. Line 72: *“For example, understanding the concept of a fish swimming in water first requires understanding what fish and water are.” From the point of view of conceptual role semantics (or “theory theory”), understanding the relationship between fish and water is precisely what is important for learning what those terms are (rather than some “dictionary definition” of their inherent properties). See, for instance, <https://arxiv.org/abs/2208.02957>.*

We appreciate the pointer to this paper; we’ve added a citation along with a note to this effect (page 3).

3.4. Figure 5A: *the traces look almost like mirror images of one another. What is the reason? Is this caused by time-points which are vs. are not covered by questions?*

Sort of! First, we have a similar intuition to the reviewer that the shapes of those traces reflect how the questions “cover” the different moments of the lecture (e.g., Fig. 4). Second, the traces in Figure 5A aren’t quite mirror images. This is actually a sort of visual artifact resulting from how participants’ performance on certain questions (that cover particular intervals of *Four Fundamental Forces*) changes before vs. after watching that lecture.

The main intervals in the lecture that give rise to the “mirror image” appearance in Figure 5A are (roughly) between 3–4 minutes and 9–10 minutes. In our new Supplementary Figure 3, we show time-varying correlation plots for each individual question. In that figure, we can see that Questions 2, 3, 13, and 8 are the most correlated with those intervals in the lecture (and therefore carry the most influence over the time-varying knowledge estimates at those intervals). It happens that participants tended to get those questions wrong more often (relative to the other questions) before watching the *Four Fundamental Forces* lecture, but they tended to get those questions *right* more often (again, relative to the other questions) *after* watching that lecture. So the shapes of those traces actually reflect participants’ behaviors (quiz performance), which seems to differ in meaningful ways for questions that cover different content areas.

3.5. Line 327: *could the authors please say more about how this localization is non-trivial? What is shown is that the topic model can identify which broad topic is addressed by each question (i.e., not which part of a lecture, but merely which lecture). Therefore, is this finding not fully predicted by Figure 1C? In what way does it provide new*

information beyond Figure 1C? Moreover, if the authors simply constructed a Voronoi diagram where each point in space is assigned to its nearest question, would the result not be highly similar? (I am not saying that it is, but this is a much simpler method and, in a sense, a discretization of the authors' formula).

The “non-trivial” localization we’re referring to is that even though our knowledge predictions at each coordinate on the knowledge maps are based solely on the locations of the *questions*, the increases in predicted knowledge we see from one quiz’s map to the next (shown as changes in the maps’ shading at each coordinate) are specific to the regions near the just-watched *lecture’s* trajectory. In other words, changes in the predictions we make about participants’ knowledge based on this “map’s” coordinate system track how we might intuitively expect them to change following exposure to particular new content, despite not considering that content’s spatial location in computing them. This is related to the visualization in Figure 2C (we assume that the reviewer is referring to Fig. 2C, since Fig. 1 has no panel C), but the two figures are different in several important ways.

Figure 2C shows that the sets of questions we manually designed (and labeled) to be “about” each lecture tend to appear near their corresponding lecture when visualizing their first three principal components. This requires generally that the topic model capture similarities between content we expect to be conceptually related. The knowledge maps, by contrast, show more specifically that questions for which participants’ abilities to answer correctly *changed* after viewing a particular lecture tend to be embedded nearby that lecture. This requires that both the topic model *and* the non-linear manifold learning algorithm we used to construct the knowledge maps’ coordinate system (UMAP) preserve relationships that are meaningful to what individuals know and how they learn. It also requires that changes in knowledge are both sufficiently consistent and sufficiently specific across participants to produce the visual contrast we see in Figure 8.

We do agree that many of these patterns themselves are not necessarily specific to these maps, and can be observed elsewhere in our findings. For example, Figures 2 and 3 suggest that questions can be matched to their corresponding lectures, Figure 4 suggests that different questions will map onto different locations along those lectures’ trajectories, and Figure 5 suggests that increases in knowledge from one quiz to the next will be lecture-specific and consistent across participants. However, our ability to automatically capture these patterns simultaneously in low-dimensional, intuitive, human-readable “maps” (e.g., which have been transformed to reflect Euclidean distance, and from which we can recover word clouds like those shown in Fig. 8C) is a non-trivial advance in and of itself.

The reviewer’s idea of “filling in” these maps using Voronoi diagrams is an interesting one, and would certainly lead to neat visualizations! However, based on other results we report in our manuscript, we suspect this would be a less truthful representation of participants’ (probable) knowledge at each individual coordinate on these maps. As described in our response to Reviewer 2’s comment 2, we show in Figure 7 that knowledge tends to “fall off” gradually with respect to distance in our text embedding space, and in Figure 6 we show that participants’ knowledge about a particular embedding-space coordinate can be predicted by a weighted combination of multiple coordinates for which their knowledge is “known.” By contrast, a Voronoi diagram such as the reviewer describes

would cast success on each individual quiz question as the sole, static determinant of our knowledge predictions for all knowledge-map coordinates to which it is the single closest, while having zero influence on our predictions beyond that point. As such, we see our current approach (of computing accuracy-weighted sums of radial basis functions centered on each quiz question) as more directly motivated by our other findings (as well as more theoretically consistent with our method of estimating knowledge in the high-dimensional topic space) and have elected to retain it rather than opting for a more discrete alternative.

3.6. *Choosing number of topics: did the authors examine the perplexity or coherence of the topics? These measures (especially coherence) appear to be accepted metrics for choosing the number of topics (see, e.g., <https://aclanthology.org/D11-1024.Pdf>, <https://aclanthology.org/N10-1012.pdf>).*

Reviewer 1 (comment 6) asked about how we selected the number of topics as well. Here is the relevant part of our response to that comment:

To select an appropriate number of topics (k) for our model, as a starting point, we identified the minimum number of topics that yielded at least one “unused” topic (i.e., in which all words in the vocabulary were assigned uniform weights) after training. This suggested that the number of topics might be sufficient to capture the set of latent themes present in the two lectures. We found this value to be $k = 15$ topics. We found that with a limited number of additional adjustments following Boyd-Graber et al. (2014; e.g., removing corpus-specific stop-words), the model yielded (subjectively) sensible and coherent topics.

To elaborate, we tend to take the view proposed by Boyd-Graber et al. (2014) in their “Care and feeding of topic models” paper. Essentially, they suggest that using automated (optimization-based) approaches to selecting the number of topics can lead to poor performance. (This view is also reported in myriad other papers, many of which are cited in, or cite, the Boyd-Graber et al. paper.) Instead, they suggest that in practice, it tends to work well to manually evaluate whether a given number of topics is “appropriate” using three criteria:

1. Are individual topics meaningful, interpretable, coherent, and useful?
2. Are assignments of topics to documents meaningful, appropriate, and useful?
3. Do topics facilitate better or more efficient document search, navigation, understanding, [or] browsing?

Although these criteria do not perfectly align with our current goals and approach, we felt that the topics seemed well-behaved and useful (e.g., see Tab. S2 and Figs. 3, S1, and S2). Out of curiosity, and in response to the reviewer’s comment, we also carried out a number of analyses using coherence to see how many topics were “optimal” by those measures. There are several ways to “compute” coherence and select a “best” number of topics, so we implemented seven different coherence measures proposed by several studies (Mimno et al., 2011; Cao et al., 2009; 4 measures from Röder et al., 2015; and Arun et al., 2010). For each measure, coherence is typically computed using some number n of the most heavily weighted on by each topic . We used $n = 5, 10, 15, 20, \dots, 50$, yielding a total of 70

approaches to selecting the optimal number of topics (k). Then we counted up the numbers of times each value of k was selected as the optimum:

We found that overall, according to these metrics, using $K = 17$ topics was most frequently selected as the optimal choice. The topic distributions using 17 topics also looked reasonable, although when we compared them manually to the 15 topic versions we felt they looked slightly less clean. (We did verify that our main results replicate using $K = 17$ topics). Ultimately we decided to stick with our prior “hand selected” approach.

Reviewer #1 (Remarks to the Author):

The authors have adequately addressed my comments. I appreciate the very thorough response and I endorse the manuscript for acceptance. The comparisons to BERT and word-level Jaccard similarity, and the accompanying explanations, were also very helpful for understanding what insights their approach provides.

We thank the reviewer for their positive assessment of our revised manuscript.

Reviewer #2 (Remarks to the Author):

I am happy with the responses and edits from the authors in the updated version of the manuscript. In particular, I find the across-lecture knowledge prediction and the "smoothness of knowledge" analyses to be important and interesting, and make a satisfying contribution to the full set of empirical results.

There a few new sentences that are hard to parse, along with some typos, so the authors should take care to make those fixes. I don't need to see an updated version, but if the authors find it useful I can give a few examples:

p. 14 - line 339 - "and and"

p. 23 - line 470 - "The 'spatial smoothness'... is being captured by the knowledge maps we are inferring from their quiz questions."

p. 36 - line 803 - "...from our dataset full dataset."

I should also note that new edits on lines 72 and 738 are cut off on the provided pdf and so I cannot speak to their content.

We thank the reviewer for their positive assessment of our revised manuscript. We have fixed each of the typos and wording issues identified above.

Reviewer #3 (Remarks to the Author):

The authors engaged very thoughtfully, and in great detail, with the comments of all reviewers, including my own. I sincerely thank them for taking the comments seriously, for clarifying misunderstandings on my end, for adding several very helpful analyses, and for expanding the manuscript. It is important to acknowledge the significant amount of work that went into this revision.

We thank the reviewer for their positive assessment of our revised manuscript and responses to the reviewers' comments.

My two main remaining concerns were expressed in my previous comments: the model's performance might not be as good as the manuscript claims (or it lacks a quantitative analysis demonstrating it), and it might not advance us beyond existing methods. Here, I describe how these concerns relate to the revisions that the authors made:

First, the central insights that the authors' method provides about learners' knowledge are about predicting which questions will be answered correctly / incorrectly, based on other questions that have been answered (Figure 6). In my opinion, this could be the most compelling part of the paper. However, it is currently analyzed incorrectly: the Mann-Whitney test requires that observations are independent, whereas the current data are grouped by (1) participant, and (2) question. Any tests that do not account for these groupings are anti-conservative (e.g., do not properly take outliers into account). The appropriate way to analyze grouped data is with linear mixed-effects models (LMEs; in this case, logistic). The authors mention that they chose to avoid such analyses because the overall approach is already relatively complicated. However, this is not a good reason to analyze data incorrectly. LMEs are the standard approach in several fields (e.g., psycholinguistics), and have been so for a while. They are straightforward to implement (1-2 lines of code in R) and interpret. I strongly believe that this analysis should be the one adopted.

As a point of clarification, our primary goal is *not* to predict which specific questions participants will answer correctly or incorrectly. Rather, our main insight is that fine-grained aspects of knowledge and learning can be captured automatically using our framework. The ability to predict performance on specific questions is a *test* of those knowledge and learning predictions. In other words, those tests in Figure 6 are intended as a “validation” of our framework (i.e., showing that the knowledge estimates are informative), as opposed to the primary “goal” of our approach.

The above notwithstanding, we have updated our manuscript as suggested. For each subset of questions and predictions tested in our previous Figure 6 (i.e., for each combination of (a) all questions, across-lecture predictions, and within-lecture predictions for (b) quizzes 1, 2, and 3), we test whether knowledge estimates are informative about participants' responses using generalized linear mixed models (GLMMs) with logistic link functions. Overall, our results are quite similar to what we previously reported (using *t*-tests in our initial submission and Mann-Whitney *U*-tests in our most recent submission), while also helping to clarify our framework's ability to generalize its knowledge predictions across content areas. We have also updated Figure 6 to better reflect the new GLMM model-based tests, and we have added a description of the GLMM approach to our revised *Methods* section.

Once this analysis is adopted, the authors should build two models: one that predicts accuracy (correct vs. incorrect) on each held out question only based on the % correct of that specific participant on questions from the same lecture; and another that predicts accuracy based on this same % correct measure in addition to estimated knowledge. Then, by comparing the two nested models (e.g., with the anova command in R), they could explicitly test whether (or, rather, by how much) the estimated knowledge helps to predict held-out questions over and above a simple % correct measure. This analysis would provide a direct evaluation of the predictivity of “estimated knowledge” against a baseline. It will thus add critical information beyond Figure 7. Currently, Figure 7 does not provide such a direct comparison: it shows that the probability of correct answers in the “neighborhood” of a reference question is starkly different depending on whether that reference question was correctly vs. incorrectly answered, but these stark differences do not translate to differences in “estimated knowledge” for each reference question, shown in Figure 6 – as I describe next.

As requested, we fit two models: one that incorporates a simple percent correct measure, and a second that additionally incorporates “estimated knowledge”. We compared these two nested models via a likelihood-ratio test (which the anova function the reviewer referenced performs for nested models) and computed a p -value using the parametric bootstrap procedure described in our revised Methods section. We found that incorporating estimated knowledge provides explanatory power above and beyond percent correct alone ($\lambda_{LR} = 17.452, p = 0.006$).

Although this analysis “works out” (i.e., displays the “effect” the reviewer is looking for), there are a number of issues we have identified with this suggested analysis that led us to decide not to include these results in our paper. Fundamentally, these issues relate to what (in principle) the measured proportion of correctly answered questions can possibly tell us about held-out questions.

One issue is that, mechanically, there is a practical confound in carrying out the proportion correct computations on held-out questions. We describe this on page 16:

“...our null hypothesis is that the knowledge estimates we compute based on the quiz questions' embedding coordinates do *not* provide useful information about participants' abilities to answer those questions. What result might we expect to see if this is the case? To gain an intuition for this possibility, consider the expected outcome if we carried out these same analyses using a simple proportion-correct measure in lieu of our knowledge estimates. Suppose a participant correctly answered n out of q questions on a given quiz. If we hold out a single *correctly* answered question, the proportion of remaining questions answered correctly would be $(n - 1)/(q - 1)$. If we hold out a single *incorrectly* answered question, the proportion of remaining questions answered correctly would be $n/(q - 1)$. In this way, the proportion of correctly answered remaining questions is always *lower* when the held-out question was answered correctly than when it was answered incorrectly. Because our knowledge estimates are computed as a weighted version of this same proportion-correct score (where each held-in question's weight reflects its embedding-space distance from the held-out question; see Eqn. 1), if these weights were uninformative (e.g., randomly distributed), then we should expect to see this same inverse relationship between estimated knowledge and performance, on average. On the other hand, if the spatial relationships among the quiz questions' embeddings *are* predictive of participants' knowledge about the questions' content, then we would expect *higher* estimated knowledge for held-out correctly (versus incorrectly) answered questions.”

In other words, raw proportion correct measures (if uncorrected) make exactly the wrong “predictions” about the held-out questions. We observe this in both the “percent correct alone” and “percent correct and knowledge” GLMMs, wherein percent correct in both models is assigned a negative coefficient that (per a Wald Z test) is significantly different from 0 (percent correctly only model: $\beta = -5.935, Z = -4.127, p < 0.001$; percent correct and knowledge model: $\beta = -6.7455, Z = -4.535, p < 0.001$).

Even if one were to correct for this mechanical bias (e.g., by using some sort of Bayesian and/or bootstrap-based approach), then proportion correct will be exactly *equal* for any held-out question. So

even in this “best case” scenario, where proportion correct could be estimated in an unbiased way, it would then not carry *any* predictive information about the held-out question.

More broadly, our main claim is that estimating knowledge (i.e., by taking question-specific content into account via text embeddings) can provide more nuanced insights into what students know than simply computing the proportion of questions they answer correctly. The across-lecture and within-lecture tests (Fig. 6) also show that those estimates can (a) extend across content areas and (b) enable us to “zoom in” within a given content area— illustrating two additional aspects of our approach that have no analogs in a framework that considers only the proportions of correctly answered questions.

The results the reviewer mentioned from Figure 7 are also relevant. Those analyses show how “far” (in the text embedding space) the influence of “knowing” or “not knowing” about the content at the embedding coordinate of a given question “spreads” through the embedding space. There are two findings worth noting. First, the analyses in Figure 7 provide another means of showing that knowledge space is “smooth”—i.e., that knowledge about a given concept implies knowledge about other concepts that are nearby in the embedding space. (And similarly, *lack* of knowledge about a given concept implies lack of knowledge about other nearby concepts.) Second, the analyses in Figure 7 show that if you travel far enough in the embedding space, eventually the content-specific insights we get about what a participant knows based on how they performed on a given individual question becomes statistically indistinguishable from simply knowing the overall proportion of questions they answered correctly.

While the results in Figure 6 are statistically significant for the appropriate quizzes/topics, the patterns themselves are a bit puzzling: in many of the analyses, the distribution of estimated knowledge for questions answered incorrectly is highly skewed towards 1 (perfect knowledge). Even though this distribution is significantly less skewed than the distribution of estimated knowledge for correctly answered questions, the “effect size” of the skewness appears large to me. Specifically, between 20% and 50% of these incorrectly answered questions are predicted to have a knowledge of 1. This is quite a problem for any actual application of the authors’ method, and puts a strong limit on the claim that this method provides high-resolution insights about participants’ knowledge.

We appreciate the reviewer’s point that the distributions of knowledge estimates for correctly versus incorrectly answered questions that we had included in our previous submission are confusing. Those distributions (as presented in our previous submission) are somewhat difficult to interpret in the way the reviewer is attempting here, since each “observation” reflected one knowledge estimate from one participant (i.e. the full distributions comprised many observations from each person— so, for example, participants who answered nearly all questions correctly would contribute to the extremes of those distributions). We have overhauled the entire figure (Fig. 6) and analysis using the GLMM framework the reviewer had suggested in their prior comments. In our revised Figure 6, we show a more intuitive (we think!) depiction of how predicted performance changes with estimated knowledge, in each sub-panel. Specifically, for each level of estimated knowledge we display the proportion of correctly versus incorrectly answered questions.

For example, a very rough and inaccurate estimation from the top right panel of Figure 6 suggests that the probability of a correctly answered question having a higher estimated knowledge than an incorrectly answered question is ~ 0.56 (chance is 0.5). This is very low for any practical purpose. I did this by creating a ROC curve that attempts to classify questions as “correct” or “incorrect” based on estimated knowledge and computing the area under the curve (AUC). The AUC also equals the Mann-Whitney U statistic divided by the product: $\text{num_answered_correctly} \times \text{num_answered_incorrectly}$. My sincere apologies if my calculation is way off and I made a mistake somewhere. But if it is in the ballpark, then the “predictivity” of knowledge estimates, while significant, is quite limited. In that case, the claims in the papers should be made less bold.

In our previous version of this analysis (from our prior submission), the reviewer makes a good point that predictive power was low when the majority of knowledge estimates are within a narrow range. However, now that we have revised our analysis (at the reviewer’s good suggestion!), we now have considerably more statistical power to predict performance on individual held-out questions. Previously, we were essentially considering only the knowledge estimate values themselves in “classifying” whether a question would be correctly or incorrectly answered. Now, however, using GLMMs, we are able to more closely examine the effect of estimated knowledge on the likelihood of correctly answering a held-out question while accounting for the facts that (A) some participants may be more likely to correctly answer a question than other participants, and (B) some questions may be more likely to be answered correctly than other questions (i.e., by fitting our models with random intercepts for participants and questions). In other words, while the overall range of estimated knowledge values is generally smaller when, for example, most participants answered most questions correctly (such as on Quiz 3), we are now better able to parse out the relationship between estimated knowledge and likelihood of answering a held-out question correctly *within* the range of estimates spanned by individual participants and/or questions.

To provide some additional context, the specific case the reviewer is referencing relates to our predictions of Quiz 3 responses that include all questions (regardless of content area). As can be seen in the panel, performance on Quiz 3 is quite good overall—i.e. the majority of questions are answered correctly (shown by the blue distributions). However, as also shown in the panel, the predicted probability of answering a held-out question correctly is *higher* when estimated knowledge is higher, as opposed to when estimated knowledge is lower. In other words, even in this case where performance is near ceiling, our knowledge estimates can still reliably distinguish between questions that are more vs. less likely to be answered correctly, based on a participant’s performance on the other questions from the same quiz. We have pasted in the relevant panel below (including the relevant statistical results from the GLMM analysis of that panel’s data) for reference; the *y*-axis denotes $p(\text{correct})$, blue denotes the proportions of correctly answered questions at each estimated level of knowledge, red denotes the proportions of incorrectly answered questions, and the black line displays the LME model fit to the data:

Quiz 3

Other than estimating knowledge on held-out questions, the other insights provided by the authors' method are somewhat limited in my opinion. For example, the maps in Figure 8 only show that students learn the lecture that they've watched (i.e., this is where the "change in knowledge" is apparent). This is a sanity check about the algorithm for creating the maps, but it is in itself not an independent insight, as it is derived from the information in Figure 2 and the analyses reported in the text (shown in Supp. Figure 2): the embedding of each question is closer to the trajectory of the corresponding lecture, and farther from the trajectory of the other lecture. The word clouds in Figure 8c are also a reflection of the properties of the embedding space that were verified earlier in the text. The maps indeed have finer-grained structure, but such structure is currently not quantified and remains interpretable (so these are, at best, untested "hypotheses", not insights).

Our main goal is to build a model that helps to explain and track what people know and how they learn. In Figure 8, we show how our approach may be used to visualize detailed (high spatial resolution) "maps" of knowledge and learning. In fact, those maps effectively have "infinite" resolution, since the level of detail is constrained only by the sampling resolution used to construct those maps. However, it would be computationally intractable to attempt to directly test those predictions about the "infinitely many" concepts reflected by those sorts of maps. Instead, we focus on a similar (but more practical) variant of the same idea by estimating knowledge at the embedding coordinates of held-out questions.. Specifically, we construct knowledge estimates using some of the data and then see whether estimated knowledge about the content probed by held-out questions corresponds to some extent with participants' chances of answering those held-out questions correctly.

We show (e.g., in Figures 6 and 7, along with the analyses described above that the reviewer requested) that our approach provides predictive information about participants' knowledge of held-out material.

The maps shown in Figure 8 are intended to serve several purposes. First, they provide a proof of concept that our knowledge estimation procedure can be scaled up to estimate very large numbers of coordinates in text embedding space, to the point that we can construct smooth "maps" of knowledge and learning. Second, Figure 8 shows that the specific estimates reflected in those maps make some intuitive sense. For example, on average, participants who just watched a given video show an increase in knowledge around the coordinates spanned by that video. As the reviewer notes, this provides a nice "sanity check" that the structure captured by our framework is "reasonable." However, it is also important to note that the actual predictions we make about specific questions (or, more generally, knowledge about held-out content) come from *learner-specific* information (i.e., individuals' quiz performance, rather than the across-participants average performance), which is better captured by the individual level maps shown in Supplemental Figures 7, 8, and 9. Whereas the average maps show something akin to "what people learn in general from watching a given lecture video, or a sequence of lecture videos," the individual participants' maps tell us about what *that specific individual* knows, and how their knowledge changes over time with training. Those individual-level knowledge estimates and maps are what we claim could help drive insights, e.g., by building those maps into automated tutoring systems, (human) teacher feedback systems, and so on.

(Another potential issue with interpretability is that of the topics themselves: for instance, do the authors observe that the model recovers distinct topics for each of the four fundamental forces, i.e., a topic or few that correspond to gravity, others that correspond to the strong interaction, etc.? If so, saying so explicitly is important to demonstrate the strength of the method. If not, then this is a limitation that should be acknowledged, because it's one of the "minimum" requirements for any representation of the "topics" of that lecture, at least in the intuitive sense of "topics").

To clarify, within the Latent Dirichlet Allocation framework (Blei et al., 2003), a "topic" is defined as a distribution (of weights) over words in the model's vocabulary. (The "weights" are modeled as a draw from a Dirichlet distribution, hence the name of the approach.) In other words, a topic model's "topics" do not come with ready-made human-readable labels. This is the case for nearly all modern text embedding approaches: the "dimensions" or "features" discovered by text embedding models tend to be more complicated or nuanced than easily namable concepts like "gravity" or "strong interaction" and so on. Therefore one would not expect arbitrary human-readable concepts (e.g., selected post-hoc) to "fall out" along fully separable topics in the model as the reviewer is suggesting. Blei (2012, Communications of the ACM) provides a nice intuitive overview of what the "topics" in topic models mean, versus what might be colloquially meant by the word "topic." In our paper we use the word "topic" to mean "a topic discovered by LDA"—i.e., a weighted distribution over words, whereas we use the word "concept" to represent something closer to what the reviewer is suggesting here. Formally, however, a "concept" in our framework would simply be a coordinate in text embedding space (like would be obtained for any other text).

What we *can* ask is whether the representations obtained by the model can describe the content of the lectures, questions, and students' knowledge in meaningful or useful ways. We test this in several ways:

- In Figure 2C, we show trajectories of each lecture along with the embeddings of each question. We can see visually that:
 - The blue line (Lecture 1 video) and green line (Lecture 2 video) unfold over an extended region of text embedding space, as opposed to being confined to a single point. This shows that different moments in each video are being embedded in different ways (i.e., as different blends of discovered topics).
 - The blue dots (questions about Lecture 1) tend to map onto coordinates near the blue trajectory (Lecture 1 video), whereas the green dots (questions about Lecture 2) tend to map onto coordinates near the green trajectory (Lecture 2 video). This tells us that the deeper conceptual content of the questions about each video are being captured using similar blends of topics as the videos themselves. In other words, the questions' embeddings "match up" with the appropriate regions of text embedding space (that are covered by the corresponding lecture videos). This is an important result, because the questions are not included in the topic model's training set.
- In Figure 3, we can see (in a different way) that the topic embeddings "match up" questions about each video with the correct videos:
 - The specific topics reflected by the videos overlap with the specific topics reflected by questions *about* each video (Fig. 3A)
 - The specific topics reflected by questions about Lecture 1 do *not* match up with the topics reflected in Lecture 2, and vice versa (Fig. 3B)
- In Figure 4, we can see an even more nuanced "matching," whereby we show estimates of the precise time course over which the content of a given question is covered by the corresponding lecture. In other words, it's not simply that every question is being mapped onto the "average" coordinates of the appropriate lecture in text embedding space. Rather, each question reflects a specific (limited) part of each lecture.
- In Figure 6, we show that we can use the knowledge estimates (derived using the topic embeddings of each question) to partly explain which held-out questions participants answered correctly versus incorrectly. In other words, the knowledge estimates relate to participants' behaviors.
- In Figure 7, we show how the proportion of correctly answered questions falls off with spatial distance from the text embedding coordinate of a correctly or incorrectly answered question. That those plots are structured (i.e., neither random nor "flat") indicates that the text embeddings are capturing some conceptual content of the questions *in ways that are predictive of participants' actual performance*.
- In Figure 8, we show how estimated knowledge at specific (arbitrary) regions in text embedding space changes with training. Again, that these maps are structured (and that they behave in reasonable/expected ways as training progresses) indicates that we are capturing meaningful aspects of what the participants know.

Similarly, the “moment by moment” content mastery is an interesting visualization, but there is no quantification of whether it is accurate. If I understand correctly, this visualization is derived from the information in Figure 4, so it again does not provide any independent insight that can be tested / demonstrated. More importantly, the identification of lecture snippets that correspond to each question, shown in Figure 4, is still unfortunately not quantitatively verified. I have proposed several ways in my previous comments about how the accuracy of this result can be measured, and I think they are relatively straightforward to implement in an online study (or the authors could just ask physics instructors to map each question to the corresponding part of the lecture).

The authors manually inspect the lecture snippets that were automatically identified based on each question, and report that they are overall good (providing all the data in the Supplementary Materials is very helpful!). But this is a subjective impression. For instance, Question 26 is said to correspond to two parts of the lecture, but one of them provides the right answer, whereas the other provides the wrong answer—so giving both snippets to a student who answered the question wrong might cause further confusion (as an anecdote, ChatGPT could answer this question correctly even without the lecture text, so there is nothing inherently “hard” about the question). Similarly, for Question 7, one of the identified paragraphs is quite irrelevant. For Question 4, the algorithm identifies over 2.5min of text, and this entire text does not contain the right answer (“the weak interaction really applies to very small distances”), perhaps because the word “computer” throws the algorithm off. Same for Question 11, where the identified text does not include anything about the strong force. I hope that these examples demonstrate why I believe that a quantitative estimate of the accuracy of the authors’ method is important.

We appreciate that the reviewer is asking for a specific way of characterizing the “quality” of the automatically identified matches between the identified timepoints in each lecture that each lecture-specific question is “asking about.” We provided a supplemental table reporting *all* of the matches, for every question in our dataset, to enable interested/motivated readers to judge for themselves the quality of each match. To our reading of that table (Supp. Tab. 3), most (though, as the reviewer points out, not *all*!) of these automatically identified matches appear to align quite nicely with the corresponding lecture snippets. For example, in nearly every case, the identified snippets discuss or describe whatever concept the question is asking about. The reviewer is also correct, of course, that there are a few examples where the alignment appears to be “off” or incorrect.

If our end goal was to show that we could match up questions with lecture snippets, then a new experiment along the lines of what the reviewer is suggesting might be worth running. However, the goal of the *current* paper goes beyond matching. If the matching process was truly random or uninformative, then (a) questions should not align to the correct lectures, as shown in Figure 3B, and (b) knowledge estimates generated using the questions’ embeddings should carry no predictive information about participants’ quiz performance. Further, (c) the knowledge maps in Figure 8 would not show increases in knowledge around the just-watched lectures (e.g., an increase in knowledge around Lecture 1’s region of topic space in Quiz 2, and an increase in knowledge around Lecture 2’s region of topic space in Quiz 3). That we can predict participants’ performance in several ways, and that using the text embedding to characterize performance adds interpretable structure to the problem of characterizing and tracking participants’ knowledge, suggests that the embeddings are indeed capturing meaningful content about the course materials. Our main claim is that *leveraging* that

captured content can help us understand what learners know at a finer-grained level of detail than simply ignoring those detailed characterizations about question and lecture content that are captured by text embeddings.

My second main concern regards the usefulness of the authors' method with respect to other, existing tools. The authors now provide an analysis of BERT, which is nice and informative, but I would like to still push back on two fronts. The first one is less important: given that BERT represents so many topics, as well as language structure, it is expected that its lecture trajectories would be "squashed", and that the questions would be represented somewhat separately (e.g., they have different syntax). But this does not necessarily mean that the critical information is not there. For instance, in static word embeddings like Word2Vec, antonyms ("good" vs. "bad") are located in close proximity, and this "squashing" of opposites could be said to be bad for a representation of meaning; nonetheless, the direction in the embedding space that connects antonyms defines a subspace that is meaningful, such that even words that are quite far from the "good" and "bad" vectors can have their valence estimated based on their position relative to that subspace. This is just one example of how complex knowledge exists, and can be extracted, even when at first glance it appears to lack desired properties. (By the way, similarities in BERT are also known to be influenced by "rogue" dimensions, such that the activations of each artificial unit should be z-scored based on a corpus; for instance, see: <https://arxiv.org/abs/2109.04404>. Personally, I don't think this is going to make much of a difference; I'm mentioning it just in case the authors use similar analyses for other projects).

The main issue we're describing with the BERT embeddings (e.g., in Supp. Fig. 6) is not only that the embeddings are "squashed," or that the lectures and questions use different syntax (although we agree that these both pose challenges). Rather, the larger "problem" is that the questions are not represented (very) differently from *each other*. An easy way to see this is to visually compare the timecourses of the correlations of each lecture 1 and 2 question for the text embeddings derived using LDA versus BERT, in Supplementary Figure 6. Whereas the LDA-derived embeddings for each question have a (largely) unique time course, the BERT embeddings for each question neither distinguish between different moments of the lectures, nor do they distinguish between lectures. This is quantified more formally in the bottom panels of Supplementary Figure 6. The LDA-derived embeddings for each lecture are strongly correlated with the embeddings of the *matching* lecture, and they are strongly *negatively* correlated with the embeddings of the other lecture, or of questions about different material. This tells us that the LDA embeddings are *specific* to each content area. In contrast, the BERT embeddings for every content area (across both lectures and all three sets of questions) are *all* strongly positively correlated. This tells us that BERT embeddings are *not* specific to each content area at the level of detail required for this study.

The reviewer makes some interesting points about how BERT's difficulties in this domain might arise due to the number of feature dimensions, and we appreciate the "rogue dimensions" point along with the preprint reference. Our intuition has been that those difficulties are more about *training* than about the *dimensionality* per se. Because BERT is *trained* on a very general corpus, its embeddings need to "explain" a very broad range of content. Even though BERT has more feature dimensions than our LDA model, BERT needs to "spread" those dimensions across a much wider range of content than

our LDA model (which only needs to “explain” the content of two lectures and a few dozen questions).

Although it is beyond the scope of our current paper to fully explore this, we suspect that a fine-tuned version of BERT, or fine-tuned GPT model, could do a much better job at describing the nuanced lecture and question content— perhaps even better than our LDA approach. The deeper point we were trying to make by comparing LDA vs. BERT isn’t that LDA is necessarily the “best” model, or even that it’s necessarily better than BERT or other transformer-based models. We are simply arguing (e.g., pages 27–28) that there may be a fundamental tradeoff between “generalizability” and “specificity.” When models are trained or fine-tuned on a highly specialized corpus, they can achieve high resolution in that content area. However, those specialized models would be unlikely to generalize well to other domains. When the same model architectures are trained on a much broader corpus, they can generalize well but have relatively poor performance when you “zoom in” sufficiently far in any given domain. The same is true even for *very* large models, like ChatGPT-4. You can see this if you attempt to do something highly specialized, like playing a game of chess against the default version of ChatGPT-4 (i.e., a model that contains knowledge about the rules of chess through its training corpus, but that has no chess-specific fine-tuning).

In other words, our use of LDA isn’t intended as a statement about LDA alone. LDA could be easily swapped in for another model that is able to capture the lecture and question content to a sufficient extent, without changing any of the fundamental assumptions or approaches we are proposing. When we compare LDA’s performance (when LDA is fit to sliding windows of the the lecture transcripts) versus BERT’s performance (when BERT is fit to a much more general corpus) we are not trying to claim (nor do we believe) that LDA is somehow a “better model” than BERT, or any other model. Rather, our point is that even a much larger model, fit to a much larger corpus, fails to capture fine-grained aspects of the lecture and question content to the extent that a much simpler model like LDA can, when fit to this very specific dataset. It could very well be the case that a fine-tuned version of BERT (or nearly any other widely used modern language model) could explain the lecture and question content even better than LDA.

In our paper, we show a “proof of concept” that a model that captures the conceptual content of lectures and questions, up to some “sufficient” level of detail, will be able to uncover aspects of knowledge and learning that go beyond what we can achieve using traditional “proportion correct” measures alone. And in the best case, we can start to construct complete maps (of the sort we propose in Figure 8) of what people know and how their knowledge changes over time, e.g., with training.

Comparing LDA versus other models is tangential to that goal of establishing a proof of concept. In other words, if a simple “fine-tuned” model can explain the relevant content sufficiently well to begin to explain some aspects of participants’ performance on the quizzes (over and above “proportion correct” measures that do not directly attempt to formally capture the conceptual content of the lectures or questions), then we might assume that some more sophisticated model that captures the conceptual content even better might *also* explain participants’ behaviors more fully. But our goal is

not to find the “best possible” model; we are solely trying to establish that proof of concept that capturing the conceptual content of the lectures and questions can provide some explanatory power of participants’ behaviors.

Second, and more importantly, BERT is a model that is quite good at linguistic representation, but not very good at reasoning (it is also quite small by today’s standards). It does show some evidence of learning “topics”, similarly to those the authors talk about (for instance, <https://arxiv.org/pdf/2203.14680.pdf>), but it is still quite limited. In contrast, models trained using reinforcement learning from human feedback exhibit much better reasoning abilities (for instance: <https://www.nature.com/articles/s41562-023-01659-w>). It is possible that these are the abilities that are required for, e.g., estimating knowledge based on questions. The ability to identify which span of text is relevant for answering a particular question can already be done with high accuracy by modern models (e.g., <https://paperswithcode.com/sota/question-answering-on-squad20>). Moreover, state-of-the-art, publicly available systems might have some important advantages over the authors’ model.

We appreciate these points and these references. While interesting, the “reasoning” capabilities of the text embedding models we employ is (in our view) somewhat tangential to our main focus. Neither BERT nor LDA are good at “reasoning” in any direct sense of the word. As explained above, our goal is not to pit different model architectures against each other. Our point in including comparisons between LDA and BERT is to show that not every model (even if large and complex, and even if fit to a much larger dataset) will be able to immediately explain the conceptual content of the lectures and questions sufficiently well. In other words, simply swapping in a model trained on a very general corpus (even if that model is large) is unlikely to yield good knowledge estimates. We think this is related to the “generalizability versus specificity” issues we discuss on pages 24–25. In particular, for a given model architecture and number of parameters, training the model on a broader corpus will yield greater generalizability across content areas, but less specificity in any given content area.

We also wish to clarify that the objective of our text matching demonstrations using LDA (Fig. 4, Supp. Tab. 3) is not only to match up which snippets of the lecture transcript are providing the information relevant to a given question. The text matching results we present in Supplemental Table 3 are intended to show that, overall, the moment-by-moment match we compute using correlations between topic vectors (Fig. 4) appear to also, in most cases, correspond to reasonable matches between the lecture transcripts and question text. But that is only a sanity check. What we actually “care about” in our approach is the quantifications of those matches—i.e., we need to know how much to weight the given question when we want to estimate knowledge about a given part of the lecture. Finding the best matching lecture snippet does not (in and of itself) provide a “number” that we can use to weight the contributions of different questions in estimating knowledge.

The authors state in their rebuttal “suppose one were to ask ChatGPT to match up each question with some part of a lecture. What would one “do” with that response?”. My answer is: whatever one would do with the information in Figure 4. If the data in that figure are used as a test of the authors’ model, then I am assuming those data have some use. For instance, you could refer a student who answered a question incorrectly to the relevant part of the lecture for review. The authors also state “ChatGPT also has no built-in mechanism for keeping track of what the student

knows, or how that knowledge might relate to the content of a course the student is learning from”; and “ChatGPT has no internal machinery for representing or tracking the learner’s knowledge, nor does it maintain a “theory of mind” of the learner, nor does it (in and of itself, to the best of our own understanding) have any deep understanding of the material itself”. The authors’ model also does not have any deep understanding of the material (and, if anything, understands less because it does not capture the structural dependencies between words in a text). The model also has no substantial theory of mind: what it does is implement a notion of “knowledge = vectors in a continuous space”, but ChatGPT and similar models rely on the same assumption, and hence implement this same notion. Being a high-dimensional embedding, ChatGPT has the capacity to hold and process all the information that the authors’ model stores; the fact that it is very hard (at least for me) to conceive of how it might do so, does not mean that it cannot do so or does not do so. There is active work on the ability of large language models to have a “situation model” or a “discourse model” and update it as they process more input. Also, professors are using these models to read students’ papers and fill out detailed rubrics with feedback, so these models can convey high-resolution insights about student performance. Models like ChatGPT are also generative, so they can go well beyond the authors’ model by, e.g., writing new questions. In this sense, these models could be more useful in practice because they can translate insights into actions.

We suggest that the reviewer may be misinterpreting the intended message of Figure 4. To clarify, the goal of Figure 4 is to show that LDA-derived topic vectors yield temporally “specific” matches between the questions and lectures. In other words, most questions’ correlation time series plots have a single “peak” (or some have two peaks), and most questions’ peaks occur at different parts of the lectures. That tells us that the particular “blend” of questions that will be used to estimate knowledge about each moment in the lectures (e.g., as in Figure 5) is largely unique to each moment of each lecture. If we contrast those correlation time series plots with the plots obtained using BERT (Supp. Fig. 6, right panels), it becomes clear that the BERT embeddings behave very differently: individual questions show relatively equal “weight” (i.e., correlations) over lecture timepoints, and all of the questions’ time series plots look very similar. Therefore the BERT-derived “blend” of questions used to estimate knowledge about any given lecture moment would be highly similar across moments (approaching something much closer to a “percent correct” measure).

Our point about the need to “do something” with ChatGPT’s response refers to what is needed if we want to quantify knowledge, as opposed to solely describing what someone knows in a qualitative way. Even if we know which snippets of a lecture transcript “match” with a given question, we still want to turn those matches into a knowledge estimate. That lets us move beyond conceptualizing knowledge as a binary “correct or incorrect” feature, towards instead capturing more continuous estimates of knowledge that integrate performance across *several* related questions.

The reviewer also points out that LDA has no internal representation of what a participant knows. We agree! What we are doing is using the text embeddings to define the coordinates (dimensions) of a *separate* space for characterizing and tracking knowledge. In other words, LDA gives us a way of mapping out where concepts “go” in the space, so that we can appropriately combine participants’ responses in order to build accurate estimates of what they know. This requires only that LDA maps

similar concepts onto nearby coordinates– it does *not* require LDA to maintain its own internal representation of what the participant knows.

As an aside, the activations from ChatGPT’s transformer module (if one had access to them) or from a similar model (e.g., Llama 2 70B) could (in principle) be used in a similar way to how we are using the LDA embeddings. The downside of using large “off the shelf” language models that are trained on huge (and diverse) corpora is that (due to the generalizability versus specificity tradeoff we discuss above and on pages 24–25), those models won’t necessarily have good “resolution” within an arbitrarily chosen small area within their embedding space. However, a fine-tuned version of ChatGPT (or another similar model), if one had access to the weights, could potentially be used to define even better or more accurate “knowledge maps.”

Finally, the reviewer makes a good point about how ChatGPT “translates insight into action.” Using ChatGPT to generate questions related to some content area, or to auto-grade essay questions is fully compatible with and complementary to the framework we are proposing. Those automatically generated and scored questions could serve as the input data to the sort of knowledge estimates and “maps” that we are deriving with our framework. One could even imagine an “active sampling” approach whereby mapping out what someone knows entails (a) continually identifying the part of their “knowledge space” (i.e., a region of text embedding space) that we are maximally uncertain about, (b) using ChatGPT to generate and score a question about that content area, and (c) updating the estimated knowledge map with the new information.

As an anecdotal example, I fed ChatGPT the text of the Four Fundamental Forces lecture, followed by some prompts and questions. Below are the first responses that I received in this interaction, unless otherwise specified, so there was no “tweaking” (sure, slightly different prompts might have given different results, but that is beside the point in my opinion; once we know the right prompt to use, there is no reason to use others).

[the input below was given immediately after the text of the lecture]

Answer the question below:

Why don't you and your computer experience any attraction or repulsion due to the Weak Interaction?

- A. The weak interaction only acts over extremely small distances
- B. The weak interaction between you and your computer is counteracted by the other forces
- C. You and your computer have no net charge
- D. Neither you nor your computer has enough mass to induce a significant Weak Interaction

 **ChatGPT**

- A. The weak interaction only acts over extremely small distances

You

Please type the part of the lecture that helped you answer the question above

ChatGPT

The part of the lecture that helps answer the question is:

"Now, the next strongest force-- and just to give a sense of how weak gravity is even relative to the weak interaction, the weak interaction is 10 to the 25th times the strength of gravity. And you might be saying, if this is so strong, how come this does it operate on planets or us relative to the Earth? Why doesn't this apply to intergalactic distances the way gravity does? And the reason is the weak interaction really applies to very small distances, very, very small distances. So it can be much stronger than gravity, but only over very, very-- and it really only applies on the subatomic scale. You go anything beyond that, it kind of disappears as an actual force, as an actual interaction."

[note that the above is a better than the authors' algorithm, which does not identify the relevant lecture snippet for this question; in addition, the snippet is ~1min long, compared to the author's algorithm which identified ~2.5min of text]

You

A student read the lecture above, and answered the following questions correctly:

1. Why is the gravitational attraction between you and your computer too small for you to notice?
2. Roughly how many times stronger is the Weak Interaction than gravity?
3. Why don't you and your computer experience any attraction or repulsion due to the Weak Interaction?

But that student got the following questions wrong:

1. Why does the universe have a very uneven distribution of mass but a relatively equal distribution of charge?
2. In your body, there are a tremendous amount of negatively-charged electrons. Your computer also contains a huge number of negatively-charged electrons. We know that like charges repel, but you and your computer are not repelled apart. Why?
3. What does the Coulomb Force refer to?

Based on this pattern, which concepts did the student successfully learn, and which concepts did the student not learn well?

[I first tried to get ChatGPT to immediately generate new questions based on the above prompt, instead of asking first which concepts the student learned well; that did not work well, which is why I added this "intermediate" step above]

ChatGPT

The student seems to have a good understanding of the concepts related to gravitational attraction, the relative strengths of the weak interaction compared to gravity, and why the weak interaction is not noticeable between everyday objects. However, there are challenges with concepts related to the distribution of mass and charge in the universe, the behavior of like charges in your body and the computer, and the specific reference to the Coulomb Force.

Concepts the student successfully learned:

1. **Gravitational Attraction:** The student understands that the gravitational attraction between objects like you and your computer is too small to be noticeable in everyday situations.
2. **Weak Interaction Strength:** The student correctly grasps the idea that the weak interaction is much stronger than gravity, but its effects are limited to very small distances.

You

Given what this specific student seems to know vs. what they do not know, write one new question that you expect them to answer correctly and one new question that you expect them to answer wrong.

ChatGPT

Expected Correct Answer:

Question: Why is the weak interaction more noticeable on a subatomic scale rather than on larger scales like intergalactic distances?

Expected Incorrect Answer:

Question: What is the primary factor responsible for the attractive forces between positively charged protons in an atomic nucleus?

Another example:

You

Here is another question: Which force is stronger than the Electromagnetic Force? Could you please write which part of the lecture would help someone answer that question?

ChatGPT

The force that is stronger than the Electromagnetic Force is the Strong Force. The part of the lecture that provides this information is:

"Now, the strongest of the force is probably the best named of them all. And that's the strong force. That is the strong force... And once again, just to keep it in mind relative to gravity, it is 10 to the 38th times the strength of gravity. Or it's about 100 times stronger than the electromagnetic force."

This part of the lecture compares the strength of the Strong Force to that of the Electromagnetic Force, highlighting that the Strong Force is much stronger than the Electromagnetic Force.

[This is Question 11, for which the authors' algorithm did not provide the correct snippet]

These examples are, of course, anecdotal. I did not conduct a thorough evaluation of ChatGPT. But I am showing to suggest that the authors are perhaps too hasty in dismissing large language models as lacking high-resolution insights because their semantic spaces are so large. These models might be able to do a lot of what the authors' model does, and even go beyond it by, e.g., constructing new questions. True, they cannot provide visualizations of continuous maps of a student's knowledge (Figure 5, 8), but such maps are only interpretable and useful insofar as they lead to some action (e.g., identifying which concepts to focus on for re-learning).

All this brings me back to the concern I expressed in my first review: the authors' algorithm is cool, but does it sufficiently advance us beyond the capabilities that already exist in large language models, which are available to students and instructors? I think that answering this question is important for deciding whether the paper merits publication in this journal.

These examples are great! We think this is an interesting application of ChatGPT. In addition to providing a practical tool for educators, one could imagine this approach being leveraged by students as a convenient way of searching a video to identify which parts are "about" the content of a given question. But as we describe above, this is a fundamentally different problem than we are tackling in our paper. Whereas ChatGPT is being used here to directly identify a snippet of text in a lecture transcript, our approach is asking: at each moment of the lecture, how *much* does the content overlap

with this question? We then *estimate* a snippet following those moment-by-moment characterizations. But the snippets we report are a way of demonstrating that the matching is “reasonable” (as judged subjectively by the reader). We don’t actually *use* those snippets in our computations. Similarly, although it’s very impressive that ChatGPT can provide a digestible summary of what the student does vs. doesn’t know, there is no way to quantify those estimates. The maps we propose in Figure 8 are our vision of what those quantifications of knowledge could look like—i.e., more naturally described or visualized as continuous “maps” of exactly how much a learner knows about any conceivable concept. Those maps could then be used to generate questions, search over tutorial materials, etc. (as the reviewer is suggesting). But they could also be used to quantitatively track progress over time, compare or “match up” students who might benefit from studying together, assigning tutors based on areas of expertise, and so on.

Finally, the reviewer’s point that one should not dismiss large models because their spaces are so large is well-taken, and we agree with the sentiment. In fact, in some of our other work (e.g., <https://github.com/ContextLab/chatify>), we are building tools for incorporating large language models into interactive tutorials. We think models like ChatGPT are incredibly promising and useful tools for education. However, even though ChatGPT is useful for many education-related tasks, here we are trying to build more quantitative (formal) characterizations of what learners know at different points in their training. This is needed to test specific predictions or theories, or to systematically track a given student’s knowledge over time, or to compare different students’ knowledge or learning rates, and so on.

Some minor thoughts:

- *The word-overlap analysis (Supp. Figure 5) is very helpful!*

We are glad that the reviewer found Supplemental Figure 5 helpful!

- *Perhaps change the naming of the quizzes to Quiz 0, Quiz 1, and Quiz 2? That way, the number corresponds to the number of lectures the students viewed (I found myself having to do a “minus one” operation every time I was reading something about the quizzes to figure out their position in the experiment). I realize this means re-generating all the figures, so I’m not sure it’s worth it...*

We have chosen to retain our current numbering system for the quizzes. We appreciate the reviewer’s point that it makes intuitive sense that “Quiz 0” happens before any lectures, “Quiz 1” happens after lecture 1, and “Quiz 2” happens after lecture 2. We tend to find “0-indexing” most intuitive as well. On the other hand, we’ve found that, in presenting and discussing this work, our audiences have tended (most often) to find “1-indexing” more intuitive.

- *The authors test whether “knowledge about a given concept implies knowledge about related concepts” and “knowledge about a given concept is predictive of knowledge about similar or related concepts”. It would be helpful to the reader if the authors explicitly articulated which theories from psychology and/or education*

do not predict this pattern of knowledge (otherwise, this is a test of the validity of the model, not a psychological implication of the findings).

We suspect that most theories that incorporate some notion of attribute similarity or feature similarity would be compatible with our findings. It would be difficult to imagine a reasonable theory of learning whereby learning about a given concept *decreases* knowledge about other related concepts. The advance here is not the notion that learning can spread, but rather that we have developed a way to directly measure and track that spreading (through text embedding space).

- *In Figure 3, and Supp Figure 2, the negative correlation between Lecture A and questions from Lecture B might in part be an “artifact” of the inclusion of only two lectures in the model. The more lecture are included, leading to more diverse topics, the more this pattern might decrease? (It is still expected that questions about a lecture will correlate with that lecture more than with other lectures, but the difference might not be as strong, especially for lectures on related topics).*

In the extreme, if we trained a model on a much larger and more general corpus (e.g., as in the BERT example shown in Supp. Fig. 6), we might expect the negative correlations to disappear. If trained on *too* broad a corpus, even the increased match between a question and the lecture it is about becomes indistinguishable from the match between the question and other related content (again, as in BERT). As we discuss on pages 24–25, there seems to be an interesting tradeoff between fine-tuning (specificity) and general tuning (generality). In our paper we do not attempt to characterize exactly where the boundary between specificity versus generalizability lies, or how training on corpora of successively broader “scopes” might affect the correlations between embedding vectors. However, it would be interesting to explore this in future work!

Reviewer #2 (Remarks to the Author):

The authors clearly put in a lot of work to address Reviewer 3's concerns. My judgment of the manuscript remains similar to what it was during the last submission, though I do have some comments relating to the new content/analyses and how it is described and interpreted.

We thank the reviewer for their positive assessment of our work, and for taking the time to revisit our manuscript to provide further feedback.

1. The "predicting success on held-out questions using estimated knowledge" section is the one that went through the largest overhaul. I still find the within-lecture and across-lecture analyses to be important, though it is unfortunate that with the new methods the results are harder to interpret. The authors do a good job of trying to explain the pattern of results (sometimes knowledge estimates do predict quiz question accuracy, and other times they do not), though it is not a strong set of results with clear empirical insights. However, since in my view the value of this work is in its methodological rather than empirical contribution, I do not see this as a reason to not approve the manuscript. That being said, I do have a (a) point of confusion, and (b) question/suggestion regarding the analysis.

In general, we view the main contribution of our work similarly to the reviewer (i.e., as primarily methodological, rather than primarily empirical). That said, we appreciate and agree that in the previous version of our manuscript, the results of these analyses were somewhat challenging to interpret. This was due to the effects of two separate issues with our analyses. The first was a statistical bias we noted in our previous response letter (and describe in greater detail in our response to comment 1a) that led to an inverse relationship under our null hypothesis between estimated knowledge for a quiz question and the probability of answering it correctly. The second was a conceptual issue (described in our response to comment 1b) that we identified in the course of the current revision with how we had constructed the "null" models used in our significance tests. As detailed in our response to comment 1b, we have now addressed both of these issues and report an updated set of results in our revised manuscript (pp. 14–19). We believe that our updated results are substantially easier to interpret and provide much clearer empirical insights into whether, when, and to what extent our knowledge estimates predict participants' success on individual quiz questions.

a. In the rebuttal document, the authors explain an issue regarding the models incorporating the percent-correct measure, in which using percent-correct alone results in a negative relationship between estimated knowledge and probability of answering a question correctly (rebuttal starting at page #3). Given this strange relationship, the authors say that they have decided not to include these results in their paper. However, in the paper, the authors do include analyses that relate to this negative null result (starting manuscript page #16). I assume that the authors are referring to slightly different analyses in their manuscript vs. rebuttal, but if they decided to include analyses with this negative relationship in the manuscript, why did they leave out other ones? The fact that I am not clear why certain analyses were included or excluded makes me feel like I am missing an important aspect of these analyses, but perhaps this is a fault of my own.

We apologize for the confusion. To summarize, Reviewer 3 had proposed two separate changes to our manuscript, one of which we adopted, and one we did not. Both changes involved using generalized

linear mixed models (GLMMs) to assess the predictive power of our knowledge estimates for held-out questions, and both were affected by an issue involving a “negative relationship” in the data to which these models were fit. Specifically, holding out individual questions from a participant’s quiz responses introduces an inverse (or “negative”) relationship between their success on a given held-out question and their proportion-correct score on the remaining (“held-in”) questions. The “issue” with this relationship is not that it is negative per se, but that (as we explain below) it is fundamentally not a relationship that can be estimated numerically, and therefore incorporating it into a model of participants’ success on held-out questions (i.e., by including their remaining-proportion-correct scores as a predictor) leads to intractable issues with the model’s estimation.

The change we declined to adopt was the addition of a new analysis that entailed fitting GLMMs with fixed effects for participants’ remaining-proportion-correct scores in addition to their estimated knowledge. As described below, we identified a number of issues with this approach, one of which is that these models cannot be accurately estimated. The other change Reviewer 3 proposed (and which we *did* adopt) was replacing the Mann-Whitney *U*-tests previously reported in Figure 6 with analogous tests based on GLMMs. While the models we fit for these tests didn’t directly consider participants’ remaining-proportion-correct scores (only their estimated knowledge), under our null hypothesis, our knowledge estimates reduce to remaining-proportion-correct scores and exhibit the same problematic negative relationship with success on held-out questions. (This is the “negative null” we described in our manuscript when introducing this set of results.) In this case, our inability to accurately model this “negative null” relationship led to a conservative bias in the significance tests presented in our previous submission, which we have now corrected for in our revised manuscript.

We describe this “negative relationship” issue and its effects on both proposed analyses in greater detail below. We describe our approach to correcting for it (in the analyses shown in Figure 6) in our response to comment 1b.

First, suppose a participant correctly answers n out of q questions on a given quiz. If we hold out a correctly answered question from their quiz responses, their proportion-correct score on the remaining questions will be $(n - 1)/(q - 1)$, whereas holding out an incorrectly answered question would yield a remaining-proportion-correct score of $n/(q - 1)$. In this way, any given participant’s remaining-proportion-correct score will always be *lower* for held-out questions they answered correctly than for those they answered incorrectly—in other words, within participants, remaining-proportion-correct is inversely (or “negatively”) related to success on held-out questions.

In both of the GLMM-based analyses Reviewer 3 proposed, they requested we fit models with random effects for participant and question identities to account for potential groupings by those factors in our data. Fitting these models therefore entails (in part) estimating any fixed effects they include based on each individual participant’s (and question’s) observations, separately. For the analysis we declined to adopt, which involved fitting models with fixed effects for remaining-proportion-correct, this would include estimating the “negative” within-participant relationship described above. However, this isn’t a statistical relationship we can estimate through regression, but rather it is simply a mathematical byproduct: each participant’s remaining-proportion-correct score can take on only two possible values

$(n/(q - 1)$ or $(n - 1)/(q - 1)$, for their particular n), and which of those values it takes on for a given observation is fully determined by the value of the response variable we're trying to predict (i.e., their success on the held-out question). In other words, this relationship is perfectly deterministic and lacks any residual variance that would allow for statistical estimation. Attempting to fit a logistic model to this relationship results in "complete separation" in the response variable, as the combination of participants' identities and their remaining-proportion-correct scores perfectly distinguishes between correctly and incorrectly answered held-out questions. If both of these terms were modeled as fixed-effect predictors (e.g., as in a standard logistic regression), this would lead to infinite parameter estimates, making the likelihood function non-identifiable and causing the model to fail to converge altogether. Modeling participants as random effects masks this problem to a degree, as partial pooling across random-effect levels shrinks these estimates towards zero such that it is *possible* for the model to converge (hence our ability to report a result from this analysis for illustrative purposes in our previous response letter). However, this "shrinkage" does not "fix" the underlying circularity in the model's formulation, nor the complete separation it creates. Instead, the model the reviewer had proposed would produce unstable and misleading results, as it assigns extreme values to these otherwise-infinite parameters (and artificially shrinks or inflates other parameter estimates to compensate) in an attempt to accommodate a "relationship" that is really a tautology in disguise.

These issues also affected the GLMM-based analyses we adopted for Figure 6, but in a slightly less direct way. For each of these analyses (i.e., each panel of Fig. 6), we performed a likelihood-ratio test comparing two different GLMMs: (1) a "full" model we fit to explain participants' success on held-out questions given their estimated knowledge at those questions' embedding coordinates, with random effects for participants and questions, and (2) a "null" model we fit to explain the same data given only the full model's random effects. Formally, these tests assess whether the likelihood of our observed data is significantly greater under the full model than under the null model—in other words, whether a model that considers participants' estimated knowledge explains their success on held-out questions significantly better than one that does not (or equivalently, whether knowledge estimates provide significant explanatory power). However, because the models we're comparing differ by only a single fixed effect, these tests are also (asymptotically) equivalent to testing whether the relationship between estimated knowledge and (the log-odds of) success on held-out questions is significantly different from *zero*—which does not necessarily reflect the expected relationship under our null hypothesis. This is because our estimates of participants' knowledge for held-out questions are simply weighted versions of their remaining-proportion-correct scores, where the weight we assign to each held-in question reflects its embedding-space distance from the held-out question (with smaller distances corresponding to larger weights).

Our core assumption in constructing these estimates—which we are testing in these analyses—is that a participant's ability answer a particular question explains more about their ability to answer other questions at nearby embedding coordinates than about their ability to answer questions at far-away coordinates (i.e., that knowledge of the concepts tested by these questions is "smooth" with respect to distance in embedding space). If this is *not* the case, as we assume under our null hypothesis (i.e., if the between-question distances we leverage as weights are randomly distributed with respect to

participants' correct vs. incorrect responses), then on average, our estimates of participants' knowledge will approach their *unweighted* remaining-proportion-correct scores and exhibit the same inverse relationship with their success on held-out questions. Note that this applies specifically to the "All questions" and "Within-lecture" versions of these analyses (top row and middle two rows of Fig. 6), wherein we estimate knowledge for each held-out question using the remaining questions from the *same* pool as the one we held out (i.e., all remaining questions from the same quiz, or all remaining questions about the same lecture, from the same quiz). By contrast, in the "Across-lecture" analyses (Fig. 6, bottom two rows), we estimate knowledge for each question about one lecture using only questions about the *other* lecture. In this case, "holding out" a particular question does *not* alter a participant's proportion-correct score on the questions we use to estimate their knowledge.

The above means that our significance tests for the "All questions" and "Within-lecture" analyses were overly conservative, because they assumed a "more positive" null relationship (i.e., zero) than we would actually expect given truly uninformative knowledge estimates. Any predictive power our knowledge estimates *do* afford would therefore have to surpass a relatively high threshold to overcome this "negative baseline" and yield an apparent positive relationship. Although assessing our results against the true "negative null" (as the reviewer suggests in comment 1b) *would* in theory have been a less conservative test, it was unfortunately not clear to us how such a comparison could be operationalized since this null relationship (between unweighted remaining-proportion-correct scores and success on held-out questions) is not a "real" statistical relationship we can represent numerically in order to test hypotheses against. At the same time, we felt that Reviewer 3's proposed GLMM-based approach *was* worth adopting for these analyses, as we agreed with their assessment that it accounted for important effects in our data that our prior *U*-tests had not (e.g., baseline differences in participants' performance and questions' difficulties). In weighing these factors, we ultimately concluded that a highly conservative test would be preferable to a potentially anti-conservative one, and opted to include these analyses in our previous submission along with a brief explanation of why an apparent negative relationship actually reflects a null result. However, we acknowledge this was not an ideal solution. As we describe in response to comment 1b, in our newly revised manuscript, we have now devised a correction for this conservative bias that does not require explicitly modeling this negative null relationship and instead addresses the underlying problem that led to it by ensuring that participants' remaining-proportion-correct scores do *not* vary with their success on held-out questions (as was already the case in the "Across-lecture" analyses). We have also added text to our revised *Methods* section describing this "negative relationship" issue and motivating our approach to correcting for it (pp. 36–38).

Finally, we note that despite having now corrected for this issue, we have still elected not to adopt the additional GLMM-based analysis Reviewer 3 had proposed, as we feel there are remaining problems with its formulation. For this analysis, Reviewer 3 had requested we use a likelihood-ratio test to compare two models (both fit with the appropriate random effects): one with fixed effects for both remaining-proportion-correct *and* estimated knowledge, and one with a fixed effect for remaining-proportion-correct but *not* for estimated knowledge. The goal of this proposed analysis was to "provide a direct evaluation of the predictivity of 'estimated knowledge' against a baseline"—i.e., to

isolate the predictive information uniquely contributed by our knowledge estimates from that already afforded by traditional proportion-correct scores. We agree that this is an appropriate baseline against which to evaluate our knowledge estimates, but do not believe that including these scores as a separate predictor is the right way to accomplish this. Rather, a likelihood-ratio test comparing these two models would assess whether considering participants' estimated knowledge *in addition to* their proportion-correct scores better explains their success on held-out questions than considering their proportion-correct scores alone. In other words, this analysis treats estimated knowledge as a measure intended to *supplement* traditional proportion-correct scores, whereas we propose estimating a learner's knowledge as an *alternative* to computing their simple proportion-correct score. Since knowledge estimates are simply weighted proportion-correct scores, these two measures contain partially redundant "information" (and in fact are perfectly collinear under our null hypothesis). Instead, the aspect of our knowledge estimates that *can* be thought of as "supplementing" traditional proportion-correct scores is the particular *weights* we use in constructing them. Thus an analysis that achieves the intended goal of this proposed analysis would be one that isolates the predictive information uniquely contributed by these weights from that of the unweighted proportion-correct scores we apply them to. Since our knowledge estimates reduce to unweighted proportion-correct scores under our null hypothesis, and the bias correction we now employ ensures these scores do not vary with a participant's success on held-out questions, this is exactly what is now accomplished by the likelihood-ratio tests we report in Figure 6. In fact, if we were to apply the same bias correction to this proposed analyses, the estimated effect of remaining-proportion-correct would be exactly zero, since there would be no within-participant variation for it to explain, and its between-participant variation would be the same information already captured by the model's per-participant random intercepts—making this analysis mathematically equivalent to the analyses shown in Figure 6. In our view, this reflects what information an individual's proportion-correct score fundamentally *can* provide about their ability to answer a particular quiz question (that is, none beyond their baseline probability of correctly answering any question), and why the question-specific knowledge estimates we compute therefore yield "higher-resolution" insights.

b. In any event, the situation in which the null result is a negative relationship causes confusions and complications (regarding the results displayed in Fig. 6). If the null result is negative (and not zero), then shouldn't the significance of the results be assessed relative to that negative value (rather than zero)? For example, might there be some observed relationship that may not itself be significantly positive, but still different from the negative null? If so, and if I understand the author's analysis correctly, then might they not be using a significance test that is actually too conservative? This might be a situation in which generating a null distribution through permutation testing might be the best solution; then the significance of the observed relationship can be compared to the null distribution to determine significance.

In brief, the reviewer is largely correct in their assessment. This "negative null" phenomenon did indeed render our significance tests highly conservative (specifically for the "All questions" and "Within-lecture" analyses) since these tests effectively assume a null relationship of zero, rather than the "negative" one we would expect. Unfortunately, since this negative null actually reflects a mathematical dependency rather than a "real" statistical relationship with a finite magnitude we can

quantify numerically, any typical approaches we might take to constructing a null hypothesis around it would either fail or not be statistically valid. For example, permutation testing would not work here because this “negative null” arises from the direct correspondence between a specific participant’s success on a specific held-out question and their remaining-proportion-correct score excluding that question. Any permutation of the data that breaks this correspondence would therefore no longer exhibit the negative relationship it was intended to capture. In other words, our observations are inherently not exchangeable under our null hypothesis because this negative relationship is not one that could have occurred by chance (and thus could be approximated through random permutation), but rather one that exists due to the circular dependency *within* each observation.

We also considered a number of other methods of accounting for this negative null relationship in fitting these models (e.g., adjusting for it with a global offset term, using GLMMs that incorporate Bayesian priors, residualizing estimated knowledge against remaining-proportion-correct, permuting the questions’ embedding coordinates rather than correct/incorrect labels, etc.) and ultimately came to a similar conclusion as above in each case. We therefore devised a correction procedure that instead works by *eliminating* this dependency from our data prior to fitting the models. Specifically, this correction ensures that the knowledge estimates we compute for questions a given participant answered correctly and incorrectly are based on the *same* underlying proportion of correctly answered questions, and therefore have the same expected value under our null hypothesis. In our revised manuscript, we refer to this as a “rebalancing procedure” to mirror the term used to describe analogous approaches to correcting for an analogous problem in leave-one-out cross-validation (e.g., Austin et al., 2024: <https://arxiv.org/abs/2406.01652>).

To summarize, suppose again that a participant correctly answers n out of q questions on a given quiz: their remaining-proportion-correct score will be $(n - 1)/(q - 1)$ when we hold out a correctly answered question and $n/(q - 1)$ when we hold out an incorrectly answered question. To correct for this difference, when we hold out a given question, we identify from the set of $q - 1$ remaining questions all those with the opposite “correctness” label of the held-out question (i.e., when holding out a correctly answered question, we identify all remaining incorrectly answered questions, and vice versa). We then additionally exclude each of these opposite-label questions, in turn, and estimate the participant’s knowledge for the held-out question using the remaining $q - 2$ questions. This yields a set of knowledge estimates for the held-out question that are each derived from an underlying proportion-correct score of $(n - 1)/(q - 2)$, regardless of whether the held-out question was answered correctly or incorrectly. We then average over these estimates to obtain a “rebalanced” estimate of the participant’s knowledge for the held-out question. This rebalanced estimate also has an expected value of $(n - 1)/(q - 2)$ under our null hypothesis (since it’s an average over estimates with that expected null value), but effectively “spreads” the shift in its underlying remaining-proportion-correct score equally across all opposite-label questions’ contributions.

In our revised manuscript, we use these rebalanced knowledge estimates to fit GLMMs for the “All questions” and “Within-lecture” analyses shown in Figure 6. Importantly, this means that the null-hypothesized relationship between estimated knowledge and the log-odds of success on a

held-out question is now zero (as was already the case in the “Across-lecture” analyses), and is therefore accurately reflected by the null models used in our likelihood-ratio tests. This also means that any non-zero relationship we observe in these analyses now reflects predictive information specifically afforded by the embedding-space distances we use to weight individual questions in constructing our knowledge estimates.

Finally, in developing this rebalancing procedure, we identified a separate issue with how we had constructed the “null” models used in these tests. Previously, in instances where the data supported estimating group-specific deviations in the effect of estimated knowledge (i.e., fitting a “full” model with “random slopes” for participants and/or questions), we had excluded these random slope terms from our null models in addition to the fixed effect. This is common practice when performing likelihood-ratio tests for the purpose of variable selection, where the goal is typically to determine whether considering a variable “in any capacity” improves the model’s fit to the data. However, since we are using likelihood-ratio tests for the purpose of hypothesis testing, this effectively conflated tests for two different null hypotheses: that there is no relationship between estimated knowledge and the log-odds of success on held-out questions, and that this relationship does not vary among participants and/or questions. In other words, removing these random slopes from our null models actually made our prior likelihood-ratio tests somewhat *anti-conservative*, since an observed significant result could potentially reflect explanatory power contributed by multiple different terms. This was far overshadowed by the more substantial conservative bias in the “All questions” and “Within-lecture” analyses, but led to false-positive results in the “Across-lecture” analyses for Quiz 1. In our revised manuscript, we have now remedied this issue by retaining these random slope terms in our null models when they are also present in the corresponding full models.

2. *The authors discuss their LDA embedding space in relation to more complex (BERT) and simpler (word-matching) models. I have (a) a comment on manuscript organization, and (b) a question about what this means in practice.*

a. It seems strange to take up so much space with the comparisons between LDA and BERT while the BERT results are relegated to the supplementary materials. As a style suggestion, it would seem cleaner/more efficient to either move some of these discussions to the supplementary material alongside those results, or to condense the current material and keep in the discussion.

We appreciate this suggestion. We agree it feels odd to devote a large portion of our *Discussion* section to these comparisons, since (while interesting and useful) they are somewhat tangential to our main results and overall narrative. At the same time, given that both of these comparisons (between LDA and BERT, and LDA and simple word-matching) were motivated by similar questions from multiple reviewers, we suspect that some discussion of their implications will also be of interest to readers more broadly. We have therefore elected to move the bulk of text concerning these comparisons to our *Supplementary Information* file while retaining a higher-level overview of what we see as their most relevant implications in our *Discussion*. In our supplement, we have added and expanded on the text related to lower-level comparisons between specific elements of these figures in a new section titled *Supplementary results* (beginning on p. 22 of our *Supplementary Information* file).

b. The authors claim that their LDA embedding space is a “sweet spot” in between LLMs like BERT and simpler word-matching models. How specific is this to the particular lectures and quizzes used in the current study? Might e.g. LLMs work better than LDA for other researchers wanting to use these methods? If this is the case, then how should other researchers using their model determine which kind of embedding space to use? What kind of test can be run to determine the correct level of granularity in the embedding space? The proposed model will not be that useful of a tool if it takes thorough testing to determine the appropriate kind of embedding space before any actual knowledge analysis is done.

The reviewer poses a number of interesting questions here. We will attempt to address each in turn.

The authors claim that their LDA embedding space is a “sweet spot” in between LLMs like BERT and simpler word-matching models. How specific is this to the particular lectures and quizzes used in the current study?

To clarify, it is not our intention to argue that LDA is universally “better” or “worse” than BERT or other large language models, nor do we intend our manuscript to make any claims or comparisons regarding the general usefulness of different model algorithms or architectures. Rather, our point is that if and when our goal is to obtain text embeddings that distinguish between highly similar content within a relatively narrow conceptual domain (e.g., nearby moments in a brief physics lecture, as in our current paper), we suggest that it is beneficial to use a text embedding model trained specifically on the content we want to be able to “explain,” rather than one that has been trained to explain a far broader and more diverse range of content (like BERT and other LLMs).

This reflects a fundamental tradeoff in constructing text embedding models (and language models more broadly), between “generality” and “specificity.” One key factor that determines where a particular model falls along the spectrum of this tradeoff is the breadth of conceptual content included in its training corpus. Essentially, the enormous and highly diverse text corpora used to train modern LLMs (in the case of BERT, roughly 3.3 billion words across thousands of books and all of English Wikipedia) enables them to represent content from a vast array of conceptual domains in a way that models trained on smaller, domain-specific corpora (like our LDA model) fundamentally cannot. However, since these models’ embedding spaces must be able to represent any content within the enormous scope of their training materials, this necessarily limits the level of specificity (or “resolution”) with which they can characterize subtle differences between content within a more narrow scope. In other words, while a model trained exclusively on content from a given domain can dedicate its entire feature space to capturing subtle similarities and distinctions that are meaningful within that domain (at the cost of generalizing poorly beyond that domain), a general-purpose model must apportion its representational capacity across a much broader conceptual landscape.

One way to think about this is to substitute the semantic features we want to capture using text embeddings for physical features we might instead want to capture through photographs. Suppose that rather than relating and distinguishing lectures and quiz questions based on their conceptual (semantic) features, we wanted to (either qualitatively or algorithmically) relate and distinguish different individuals based on their facial features. If we used photographs containing only those

individuals' faces, we might be able to do this fairly well. But if we were to "zoom out" too far when taking these pictures, to the point where the vast majority of their "space" was devoted to representing content other than the subject's face (e.g., the surrounding environment), and the subject's face was represented using only a small number of pixels, then the particular features we care about capturing would be blurred and lost. We might still be able to detect that each image contains *a* face, but we would fail to capture the more subtle and specific features that identify and distinguish *different* faces.

This is what we see as the key take-away from the comparisons between our LDA model, which we trained exclusively on the lectures used in our study, and BERT, whose embedding space must "span" its far larger and more diverse training corpus (Supp. Fig. 10). The embeddings afforded by our LDA model capture the lecture and question content at a "resolution" that enables us to distinguish between questions about broadly similar but subtly (and meaningfully) different content areas, and to "match" individual questions to specific sections of lecture content. By contrast, BERT assigns both lectures and all quiz questions to the same tiny region of its embedding space, indicating that they are all broadly "about" the same domain of physics concepts, but blurring the more subtle distinctions between the different concepts they cover within that domain.

To extend this facial-feature analogy, we could also imagine taking pictures that are "zoomed *in*" too far, to the point where each image captures only a small patch of skin or hair. In this case, we would similarly fail to capture the features relevant to our goal, as we would be considering only fragments of them—for example, a given patch of skin might be part of a nose, ear, chin, or something else, but we lack the surrounding context needed to determine this. This is what we see as the key take-away from our comparisons to simple word-matching "models" (Supp. Fig. 11). If a given question contains the word "force," that question could be asking about concepts related to the gravitational force, electromagnetic force, strong or weak nuclear forces, or something else. But the distinction between these possibilities is blurred and lost because the abstract concepts discussed in the lectures and referenced by the quiz questions are not defined or distinguishable at the level of individual words.

This is the "sweet spot" notion we describe in our manuscript, between simpler and more complex methods we could use to capture and relate the lectures' and questions' contents. While larger, more complex LLMs "take too *broad* a view" in characterizing their semantic contents, simpler word-matching approaches "take too *narrow* a view," and as a result, both alternatives fail to capture conceptual similarities and differences at a level of granularity that would allow us to meaningfully estimate knowledge for the different concepts covered within the space of a single course lecture. But by training a model specifically on the course materials for which we want to estimate participants' knowledge, we are essentially defining our "level of zoom" directly based on the content of interest itself, enabling the model to devote the full extent of its embedding space to characterizing subtleties within that content, so that the relevant conceptual features "come into focus."

In this way, the reviewer's question of whether this "sweet spot" is specific to the particular course materials used in the current study has two answers. Given that we trained our text embedding model on the specific lectures our study's participants viewed and answered questions about, the precise embedding space the model identified from that content is, naturally, optimized for characterizing that

content, and we would not expect it to be equally useful for characterizing other very different content. However, the ability to identify an analogously useful embedding space defined by the contents of an arbitrary set of course lectures is a central component of our modeling framework. In other words, the distinction we are drawing here is that the “sweet spot” we refer to in our manuscript is not the exact embedding space our model identified for *these* course lectures, but rather the *approach* we take to defining an embedding space for any given set of lectures (i.e., training the model on overlapping sliding windows of their transcripts). To reiterate from our previous response letter, we by no means believe LDA is the *only* model capable of identifying this sort of “sweet spot” embedding space—in fact, if a different “LDA-like” model were found to characterize the lectures’ and questions’ contents with similar (or possibly even greater) fidelity, that model could easily be “dropped in” in lieu of LDA without changing any of the core assumptions of the broader framework we are proposing. However, we suggest that an important characteristic of any alternative model that *could* be comparably well-suited for this task (i.e., what we mean by “LDA-like” in this context) is that it can be adequately trained on the specific course materials at hand.

We have refined the language in our updated *Discussion* section to more clearly frame the comparisons between our LDA model and BERT in terms of the conceptual breadth of their training corpora (as opposed to these specific models themselves), and to more clearly present this “sweet spot” idea in terms of the different “semantic scales” at which our approach and other alternatives characterize the lectures’ and quiz questions’ contents (pp. 24–26).

Might e.g. LLMs work better than LDA for other researchers wanting to use these methods? If this is the case, then how should other researchers using their model determine which kind of embedding space to use?

Following from our response above, we do not believe that LLMs are likely to “work better” for other researchers interested in adopting our framework, given the misalignment between their “generalist” embedding spaces and the level of specificity needed to successfully identify and distinguish concepts on the scale at which our framework applies (i.e., within individual course lectures).

That said, a common way of improving generalist LLMs’ performance on domain-specific tasks is to “fine-tune” them on a domain-specific text corpus. In our previous response letter, we had speculated that a fine-tuned LLM might capture nuances in the lecture and question content comparably well to—or perhaps even better than—our LDA model. In the interest of answering the reviewer’s question here, we decided to explore this possibility further, and in doing so realized there are some additional reasons why LLMs would likely be poorly suited for our particular application that wouldn’t be solved through fine-tuning. We feel strongly that that a discussion of these more technical considerations around different model architectures, fine-tuning techniques, linguistic structures, and so on is outside the scope of our current paper and would distract from our main focus of presenting our more general knowledge estimation framework (similarly to how the extended comparisons with BERT felt out of place in our *Discussion*). However, since the reviewer is specifically asking about the possibility of incorporating LLMs into our framework, we do think these considerations bear mentioning in response. To help illustrate, we fine-tuned BERT using the same text corpus to which

we fit our LDA model (i.e., sliding windows of the two lectures’ transcripts) and reassessed its ability to yield “useful” embeddings in the context of our framework (i.e., embeddings that enable us to distinguish lectures and questions about different content areas, and “match” individual questions to specific sections of lecture content). The plots shown below are analogous to those displayed for our LDA model and “base” (i.e., not fine-tuned) BERT in Supplementary Figure 10:

In summary, while fine-tuning leads to a marginal improvement compared to “base” BERT, the model’s performance still falls far short of our domain-specific LDA model’s. For example, the lectures and questions related to different content areas are now represented more distinctly than they were prior to fine-tuning (e.g., the range of correlation values in the heatmap above is ~16 times greater than that shown in Supp. Fig. 10), but we still fail to recover the temporally specific “matches” between questions and periods of lecture content afforded by the LDA embeddings (i.e., compare the correlation time series plots above to Figs. 4A & 4B). This highlights an important difference between fine-tuning a generalist LLM and training a domain-specific model from scratch: even after fine-tuning, LLMs continue to allocate a portion of their representational capacities to content outside the tuning corpus. (This is very much by design, with failure to do so termed “catastrophic forgetting.”) This implies that even for two hypothetical models with identical architectures, parameters, inference procedures, and so on, one that is pre-trained on a domain-general corpus and

then fine-tuned to a domain-specific corpus will necessarily afford lower “specificity” for the semantic (conceptual) content within that domain than one that was trained on that domain from scratch.

Despite this, in many cases this retention of domain-general “knowledge” after fine-tuning is a desirable behavior, and in fact a notable advantage to using a fine-tuned generalist model over a domain-specific model. For example, it can greatly benefit performance when the domain of content the model will need to “handle” is either too large to be fully included in a training corpus, or not fully known or available at the time of training (e.g., if our goal was to characterize scientific papers uploaded to arXiv, it would be intractable to train a model on *every* paper ever uploaded, and impossible to train it on *future* uploads). But this benefit doesn’t apply in the context of our framework since the content we (or other researchers and practitioners) might want to characterize (i.e., specific course materials for which we want to estimate learners’ knowledge) will necessarily be known and available in advance. And far from being prohibitively time- or resource-intensive, training a domain-specific model on course materials comparable to those used in our study would be fast and easy (e.g., training our LDA model on the lecture transcripts takes less than one second—roughly 1/1000th the time it took to fine-tune BERT to the same data).

Another reason this behavior is often beneficial is that it enables LLMs to retain representations of complex linguistic features (e.g., syntactic features) that are difficult to learn from smaller, domain-specific corpora alone. However, in our particular case, this actually ends up being a further detriment. Transformer-based LLMs are almost universally trained on written text, which tends to follow much more rigid grammatical and syntactic conventions than natural speech. Through their architectures and training paradigms, LLMs are designed to leverage this structure in characterizing a given text sample’s meaning. But in our case, the text samples we want to characterize comprise transcripts of content that was originally spoken (i.e., course lectures), and therefore exhibit different linguistic structures than LLMs are optimized to leverage. For example, the transcripts contain various disfluencies, repetitions, and informalities that rarely occur in written text, while simultaneously lacking many of the grammatical elements that usually provide important “guideposts” for transformer models’ attention mechanisms, such as function words, discourse markers, and sentence boundaries (e.g., punctuation). To provide a concrete example, here is an excerpt of the first ~25 seconds of the first lecture video used in our study:

“what I want to do in this video is give a very high-level overview of the four fundamental forces four fundamental forces of the universe and I’m going to start with gravity I’m going to start with gravity and it might surprise some of you that gravity is actually the weakest of the four fundamental forces that’s surprising because you say wow that’s what keeps us glued not glued but it keeps us from jumping off the planet”

While it might be possible to manually edit the lecture transcripts to exhibit a structure more typical of written text, we view our ability to leverage automatically generated, freely available transcripts (from the YouTube API) as an important part of our framework’s potential for scaling up to larger and more diverse sets of course lectures, and being useful to other researchers without requiring significant

upfront effort or investment. Even if the transcripts themselves *were* structured more like typical written text, our approach to constructing “trajectories” representing the lectures’ dynamic contents (by projecting sliding windows of their transcripts into text embedding space) means that the individual text samples (i.e., sliding windows) the model “sees” often don’t correspond to cleanly delineated sentences. Meanwhile, the quiz questions—which our goal is to “match” to the lecture content to facilitate our knowledge estimates—*did* originally appear in written form, and therefore exhibit linguistic structures that differ from the lecture transcripts, and are closer to what transformer models “expect to see.” As we note in our *Supplementary results* section, this likely poses further challenges to “matching” the questions to specific lecture content based on their embedding weights: since BERT’s representations of the *semantic* features that relate and distinguish content within this domain are highly homogenous, much of the variation in its embeddings instead reflects the lectures’ and questions’ *syntactic* differences (e.g., see the 3D PCA projections above and in Supp. Fig. 10).

Despite being an arguably “less powerful” model than modern LLMs, LDA is actually fairly robust to these challenges that the lecture transcripts’ and quiz questions’ linguistic structures pose for transformer models. Like many comparatively simple text embedding models, LDA makes the simplifying assumption that the text samples it encounters are equivalent to “bags of words.” In other words, it ignores word order and syntax entirely in characterizing a text sample’s semantic meaning and instead considers only individual words’ frequencies and co-occurrence patterns. It also naturally emphasizes “content-bearing” words (e.g., nouns, verbs, adjectives, etc.) in its representations while downplaying the influence of “low-information” words like disfluencies and discourse markers (in fact, these “stop words” are typically discarded altogether during preprocessing). This ends up making models like LDA actually quite *well*-suited to capturing conceptual similarities across modalities (e.g., between written content and transcripts of spoken content) since most of the linguistic features that tend to differ between modalities are ones that these models naturally disregard or de-emphasize. However, we want to emphasize that while we think these comparisons are interesting and relevant in the context of the reviewer’s particular question here, it is not our intention to make claims, provide commentary, or express preferences related to different model architectures in our manuscript. Importantly, we view these considerations around applying transformer models versus “bag-of-words” models to natural speech as a separate and more specific set of considerations than the general “sweet spot” notion we describe in our *Discussion*, since lecture transcripts are by no means the *only* form of content to which our framework could theoretically be applied (e.g., one could easily adapt our approach to estimate learners’ knowledge for concepts expressed in written media like books or scientific papers). By contrast, the idea that training a text embedding model on the specific content under consideration constitutes a “sweet spot” in terms of the “granularity” of its conceptual representations would hold true for any potential application of our framework.

We think using BERT as our example of a “generalist” model that “zooms out too far” is useful given its popularity as an off-the-shelf solution for generating text embeddings, particularly in behavioral research contexts. And seeing as multiple reviewers asked about the possibility of using LLMs in our framework, we suspect this example will be a useful one for readers as well. However, given the additional considerations we’ve touched on in the context of answering the reviewer’s specific

question, we feel it merits further demonstrating here that this example is a reasonable one for the point we use it to illustrate in our *Discussion*—i.e., that the lack of “within-domain specificity” afforded by generalist models does indeed reflect the conceptual breadth of their training corpora, rather than something specific to “LDA versus BERT” or “bag-of-words models versus transformer models.” To this end, below we show an analogous set of plots generated using an LDA model we trained on a random sample of 100,000 English Wikipedia articles (a small but computationally tractable fraction of BERT’s training corpus). Of note, this generalist LDA model’s embeddings of our study’s course materials exhibit patterns similar to those we highlight as meaningful in BERT’s embeddings—namely, a reduced range of within- and across-content area correlations (with most being very high and all significant), diminished specificity in matching questions to periods of lecture content, and question embeddings that are less “near to” (i.e., often fall outside the convex hull of) the lecture trajectories. With sufficient time and computing resources to train an LDA model on a corpus even closer to the size and scope of BERT’s, we might expect these patterns to converge even more closely.

What kind of test can be run to determine the correct level of granularity in the embedding space?

The idea of a specific “test” to determine the optimal level of granularity for the embedding space is an interesting one, and touches on a complex aspect of constructing text embedding models.

As described above, one key factor in determining the granularity of a model's embedding space is the conceptual breadth of content included in its training corpus. For the particular role the text embedding model plays in our framework, the optimal breadth of training content is that which allows the model to devote its full representational capacity to capturing conceptual similarities and distinctions *within* the specific content for which we want to estimate learners' knowledge, without "wasting" any of its capacity on representing other content *outside* the domain of interest. This is achieved by training the model solely on the specific course materials at hand.

The other key factor is the number of features (in the case of LDA, "topics") the model is able to "use" to represent the content we want to embed (i.e., the dimensionality of the embedding space). Identifying an optimal value for this hyperparameter is comparatively less straightforward. Over the past few decades, there have been many attempts to define standardized, quantitative metrics for selecting an appropriate number of topics to characterize a given text corpus, with historically popular choices including held-out document perplexity, normalized pointwise mutual information, and various formalizations of "topic coherence." We discussed and even implemented some of these measures in response to a comment from Reviewer 3 on our initial submission. However, there is growing consensus within this field that such automated evaluation metrics for topic models are fraught, and optimizing a model's embedding space based on them tends to yield representations that are less useful in practice (e.g., Chang et al., 2009; Bhatia et al., 2017; Lipton, 2018; Doogan & Buntine, 2021; Hoyle et al., 2021; Stambach et al., 2023). We tend to share the perspective described by Boyd-Graber et al. in their 2014 book chapter entitled "Care and feeding of topic models," which emphasizes performing more holistic evaluations designed to assess the model's usefulness for the specific task at hand. In our case, a "useful" model is one whose embeddings enable us to (A) distinguish between lectures and quiz questions about different content areas, and (B) match individual questions to relevant, temporally specific sections of lecture content, so that we can meaningfully infer learners' knowledge of that content from their quiz responses.

In this way, the best "test" of whether a given model's embedding space is appropriately "granular" for use in our framework is the sorts of diagnostic visualizations we show above and in Supplementary Figures 10 and 11. Indeed, this is the exact purpose we intended those figures to serve. In Supplementary Figure 10, we show how our LDA model's embeddings of the lecture and question content differ from those obtained from a model whose embedding space is *not* sufficiently granular (i.e., BERT) in order to highlight how the latter model's representations fail to meet the criteria for "usefulness" in our framework. In Supplementary Figure 11, we show how our LDA model's embeddings differ from semantic representations that are *too* granular in order to highlight similar shortcomings. (While we didn't explicitly fit a second model for Supp. Fig. 11, the Jaccard index of two text samples is equivalent to their similarity in an embedding space where every unique word is a separate binary feature.)

In our new *Supplementary results* section, we discuss the comparisons shown in these figures in greater detail, with a focus on what their specific visual elements indicate about a model's usefulness in the context of our framework. We hope this can serve as a high-level guide that (along with the

publicly available code we provide for fitting our models and generating these figures) can help researchers interested in adopting our approach apply and interpret these holistic “tests” themselves. As with virtually all natural language processing tasks, our text embeddings benefit from some degree of manual oversight and hand-tuning. But fortunately, the speed with which an LDA model can be fit, evaluated, tweaked, and re-fit to tune performance generally makes this process fast and easy.

3. In their explanation of their Fig. 6 results, the authors suggest that one possibility is that participants forgot some material: “If this forgetting happens in a relatively “random” way (with respect to spatial distance within the embedding space), then it could explain why some held-out questions about *Four Fundamental Forces* were answered incorrectly, even if questions at nearby coordinates (i.e., about similar content) were answered correctly.” (p. 18).

This seems to contradict their theory/finding regarding the “smoothness” of knowledge space. If forgetting happens randomly relative to the spatial location of knowledge, this evokes a kind of “Swiss cheese” knowledge space with holes randomly scattered about—in this case, knowledge would NOT be smooth, as the authors seem to claim. I can’t reconcile these two claims. Perhaps the smoothness visualized in Fig. 7 is a function of averaging/combining across participants, but then making the general smoothness claim would not be so warranted.

We appreciate the reviewer pointing out this inconsistency. We agree that the speculative explanation we had offered for this result (that within-lecture knowledge estimates do *not* predict success on Quiz 3 questions about *Four Fundamental Forces*) didn’t fit with our broader thesis about the “smoothness” of knowledge. While our interpretations of the results shown in Figure 6 have broadly changed (and, in our opinion, become more intuitive) now that we have addressed the issues described in our responses to comments 1a and 1b above, it’s worth noting that in every version of these analyses we have run to date (*U*-tests, conservatively biased likelihood-ratio tests, and now bias-corrected likelihood-ratio tests), these particular knowledge estimates have been the only ones we compute for Quiz 3 to *not* predict participants’ success on held-out questions. The consistency in this pattern of results across multiple changes to our analysis approach would seem to suggest that while estimated knowledge *does* in general predict success on Quiz 3 questions, there is some genuine effect in participants’ performance on the particular subset of questions used in this analysis that impedes our ability to distinguish between correctly and incorrectly answered held-out questions.

We did some deeper digging into the data for this analysis to try to determine what this effect might be. To summarize, we believe this consistent null result reflects a particular ceiling effect in participants’ quiz performance (rather than some broader aspect of their behaviors or knowledge). We describe this on pages 16–17 of our revised manuscript:

Speculatively, the Quiz 3 results suggest that the within-lecture knowledge estimates may be susceptible to ceiling effects in participants’ quiz performance. On Quiz 3, after viewing both lectures, no participant answered more than three *Four Fundamental Forces*-related questions incorrectly, and all but five participants (out of 50) answered two or fewer incorrectly. (This was the only subset of questions about either lecture, across all three quizzes, for which this was true.) Consequently, for 90%

of participants, our within-lecture estimates of their knowledge for *Four Fundamental Forces*-related questions that they answered incorrectly leveraged information from at most a single other question they were *not* able to correctly answer. This likely hampered our ability to accurately characterize the specific (and by the time they took Quiz 3, relatively few) aspects of the lecture content these participants did *not* know about, and successfully distinguish them from the far more numerous aspects of the lecture content they now *did* know about.

Essentially, as participants acquire more knowledge over the course of our experiment, a few things tend to happen that have opposing effects on our ability to predict their success on held-out questions.

On one hand, with each successive quiz, the patterns in participants' correct and incorrect responses will tend to become more faithful reflections of their "true" knowledge. Intuitively, this occurs because after participants view each lecture, their correct responses are by and large more likely to reflect "real" knowledge of the lecture content rather than spurious "noise" due to successful guessing based on vague intuitions. Additionally, as we describe on pages 18–19 of our revised manuscript, we suspect that after viewing the lectures, participants' internal representations of their conceptual contents may become more closely aligned with the conceptual representations learned by our text embedding model, since the model was explicitly trained on those lectures' transcripts. Together, if participants' quiz responses come to better reflect their knowledge of the concepts being tested, and the relationships between the quiz questions' embeddings come to better reflect how participants organize their internal representations (and/or knowledge) of those concepts, then our ability to predict success using estimated knowledge should increase with each quiz and be greatest on Quiz 3. Accordingly, one high-level pattern that has persisted across all versions of these analyses (irrespective of various biases that had nudged some results above or below the " $p = 0.05$ " threshold in prior versions) is that the predictive strength of estimated knowledge tends to increase over successive quizzes (e.g., compare odds ratios & λ_{LR} values between columns in each row of Fig. 6). This would also explain why our majority-significant results for Quiz 3 have been robust to these fairly substantial changes to our analysis approach.

On the other hand, as a participant gains more knowledge and answers more quiz questions correctly, our estimates of their knowledge become increasingly "saturated" (i.e., relatively high for all content) and the "signal" distinguishing content they do versus do not know about becomes somewhat weaker. This occurs because when a participant's quiz responses contain fewer incorrectly answered questions, those responses provide more limited information about content that participant does *not* know about. The information contributed by these "negative samples" of a participant's knowledge (i.e., questions they answered incorrectly) plays an important role in our knowledge estimates: while correct responses help identify underlying conceptual themes that are *common* among content a participant knows about, incorrect responses help determine which of those themes are *specific* to content they know about, by identifying conceptual themes that are also represented in content they do *not* know about. Within our framework, themes that are "common" among sets of quiz questions appear as similar patterns in those questions' topic activations when we transform them using our topic model.

Themes that are relatively “specific” to a particular set of questions then appear as patterns of topic activations that are more similar to each other than they are to those topics’ activations for other questions. When we estimate a participant’s knowledge for a given held-out question, that estimate gets “pulled” away from their unweighted remaining-proportion-correct score in the direction of their binary success on held-in questions that exhibit similar conceptual themes, with a “strength” proportionate to how specific those shared themes are (i.e., the correlation between the held-out question’s topic activations and a given held-in question’s topic activations, rescaled relative to that correlation value for other held-in questions).

That we have consistently observed significant positive relationships in four of the five analyses we perform for Quiz 3 questions indicates that, in general, our knowledge estimates *can* successfully distinguish conceptual themes that are specific to known versus unknown content even when relatively few “negative samples” are available because participants’ quiz performance is near ceiling. However, the nature of the “Within-lecture” analyses further compounds the challenges posed by this ceiling effect: since all questions we consider in these analyses pertain to content from the same lecture, they will tend to exhibit largely similar sets of conceptual themes overall (i.e., all questions’ topic activations will tend to be highly correlated), and the particular themes that tend to differ between correctly and incorrectly answered questions will present as relatively subtle differences between those questions’ overall patterns of topic activations. Our results suggest that for *Birth of Stars*-related questions on Quiz 3, we are still able to leverage these subtle differences to distinguish between known and unknown content. But for *Four Fundamental Forces*-related questions, for which our knowledge estimates could rarely leverage more than a single incorrectly answered question, we lacked sufficient information to reliably characterize these subtle, specific differences. We note that this is not a problem for our knowledge estimates per se, but rather it is a problem specific to *validating* our knowledge estimates. In other words, if a learner demonstrates perfect performance on every question, our framework will predict that they will also likely perform well on *other* questions (which seems to us a reasonable generalization). But since our testing framework requires us to show a contrast between correctly versus incorrectly answered questions, ceiling performers appear to be poorly predicted (since they have very little contrast between what they know most versus least).

Editorial requests

Please include a sentence on the theoretical motivation for this work at the beginning of the abstract.

We have added a sentence on the theoretical motivation for our work to the beginning of our abstract.

Please provide a point-by-point response to the comments by Reviewer 4. In addition, in line with Reviewer 4's comments, we ask that you include a deeper discussion of the limitations and tone down claims regarding the practical application of these findings.

We have provided a point-by-point response to Reviewer 4's comments below. The reviewer's comments are italicized, and our responses are in bold. In response to the reviewer's comments, we have added language to our *Discussion* section to better acknowledge the limitations of our framework's practical application in its current form, and highlight what would be required to translate our findings into the sorts of meaningful feedback or interventions the reviewer describes.

Please include a 'Supplementary Information' header at the start of the supplements document.

We have added a "Supplementary Information" header to the beginning of the document.

Thank you for generating a Zenodo DOI URL for your Github page and including this in the data and code availability statements. Currently you still list both the regular Github URL and the DOI URL in the data and code availability statements. Please remove the regular github URL, the DOI URL is sufficient.

If it is acceptable, we would prefer to include both the DOI URL and GitHub URL in our data and code availability statements (similarly to other recently published papers on *Nature Communications's* website). However, if this is not allowable, we consent to the Data Availability statement being edited to read:

All of the data analyzed in this manuscript are archived on Zenodo at <https://doi.org/10.5281/zenodo.17783435>. All Khan Academy content is available for free at <https://www.khanacademy.org>.

and to the Code Availability statement being edited to read:

All of the code for running our experiment and carrying out the analyses is archived on Zenodo at <https://doi.org/10.5281/zenodo.17783435>.

Please ensure the references are in the standard Nature format and follow the sequence: author list, title of paper, name of journal, volume number, initial-final page numbers or article number (year).

We have converted all references to the standard Nature format.

Please provide us with more information on the creation of fig 1 and fig 8c. Were these images and every element of these images created by you and/or your co-authors? Were they created using any previously created elements? If

the images are from a database please supply proof of permission for their use (receipt, express permission from creator, confirmation of compatible Open Access licence or Public Domain).

Figure 1 includes a stock asset (the computer monitors) from Adobe Stock, obtained by the authors under Dartmouth College's Education/Institutional Adobe Stock license. We have included the License Record for this Stock asset at the end of this file, and the license terms are available at <https://www.adobe.com/cc-shared/assets/pdf/legal/servicetou/stock-product-specific-terms-en-us-20250808.pdf>. We note that this asset is not considered an "Editorial Work" and therefore does not require attribution. All other elements of Figure 1 were created from scratch by the authors.

Figure 8c was created solely by the authors and does not include any previously created elements. The code we wrote to generate Figure 8c may be found at https://github.com/ContextLab/efficient-learning-khan/blob/master/code/notebooks/main/7_knowledge-maps.ipynb.

Please can you confirm if you have used any application or software to create the images, Please provide the permission proof or licence.

Figure 1 was created using Adobe Illustrator. Figure 8 was created using the Python programming language and arranged into panels using Adobe Illustrator. The Python programming language is freely distributed under the PSF License Agreement, available at <https://docs.python.org/3.9/license.html#psf-license-agreement-for-python-release>. Dartmouth College's Education/Institutional Adobe license is not directly available to us in a shareable form, but proof of Dartmouth's license is visible in the License Record included at the end of this file (see "Customer" in the top right).

Please remove the hyperlinks to the figures and tables from the supplementary information file and provide it in the plain text.

We have removed the hyperlinks from the Supplementary Information file.

Reviewer comments

Reviewer #4 (Remarks to the Author):

The manuscript by Fitzpatrick and colleagues describes an application of text embedding models to measure and “map” conceptual knowledge during and after learning about some topics in physics (watching videos and then taking quizzes). I did not review prior versions of this manuscript but have been asked to assess the responses of the authors to prior rounds of feedback. Given this context, below I provide my own evaluation of the manuscript (mostly focused on the responses to Reviewer 3’s concerns), but I do not specifically suggest or request any further edits to the manuscript.

First, it is clear that the authors have put a tremendous amount of work into responding to prior comments from reviewers. I think they have been about as responsive as possible and the paper is very well written. In many respects it is a beautiful paper. Reviewers 1 and 2 seem generally satisfied with the revisions. Reviewer 3, however, raised a number of concerns related to details of the statistical analyses, the specific models used, and the overall significance of the findings. My impression is that the authors have satisfactorily addressed Reviewer 3’s comments related to the statistical analyses and have appropriately contextualized the choice of model (with helpful comments about other models). However, I do share some of Reviewer 3’s concerns about the significance of the work—and these concerns are, I think, less addressable unless the scope of the paper was substantially expanded (which I think is infeasible). Thus, I have a mixed impression: the authors have done some very commendable work on this paper and there is certainly a wow factor to the methods and visualizations (which may be enough for the paper to have impact), but I do have concerns about the actual significance of the work.

Comments

1. My overarching reaction to the manuscript is that it is more impressive than it is significant. Reviewer 3 shared a similar reaction: ““the authors’ algorithm is cool, but does it sufficiently advance us beyond the capabilities that already exist in large language models, which are available to students and instructors.” Reviewer 3 was perhaps more narrowly focused on the advance of the current methods relative to other large language models or to ChatGPT, but I think the point applies more generally. I might differ from Reviewers 1 and 2, but, to my mind, the empirical results are more of a sanity check than meaningful results on their own. That is, I don’t think these results speak to a theory or debate in the field of cognitive science. Rather, as the authors acknowledged, this is more of a methods paper and I think the empirical results mostly demonstrate that the method is doing what it “should do.” For example, showing that subjects know more about a topic after watching the video or that knowledge about a topic “predicts” performance for similar questions is something that most people would take at face value. Likewise, the fact that questions mapped to fairly specific time points in the lectures is something that almost needs to be true. If the questions are drawn from the lecture content (which they are), then it would be a red flag if the method couldn’t map questions to timepoints in the lecture. Thus, for each of these results, if the model failed to show these effects, it would be a reason to tweak the model, not a meaningful result. (And I think the authors would mostly agree with all of this). In fact, there are quite a lot of degrees of freedom that went into getting results that “make sense.” This is, of course, more evident in the rebuttal than in the actual manuscript.

We agree with the reviewer that the primary focus of our manuscript is methodological rather than empirical: We present a mathematical framework for characterizing learners' knowledge and how it changes with training. Given this goal, the analyses and results we present in our manuscript are primarily intended to validate our approach and demonstrate its capabilities, rather than speak to a particular theory or debate in the field of cognitive science. In our view, an important validation of any model of a complex, real-world process is its ability to recover high-level phenomena we might intuitively "know" should occur, in order to establish a credible basis for any further insights the model might afford. (E.g., if we were developing a model of large body interactions in the field of astrophysics, we might first want to confirm that it can explain a gravity-like attraction between bodies, despite that we already "know" one exists.) In our case, we show that, for example, our approach can map individual quiz questions onto specific, relevant sections of lecture content (Fig. 4; Supp. Tab. 3; Supp. Figs. 3–4) and recover increases in learners' knowledge about each lecture's content after they view it (Fig. 5). The value of these results is not that they would be unobtainable via any other method (e.g., one could qualitatively match questions to lecture content and manually score participants' quiz responses to achieve something similar), but rather in demonstrating (A) that our approach can recover and quantify these patterns *automatically*, and (B) that the latent space in which we embed the lectures and questions to characterize relationships between them captures something meaningful about their underlying conceptual contents.

Following these results, we then present insights that *wouldn't* be obtainable via traditional methods, and which depend on both (A) and (B) being true. For example, we show that we can leverage the automatically discovered conceptual relationships between quiz questions to reliably predict learners' success on specific held-out questions from their performance on specific held-in questions (Fig. 6). We also characterize how participants' knowledge "falls off" with increasing "conceptual distance" from content they know about (Fig. 7) and demonstrate how our approach can be extended to predict knowledge for any arbitrary content expressible by the text embedding model we trained (Fig. 8). In other words, while the initial set of "sanity-check" results are not intended to constitute significant empirical insights in and of themselves, they play an important role in validating the foundation we subsequently build upon to obtain more meaningful insights into learners' knowledge and how it may be characterized via our framework.

2. I think the question is whether the paper does enough, as a methods paper. (Again, I think this is where Reviewer 3's concerns were, as well). As I note above, I think the authors have done enough on the statistics/analyses front to make a compelling case that the method "works." That said, the paper does not introduce a new large language model, nor does it even use the latest models. Thus, I think the value as a methods paper comes down to the specific way in which existing models are leveraged to 'map' knowledge. Ultimately, I am not particularly convinced that there is much practical value in this method. The authors motivate the ideas in terms of teachers that might want to map what students know. Putting aside concerns about the actual feasibility of a teacher using this, the immediate question that came to my mind was: what is the advantage (gain) of this method relative to much simpler approaches. The authors compare their method against a strawman of a single test grade (e.g., 85%). The argument is that a single test grade makes it impossible to assess which specific topics or concepts a student has learned. That's true, of course. But, for most of us that teach classes, it is very common (and almost a necessity) that we structure

exams so that questions are organized around sub-topics. For example, we might deliberately include a certain number of questions per subject area. It is therefore quite trivial to have sub-scores by topic area. While I genuinely loved the visualizations that were presented in the current paper (again, it is beautiful work), I really struggled to see how or whether these visualizations provide more useful information than sub-scores on an exam. Even for the analyses that seek to predict performance on held-out questions, I was left wondering whether this method is any more sensitive than simple conditional probabilities based on sub-scores. For example, if a student answered the first 4 questions on a sub-topic correctly, what is the probability of answering question 5 correctly? What if they only answered 2 of the first 4 questions correctly? For a methods paper, I think that there is somewhat of a burden to demonstrate that the new method being presented outperforms existing methods. And, ultimately, I was not convinced that the current paper does this. Yes, the high resolution coordinate spaces are impressive and they certainly give the impression of being much more nuanced than a sub-score, but it is harder to see the actual application of that. Can higher-resolution feedback be provided to a student or teacher? And would this higher-resolution feedback make a meaningful difference to learning outcomes? In these respects, there is no clear evidence or argument that this new method is superior to much, much simpler alternatives.

The reviewer is correct that, as opposed to developing a new large language model, the goal of our work is to develop a mathematical framework for characterizing knowledge and learning. It is also true that our framework leverages (in part) an older, “simpler” text embedding model (Latent Dirichlet Allocation) rather than the “latest” LLMs. As the reviewer notes above, we contextualize this choice of model in our manuscript, supplement, and responses to previous reviewers’ comments: In brief, while state-of-the-art LLMs are useful for a wide range of applications, there are certain fundamental properties of these models (such as the enormous conceptual breadth of their training corpora) that tend to make them less suitable for our particular use compared to smaller models that can be quickly trained on the specific content for which we want to characterize learners’ knowledge (see our *Discussion* and *Supplementary Results* sections, Supp. Fig. 10, and response to comment 2b in our third response letter).

Regarding the value of our approach to predicting success on held-out questions versus doing so based on conditional probabilities derived from sub-scores: this is precisely what is tested by the “Within-lecture” analyses shown in Figure 6. Here, for each of the three quizzes, we consider only the subset of 5 questions a given participant answered about one of the two lectures viewed in our study. We hold out each of these 5 questions, in turn, and estimate the participant’s knowledge at its embedding coordinate using their performance on the remaining 4 questions. Suppose that of these 4 questions, a participant answered 2 correctly. If, instead of using our knowledge-estimation approach, we simply treated this partial score as their probability of answering the 5th (held-out) question correctly, we would obtain a probability of 0.5. Importantly, however, this probability would be 0.5 regardless of *which* 2 of the first 4 questions the participant answered correctly, or *what* the 5th question actually was. Since this probability would be the same for any potential 5th question, it would necessarily provide no signal by which to distinguish questions this individual is more or less likely to correctly answer.

The greater sensitivity of our knowledge-estimation approach comes from considering the latent conceptual themes probed by the specific questions an individual answered correctly and incorrectly, and how they reflect the conceptual themes probed by the specific question we want to predict their ability to answer. In the middle two rows of Figure 6, we show that, for 5 of the 6 tests we performed using only the subset of quiz questions about a single content area, estimated knowledge was reliably higher for correctly answered held-out questions than for incorrectly answered held-out questions. To perform each of these statistical tests, we fit a generalized linear mixed model to explain participants' success on held-out questions given their estimated knowledge for them (with random effects to account for variation among participants and questions). We then performed a likelihood-ratio test comparing this model to a null model which assumed a given participant's estimated knowledge was the same for all held-out questions—as would be the case if we instead used the simple probabilities described above. Thus we do explicitly demonstrate through statistical analyses that our approach outperforms the existing method the reviewer describes.

More broadly, our point is that a sub-score on exam questions about a particular “sub-topic” is no different from an overall score on a shorter exam with a narrower conceptual focus. Sub-scores might provide some information about “how much” different individuals know about a certain sub-topic, or their different likelihoods of correctly answering any additional question about it, but they afford no information about *what* knowledge an individual has about that sub-topic, or *which* specific questions about it they might be more or less likely to correctly answer. While our “Within-lecture” analyses define “sub-topics” at the scale of the content covered in an 8- to 10-minute lecture video, this would be true of any scale of sub-topic about which one could ask a learner 5 meaningfully different questions—a given section of a lecture might cover a sub-topic of the full lecture, which might cover a sub-topic of a larger academic unit, which might in turn represent a sub-topic of a full-length course, and so on. We would therefore disagree with the characterization of our comparison to a simple test grade (e.g., 85%) as a “strawman.” Manually defining “sub-topics” at any conceptually meaningful level would ultimately entail characterizing learners' knowledge about each of those sub-topics via an analogously impoverished proportion-correct score. And while sub-scores on more specific categorical sub-topics might provide some additional precision over sub-scores on broader sub-topics, an important advantage of our approach is that we model the relationships between to-be-learned concepts *continuously*, allowing us to characterize knowledge at an arbitrarily high level of specificity (e.g., a single point in a high-dimensional space defined by a particular conceptual domain or subdomain of interest).

3. One way to demonstrate the value of the method would be to actually run a classroom experiment showing that students and/or teachers benefitted from the high-resolution maps compared to some simpler alternative. I realize the authors would argue that this is beyond the scope of the current paper (and I agree), but I think if the scope of the current paper is simply to demonstrate the method without demonstrating the actual value of the method, then it is necessarily a less significant contribution.

We agree that such a study is beyond the scope of our current paper. However, we do think that our existing analyses, such as those we describe in response to the reviewer’s comment above, demonstrate the value of our approach over common existing measures.

4. Relatedly, Reviewer 3 raises the point that it’s not clear the current model would outperform ChatGPT in the core function of assessing conceptual knowledge. I tend to agree. I do get the authors’ points that their method can track and map learning in a way that ChatGPT isn’t capable of (I agree with this), but, to me, it really comes down to the potential value of the method. This gets back to my question of whether the high resolution maps can actually be translated to some meaningful feedback or intervention. But if the goal is simply to have a user-friendly, powerful method for assessing conceptual knowledge, I think ChatGPT is probably the more appealing option.

We certainly agree that ChatGPT is a powerful tool with a user-friendly interface. Importantly, however, the goal of our manuscript is not to develop a market-ready, consumer-facing product like ChatGPT, nor do we view our framework as an “alternative” to ChatGPT or other LLMs in its core function or purpose. Rather, our approach aims to accomplish the separate and specific goal of constructing a mathematical representation that captures an individual’s conceptual knowledge and how it changes as they learn. As the reviewer notes, ChatGPT (and other LLMs) currently do not maintain or leverage any such representation—and we would thus argue that LLMs themselves are incapable of assessing conceptual knowledge in a rigorous or meaningful way.

Both here and in comment #2, the reviewer raises questions regarding how the knowledge representations we construct might be translated into actionable feedback or interventions for learners. We agree this is an important and exciting direction for future work! Just as additional research effort was required to translate the original transformer architecture introduced by Vaswani et al. (2017; “Attention Is All You Need”) into concrete, user-friendly tools like ChatGPT, the same is true of the nascent computational framework we propose in our manuscript. However, as in many papers that present a novel modeling approach, we provide a basic demonstration of one way this could be accomplished. In Figure 8C, we show that we can recover a distribution of weights over words representing the abstract conceptual content underlying any given point on a knowledge map—e.g., a point where an individual’s estimated knowledge is particularly high or low. Here, for visualization purposes, we display these distributions as word clouds (where each word’s size denotes its relative weight). However, one could easily imagine instead using these sets of weights, along with the knowledge estimates at their corresponding coordinates, as inputs to a separate system that translates these mathematical representations of concepts and knowledge into human-digestible feedback or tangible interventions—such as a large language model. In fact, we had described this very possibility in our prior submission’s *Discussion*:

“Another application for large language models that does not require explicitly modeling the content of individual lectures or questions is to leverage these models’ abilities to generate text. For example, generative text models like ChatGPT [56] and LLaMa [69] are already being used to build a new generation of interactive tutoring systems [e.g., 46]. Unlike the approach we have taken here, these generative text model-based systems do not explicitly model what learners know, or how their

knowledge changes over time with training. One could imagine building a hybrid system that combines the best of both worlds: a large language model that can generate text, combined with a smaller model that can infer what learners know and how their knowledge changes over time. Such a hybrid system could potentially be used to build the next generation of interactive tutoring systems that are able to adapt to learners' needs in real time, and provide more nuanced feedback about what learners know and what they do not know."

In ongoing (unpublished) work, we have already begun developing such hybrid systems that combine our approach for representing a learner's knowledge with a large language model that leverages this representation to suggest additional materials to review and adaptively construct future assessments based on them. At its core, this iterative process of constructing and updating an internal representation of a learner's conceptual knowledge, and drawing on it to personalize their instruction, mirrors how a real human teacher might approach guiding a student one-on-one. As Reviewer 3 noted in their initial review, "*there is a vast literature spanning 2-3 decades on the importance of specific feedback to students.*" We think the future development and refinement of such automated systems represents an exciting opportunity to extend this sort of individualized feedback and instruction to students for whom it might not typically be available (e.g., those in online courses without access to a live teacher or tutor). In developing a formal method of representing an individual's conceptual knowledge and how it changes over time, our manuscript presents a "prototype" for an essential component of such a system that is currently absent from modern language models.

In our revised manuscript, we have expanded and refined the above quoted paragraph to further clarify the limitations on our framework's ability to provide concrete, actionable feedback to students in its current form, and highlight this as a direction for future work.

Adobe Inc.
345 Park Avenue
San Jose, CA 95110-2704
United States

License Record Information

Customer	NERCOMP – Dartmouth College
Date	12/12/25, 12:25AM
Assets	1

License Record

License				
DATE	ASSET	FILE TYPE	LICENSE	REFERENCES
12/12/25, 12:25AM	 #315608634	Vector ai/eps 8192 x 3206 px	Education License	

Total

NUMBER OF ASSETS	1
------------------	---

General Information.

The license rights to use the Stock Assets listed on this license record were granted by Adobe to the Customer that purchased the Credits pursuant to Customer's written agreement with Adobe that grants Customer the license to the Works listed in this license record ("Agreement"). "Customer" means the entity listed in the "License Record Information" section of this license record.

Subject to compliance with the terms of the Agreement, Customer may transfer all of its license rights and restrictions to a Stock Asset to its client via an enforceable written agreement between Customer and Customer's client that includes terms no less restrictive than their Agreement or the minimum terms listed below.

The license purchased for a Stock Asset may only belong to Customer or its client (not both). For clarity, if a license is transferred by Customer to Customer's client via an enforceable written agreement, (A) the client is only entitled to the specific license that Customer purchased for the specific Work; and (B) the client's agreement is with Customer and not with Adobe. Any terms that are used in this license record that are defined in the Agreement have the meaning stated in the Agreement.

Disclaimer.

This license record is provided for record-keeping purposes only. Prices do not include applicable taxes. Adobe provides the information in the License Record Information section, certain information about the specific Work (Date, Thumbnail, Contributor Name, File Type and Price) and editorial use restrictions (if any). All other information in this license record is provided by Customer (including but not limited to the License Record name, Note section and the "References" column) but any Customer-provided information does not amend, revise or replace any terms or conditions in this license record or the Agreement.

Minimum Terms.

1. Works Standard License. A Standard License means a non-exclusive, perpetual, worldwide, non-transferable, non-sublicensable license to use, reproduce, archive, modify, and display a Work in all media, up to 500,000 times as further described in section 5(A) (Standard License Restrictions) for: (A) advertising, marketing, promotional, and decoration purposes; and (B) personal or non-commercial uses.

2. Works Extended License. An Extended License means the right to use a Work in any manner permitted under a Standard License, without the 500,000 limitation stated in 5(A) (Standard License Restrictions) and, additionally, the right to distribute the Work as incorporated into any item of merchandise, template file, or other work of authorship.

3. Audio Extended License. An Audio Extended License means a non-exclusive, perpetual, worldwide, non-transferable, non-sublicensable license to (A) combine an Audio Work with video, audio, and other materials, to create an unlimited number of Projects; (B) adapt, edit, and modify the Audio Work in Projects; (C) reproduce, copy, transmit, broadcast, display, publicly perform, and otherwise distribute the Audio Work as incorporated into a Project, including in radio, television, paid-access streaming video services, paid-access on-demand video services, theatrical releases, computer software applications (including mobile applications and video games), and physical point of sale locations (such as shopping malls, point of sale systems, in-store displays, and showroom videos); and (D) use Projects for any purpose, including advertising, marketing, promotional, and commercial.

4. General Restrictions for all Licenses. Customer must not: (A) use a Stock Asset in a way that allows a third party to use, download, extract or access the Stock Asset as a stand-alone file, or in a way that exceeds the scope of this license to the Stock Asset; (B) take any action in connection with a Stock Asset that infringes the intellectual property or otherwise violates any third party rights, including without limitation the moral rights of the creator of the Stock Asset and the rights of any person who, or any person whose property, appears in the Stock Asset; (C) register, or apply to register a trademark, design mark, service mark, sound mark, or tradename that uses a Stock Asset (in whole or in part), or claim ownership rights in an attempt to prevent a third party from using a Stock Asset; (D) use the Stock Asset in a manner that is pornographic, defamatory, or that violates any applicable laws, rules, or regulations ; (E) use a Stock Asset in a manner, or in connection with a subject, that a reasonable person could consider unflattering, immoral, or controversial, taking into account the nature of the Stock Asset, examples of which could include, without limitation, ads for tobacco; adult entertainment clubs or similar venues or services; implied or stated endorsements of political parties or other opinion-based movements; or implying mental or physical impairment; (F) use a Work in an editorial manner without the credit line or attribution, placed in a way that is reasonable to the applicable use, in this format: "Contributor Name / stock.adobe.com", or as designated on the Adobe Stock website for enterprise customers; (G) remove, obscure or alter any proprietary notices associated with a Stock Asset, or give any express or implied misrepresentation that Customer or another third party is the creator or copyright holder of a Stock Asset; or (H) use a Stock Asset or any title, caption information, keywords or other metadata associated with a Stock Asset for any (1) machine learning or artificial intelligence purposes; or (2) technologies designed or intended for the identification of natural persons.

5. Additional Restrictions. In addition to the restrictions in section 4 above:

A. Standard License Restrictions. For Standard Licenses, Customer may not, in the aggregate, (1) cause or allow any Work to appear on more than 500,000 printed materials (including copies); and/or (2) incorporate the Work into a performance, broadcast or digital production if the audience is expected to be greater than 500,000 viewers. This restriction does not apply to Works that are only displayed on websites, social media sites or mobile applications. Unless Customer has an Extended License for a Work, Customer may only distribute the Work as incorporated into an item of merchandise if (1) the Work has been modified to the extent that the new work, as incorporated into such merchandise, is not substantially

similar to the Work and can qualify as an original work of authorship; or (2) the primary value of such merchandise does not lie with the Work itself. For clarification, under the Standard License, producing an unmodified Work on a poster intended for resale or distribution is not permitted, as the primary value would lie in the Work itself.

B. Internal Use License Restrictions. Unless Customer has an Extended License for a Work, Customer may only use, reproduce, archive, modify and display the Work, in all media, for Customer's internal display purposes only. Customer may not use the Work in any way that allows for any non-User, including Customer's clients, to use, download, extract, view, or access the Work.

C. Digital Use License Restrictions. Unless a Customer has an Extended License for a Work, Customer may not (1) cause or allow any Work to appear on any printed materials; or (2) incorporate the Work into merchandise, including but not limited to digital merchandise, intended for sale or distribution, including on-demand products.

D. Social Media Use License Restrictions. Unless a Customer has an Extended License for a Work, Customer may use, reproduce, modify, post and upload the Work only on social media platforms.

E. Editorial Use Restrictions. For Works designated as "editorial use only" in the On-demand Services, Customer may only use those Works (A) in a manner that maintains the editorial context and meaning of the Work; (B) in relation to events or topics that are newsworthy or of general interest to the public; and (C) in compliance with any additional third party licensor restrictions displayed on the Adobe Stock website in the details panel of such Work. Customer must not (1) use these Works for any commercial (i.e., promotional, advertorial or merchandising) purpose; or (2) modify these Works, except for minor adjustments for technical quality or slight cropping or resizing. Customer must include the accompanying credit line or attribution, placed in a way that is reasonable to the applicable use, in this format: "Contributor Name/stock.adobe.com", or as designated on the Adobe Stock website. If Customer wishes to use an editorial Work for a commercial purpose, Customer must (i) obtain a license directly from the copyright owner of the Work; and (ii) secure additional permissions as necessary.

F. Audio Works Restrictions. In addition to the general restrictions in section 4 above, Customer must not (A) use any Audio Work in a way that alters its fundamental character, such as to create remixes or mashups; or make any other alterations for the purpose of creating new music; (B) use any Audio Work as a theme song in a Project type listed in section 3 (C) above; (C) incorporate any Audio Work into any electronic template or design template application (e.g., a web design or presentation template, or templates for electronic greeting cards or business cards); (D) offer any Audio Work on a stand-alone basis or as a listening-only experience, such as a track on an album; or (E) upload or make available any Audio Work as combined solely with a still image or simple one-shot video on a streaming platform (e.g. creating a playlist by using an Audio Work in combination with a visual element, where the visual element adds little to no value).

6. Attribution Requirements. If the Stock Asset is used in (A) a context where any other stock content provider receives attribution, Customer also must include substantially similar attribution for Adobe Stock; and (B) an audiovisual production, Customer must use commercially reasonable efforts to include attribution for Adobe Stock in accordance with industry standards, and where possible in the following format: (1) for Works: "[Contributor Name]/stock.adobe.com"; and (2) for Audio Works: "[Song Title] performed by [Artist's Name]/via Adobe Stock".

7. Special Terms for CC Pro Images.

A. "Pro Images" means only those photographs, illustrations, 3D assets, templates, and vectors designated as "Standard Content" at stock.adobe.com (in the Price filter section) that Customer may license as part of CC Pro or Adobe Stock Pro. For clarity, an asset type is not a "Pro Image" unless expressly listed in this definition.

B. Pro Images are licensed to Customer under an Extended License. The Extended License is perpetual for the specific uses of the Pro Images that Customer has used prior to the end of the License Term, including the Grace Period as applicable. The License Term will be deemed to continue through any seamless extensions or renewals of the applicable Sales Order. Only Users licensed for CC Pro or Adobe Stock Pro may use the Adobe Stock APIs to access Pro Images.

C. Upon the earlier of the termination or expiration of the License Term or Agreement, Customer will have a 30-day grace period ("Grace Period") to use any Pro Image downloaded and paid for prior to such expiration or termination, and any Pro Image used in a project remains subject to the terms of this Agreement. Any Pro Image downloaded and paid for by Customer prior to expiration or termination, but not used before the end of the Grace Period, is not considered licensed. Customer may not download Pro Images during the Grace Period. Other than as permitted during the Grace Period, Customer may not use a Pro Image for the first time, or in a new context (such as on new or different merchandise), after the expiration or termination. Immediately after the Grace Period ends, Customer must delete all unused Pro Images.

The authors apply a computational method called “topic modeling” to text transcribed from instructional videos (on two topics in physics). The result is a quantitative representation of the content of each moment of the video in the form of vectors embedded in a multidimensional space. They then represent the content of questions about these lectures in the same space. They report that this data-driven modeling approach (1) can match questions to their appropriate lectures, and even to the specific section of the lecture which covers the relevant content; (2) can capture how the knowledge of human learners evolves from pre-lecture to post-lecture; (3) estimate whether a learner will answer a question correctly or not based on their answers to other, related questions; (4) estimate the learner’s knowledge of parts of the embedding space that were not explicitly covered by the lectures (or questions).

Whereas I find the overall approach interesting, I do not believe this work, in its current form, merits publication in this journal. The main reason is that I think there is a discrepancy between the claims that the authors make and the data they present. Specifically:

1. Most generally, throughout the paper different research goals are mentioned, and I found it somewhat difficult to discern what exactly the authors were aiming to do. The paper would benefit from clarifying the research goals. Of course, a single paper can achieve multiple goals, but in such a case it would be useful to state all of them together.
 - 1.1. The authors might be trying to come up with a new tool that would be applicable to educators. If application is one of the goals, I am not sure why the authors are not using available technology that can be more quickly adopted, such as large language models (LLMs) like GPT-2. Activity vectors from hidden layers in such models are available, so text excerpts from lectures and questions could be all embedded in the space of hidden activity, and it’s plausible that those embeddings would be just as good as those obtained with the topic model. Moreover, such a model would not have to be re-trained for every new domain (whereas the author’s model would likely have to be re-trained on topics that are far from physics, such as literature, history, political science, computer science, etc.). One might argue that LLM representations are uninterpretable, in contrast to the “topics” (dimensions) of a topic model, but the current paper seems to have no interest in interpreting those “topics” anyway (they are not used in any analysis), so that argument is irrelevant. In fact, one might even imagine doing away with hidden activations altogether, and just giving ChatGPT a lecture text and a question, asking it to identify which part of the lecture is referred to by a question, and using the number of utterances between this part of the lecture and any other part to conduct similar analyses to those proposed by the authors (e.g., constructing knowledge maps). If the goal is to propose scalable tools for wide adoption, it appears that testing such models to see whether they could actually work is an important first step before proposing methods like those advanced by the authors.
 - 1.2. To follow up on the issue of technology: ways to evaluate the specific knowledge of students already exist. For instance, some instructors give specific feedback on specific questions in online quizzes; I personally weigh different questions based on whether they require knowing things that were explicitly stated in class vs. near-transfer to new examples vs. far-transfer that also requires combination of knowledge across different lectures; and, more generally, there is a vast literature spanning 2-3 decades on the importance of specific feedback to students. The question is why these methods are not more widely adopted. If the reason is related to scalability / time / effort, it is currently

unclear whether the authors' method is better suited for wide adoption compared to using something like ChatGPT (I am by no means a fan of ChatGPT, but the fact remains that it is overall surprisingly good at tasks that are not dissimilar to what the authors are trying to do).

- 1.3. The authors might also be interested in providing a computational model that tells us something about human psychology (e.g., learning processes). In this case, the representational assumptions that the model makes require some discussion. On the one hand, it would be important to compare the model to “baseline” or “control” models in order to claim that the current model is useful (compared to, e.g., averaging word2vec embeddings of words in each lecture sentence or question and using representations; this might be what the authors mean by “superficial matching?”); on the other hand, it would be important to discuss claims that embeddings models are a non-starter for complex knowledge representation because they do not have any explicit notion of causal schemas like “knowledge graphs” / “concept maps” / “theory theory”, etc. In a sense, the authors' model might occupy an “uncanny valley”, being too complicated compared to simpler models that can achieve the same empirical results without claiming any psychological reality, but not structured enough to be a quantitative theory of human learning.
- 1.4. The authors might be interested in providing a quantitative measure of how knowledge “behaves”. In this case, I think the authors' claims are over stated, especially with regard to “high-resolution insights”, as I discuss in (2) below.
2. The authors claim that they provide “nuanced insights into what learners know and how their knowledge changes with training”. I do not believe this is the case.
 - 2.1. Nearly all analyses yield the same type of knowledge that would be obtained from calculating “proportion correct” separately for each subset of questions (Birth of Stars, Four Fundamental Forces, general physics). Namely, students do not know any topics before a lecture; after lecture #1, they learn the content of that lecture but not unrelated content from lecture #2 which they haven't yet watched; and after viewing lecture #2, they also learn its content. Despite the authors' claims for higher-resolution quantification of how knowledge evolves throughout a lecture, their analyses in Figures 3, 5, 6, and 7 only provide evidence for this general pattern. For instance, we can compute % correct on 4 questions about Birth of Stars to predict the correctness of the 5th question; would this analysis be significantly inferior to the one proposed by the authors? Overall, my impression is that the authors do not quantitatively evaluate any pattern that is more nuanced than “knowledge of lecture #1 and knowledge of lecture #2 can be separated”.
 - 2.2. The authors claim: “*we show that our approach can automatically match the conceptual knowledge probed by individual quiz questions to the corresponding moments in lecture videos when those concepts were presented*”. This is inaccurate. First, the authors show that their method can match quiz questions to the overall lecture they correspond to (Figure 3B), but this is only a sanity check (as the authors acknowledge), not a technological advancement – instructors who write questions can tag them for which content they are testing, and we hardly need a computational tool to achieve this goal; in fact, this appears to be what instructors mentally do when trying to come up with questions to test different areas of content, and what the authors of this manuscript themselves did when creating questions for lecture #1 vs. #2. Second, the authors report qualitative impressions that each

question can be further matched to a particular timeframe within a lecture (Figure 4), but I do not believe they have provided any test of this impression (my apologies if I missed it!). For instance, if instructors are given the questions and are asked to match them to specific timeframes within the lecture – do they provide the same estimates? (Or, if they are given 3 timeframes identified by the model, can they tell which one corresponds to a given question?). If a student missed a question and is referred back to a particular timeframe of the lecture to re-watch, do they do better than a student referred to a different part of the lecture? Here, too, I wonder: how useful is the “high resolution” match between question and content, beyond what instructors do anyway when compose questions? (e.g., I know exactly which slides are pertinent to each question on my exams, and it’s possible ChatGPT would be able to match questions to the relevant part of a lecture).

- 2.3. The authors claim: “*we demonstrate how we can estimate moment-by-moment “knowledge traces” that reflect the degree of knowledge participants have about each video’s time-varying content, and capture temporally specific increases in knowledge after viewing each lecture*”: this claim has the same problems detailed in 2.2; namely, all quantitative analyses in the paper focus on differences in knowledge between pre- to post-lecture, not on demonstrating that the time-varying trace itself captures anything meaningful.
- 2.4. The authors claim: “*these knowledge estimates can generalize to held-out questions*”: there is not test demonstrating that this analysis (Figure 6) does anything more than capturing low-resolution differences between entire topics. Intriguingly, the results of Quiz 3 suggests that this might be the case: contra to the authors’ claim, Quiz 2 is not the most sensitive test, because passing it merely requires distinguishing between questions about lecture #1 from questions about lecture #2, which is a very low-resolution test (students haven’t learned content that they haven’t been exposed to); Quiz 3 is the one that requires the highest sensitivity, assuming that students know some parts of each lecture better than other parts of that same lecture, and that there are individual differences in which parts are more vs. less understood. Such a case would allow to test the authors’ claim that knowing *which* questions in a given lecture a student has answered correctly (rather than an overall estimate of a student’s score on that lecture) can predict *which* other questions from that lecture they will answer correctly. If the assumption of variations in knowledge throughout a lecture and across students does not hold and, instead, there is a “ceiling effect” as the authors claim (all students know all sections of both lectures very well), then the design of the study is inappropriate for testing high-resolution knowledge.
- 2.5. The authors claim: “*visual maps that provide snapshot estimates of how much participants know about any concept within the scope of our text embedding model, and how much their knowledge of those concepts changes with training*”: again, their analyses seem to capture differences *between* two distinct lectures, not across moments / topics *within* a lecture.
- 2.6. The authors claim: “*our work suggests a rich new line of questions about the geometric “form” of knowledge*”: again, the low-resolution nature of the authors’ results seem quite far from studying the geometric form of knowledge, compared to existing lines of work (both recent ones, and ones that date back 20 years). For instance, some studies have focused on the intrinsic dimensionality of content manifolds (e.g., “Intrinsic Dimension Estimation for Robust Detection of AI-Generated Texts”), or on the linear geometry of word embeddings and how it reflects structured world knowledge (e.g., “How do blind

people know that blue is cold? Distributional semantics encode color-adjective associations”). Other models have explored how the geometry of knowledge evolves over time (e.g., Jay McClelland’s early work on connectionist models of semantic knowledge; or papers like “Structured Semantic Knowledge Can Emerge Automatically from Predicting Word Sequences in Child-Directed Speech”).

- 2.7. One way to reframe what the authors are doing is to treat the two lectures as smaller sections within a long lecture in an academic course (e.g., a 90 minute class). In this case, distinguishing between the two lectures is akin to breaking an entire lecture into smaller (“higher-resolution”) parts. However, I do not believe this is how the authors conceive of their work, because if that was the case, they would not have visualizations like those in Figure 4 or 5A, which are about second-to-second changes within a 10-minute lecture.
3. Some minor comments are below. Some of these are suggestions, whereas others are thoughts that I jotted down as I was reading the manuscript and would perhaps be interesting or useful to the authors, but are equally likely to be irrelevant musings.
 - 3.1. Introduction, line 60: *“These models consider not only the co-occurrences of those elements within and across documents, but also patterns in how those elements appear across different scales (e.g., sentences, paragraphs, chapters, etc.), the temporal and grammatical properties of the elements, and other high-level characteristics of how they are used”*. I think this claim could benefit from some clarifications. For instance, doesn’t LDA treat documents as “bags of words”, looking only at co-occurrences? (The researcher can cut down a document, as is done here, but this is not a feature of the text embedding model – the model treats whatever “unit” of text it receives as a bag). In static word embeddings, co-occurrences are limited to local ones (e.g., a context window of 10 words), so there is not notion of paragraph / chapter, so there is more or less a single temporal scale being modeled. And in contemporary LLMs, some of the more “complex” features, such as grammatical features, are implicitly inferred *from* co-occurrences (there is no separate mechanism for learning these properties; a simplified form of them even arises in GloVe or word2vec).
 - 3.2. Line 63: *“A model that succeeds at capturing an analogue of “understanding” is able to assign nearby feature vectors to two conceptually related documents, even when the specific words contained in those documents have very little overlap.”* It seems to me that this task is very easy for LLMs (they can learn which words mean similar things based on distributional patterns). The harder thing is to separate documents with very similar words that talk about conceptually very different things (e.g., distinguishing between statements about A causing B vs. A correlating with B due to a shared cause vs. B causing A...).
 - 3.3. Line 72: *“For example, understanding the concept of a fish swimming in water first requires understanding what fish and water are.”* From the point of view of conceptual role semantics (or “theory theory”), understanding the relationship between fish and water is precisely what is important for learning what those terms are (rather than some “dictionary definition” of their inherent properties). See, for instance, <https://arxiv.org/abs/2208.02957>.

- 3.4. Figure 5A: the traces look almost like mirror images of one another. What is the reason? Is this caused by time-points which are vs. are not covered by questions?
- 3.5. Line 327: could the authors please say more about how this localization is non-trivial? What is shown is that the topic model can identify which broad topic is addressed by each question (i.e., not which part of a lecture, but merely which lecture). Therefore, is this finding not fully predicted by Figure 1C? In what way does it provide new information beyond Figure 1C? Moreover, if the authors simply constructed a Voronoi diagram where each point in space is assigned to its nearest question, would the result not be highly similar? (I am not saying that it is, but this is a much simpler method and, in a sense, a discretization of the authors' formula).
- 3.6. Choosing number of topics: did the authors examine the perplexity or coherence of the topics? These measures (especially coherence) appear to be accepted metrics for choosing the number of topics (see, e.g., <https://aclanthology.org/D11-1024.Pdf>, <https://aclanthology.org/N10-1012.pdf>).

The authors engaged very thoughtfully, and in great detail, with the comments of all reviewers, including my own. I sincerely thank them for taking the comments seriously, for clarifying misunderstandings on my end, for adding several very helpful analyses, and for expanding the manuscript. It is important to acknowledge the significant amount of work that went into this revision.

My two main remaining concerns were expressed in my previous comments: the model's performance might not be as good as the manuscript claims (or it lacks a quantitative analysis demonstrating it), and it might not advance us beyond existing methods. Here, I describe how these concerns relate to the revisions that the authors made:

First, the central insights that the authors' method provides about learners' knowledge are about predicting which questions will be answered correctly / incorrectly, based on other questions that have been answered (Figure 6). In my opinion, this could be the most compelling part of the paper. However, it is currently analyzed incorrectly: the Mann-Whitney test requires that observations are independent, whereas the current data are grouped by (1) participant, and (2) question. Any tests that do not account for these groupings are anti-conservative (e.g., do not properly take outliers into account). The appropriate way to analyze grouped data is with linear mixed-effects models (LMEs; in this case, logistic). The authors mention that they chose to avoid such analyses because the overall approach is already relatively complicated. However, this is not a good reason to analyze data incorrectly. LMEs are the standard approach in several fields (e.g., psycholinguistics), and have been so for a while. They are straightforward to implement (1-2 lines of code in R) and interpret. I strongly believe that this analysis should be the one adopted.

Once this analysis is adopted, the authors should build two models: one that predicts accuracy (correct vs. incorrect) on each held out question only based on the % correct of that specific participant on questions from the same lecture; and another that predicts accuracy based on this same % correct measure *in addition* to estimated knowledge. Then, by comparing the two nested models (e.g., with the anova command in R), they could explicitly test whether (or, rather, by how much) the estimated knowledge helps to predict held-out questions *over and above* a simple % correct measure. This analysis would provide a direct evaluation of the predictivity of "estimated knowledge" against a baseline. It will thus add critical information beyond Figure 7. Currently, Figure 7 does not provide such a direct comparison: it shows that the probability of correct answers in the "neighborhood" of a reference question is starkly different depending on whether that reference question was correctly vs. incorrectly answered, but these stark differences do not translate to differences in "estimated knowledge" for each reference question, shown in Figure 6 – as I describe next.

While the results in Figure 6 are statistically significant for the appropriate quizzes/topics, the patterns themselves are a bit puzzling: in many of the analyses, the distribution of estimated knowledge for questions answered *incorrectly* is highly skewed towards 1 (perfect knowledge). Even though this distribution is significantly *less* skewed than the distribution of estimated knowledge for *correctly* answered questions, the "effect size" of the skewness appears large to me. Specifically, between 20% and 50% of these incorrectly answered questions are predicted to have a knowledge of 1. This is quite a problem for any actual application of the authors' method, and

puts a strong limit on the claim that this method provides high-resolution insights about participants' knowledge.

For example, a very rough and inaccurate estimation from the top right panel of Figure 6 suggests that the probability of a correctly answered question having a higher estimated knowledge than an incorrectly answered question is ~ 0.56 (chance is 0.5). This is very low for any practical purpose. I did this by creating a ROC curve that attempts to classify questions as “correct” or “incorrect” based on estimated knowledge and computing the area under the curve (AUC). The AUC also equals the Mann-Whitney U statistic divided by the product: $\text{num_answered_correctly} \times \text{num_answered_incorrectly}$. My sincere apologies if my calculation is way off and I made a mistake somewhere. But if it is in the ballpark, then the “predictivity” of knowledge estimates, while significant, is quite limited. In that case, the claims in the papers should be made less bold.

Other than estimating knowledge on held-out questions, the other insights provided by the authors' method are somewhat limited in my opinion. For example, the maps in Figure 8 only show that students learn the lecture that they've watched (i.e., this is where the “change in knowledge” is apparent). This is a sanity check about the algorithm for creating the maps, but it is in itself not an independent insight, as it is derived from the information in Figure 2 and the analyses reported in the text (shown in Supp. Figure 2): the embedding of each question is closer to the trajectory of the corresponding lecture, and farther from the trajectory of the other lecture. The word clouds in Figure 8c are also a reflection of the properties of the embedding space that were verified earlier in the text. The maps indeed have finer-grained structure, but such structure is currently not quantified and remains interpretable (so these are, at best, untested “hypotheses”, not insights).

(Another potential issue with interpretability is that of the topics themselves: for instance, do the authors observe that the model recovers distinct topics for each of the four fundamental forces, i.e., a topic or few that correspond to gravity, others that correspond to the strong interaction, etc.? If so, saying so explicitly is important to demonstrate the strength of the method. If not, then this is a limitation that should be acknowledged, because it's one of the “minimum” requirements for any representation of the “topics” of that lecture, at least in the intuitive sense of “topics”).

Similarly, the “moment by moment” content mastery is an interesting visualization, but there is no quantification of whether it is accurate. If I understand correctly, this visualization is derived from the information in Figure 4, so it again does not provide any independent insight that can be tested / demonstrated. More importantly, the identification of lecture snippets that correspond to each question, shown in Figure 4, is still unfortunately not quantitatively verified. I have proposed several ways in my previous comments about how the accuracy of this result can be measured, and I think they are relatively straightforward to implement in an online study (or the authors could just ask physics instructors to map each question to the corresponding part of the lecture).

The authors manually inspect the lecture snippets that were automatically identified based on each question, and report that they are overall good (providing all the data in the Supplementary Materials is very helpful!). But this is a subjective impression. For instance, Question 26 is said to correspond to two parts of the lecture, but one of them provides the right answer, whereas the other provides the wrong answer—so giving both snippets to a student who answered the question wrong might cause further confusion (as an anecdote, ChatGPT could answer this question correctly even

without the lecture text, so there is nothing inherently “hard” about the question). Similarly, for Question 7, one of the identified paragraphs is quite irrelevant. For Question 4, the algorithm identifies over 2.5min of text, and this entire text does not contain the right answer (“the weak interaction really applies to very small distances”), perhaps because the word “computer” throws the algorithm off. Same for Question 11, where the identified text does not include anything about the strong force. I hope that these examples demonstrate why I believe that a quantitative estimate of the accuracy of the authors’ method is important.

My second main concern regards the usefulness of the authors’ method with respect to other, existing tools. The authors now provide an analysis of BERT, which is nice and informative, but I would like to still push back on two fronts. The first one is less important: given that BERT represents so many topics, as well as language structure, it is expected that its lecture trajectories would be “squashed”, and that the questions would be represented somewhat separately (e.g., they have different syntax). But this does not necessarily mean that the critical information is not there. For instance, in static word embeddings like Word2Vec, antonyms (“good” vs. “bad”) are located in close proximity, and this “squashing” of opposites could be said to be bad for a representation of meaning; nonetheless, the direction in the embedding space that connects antonyms defines a subspace that is meaningful, such that even words that are quite far from the “good” and “bad” vectors can have their valence estimated based on their position relative to that subspace. This is just one example of how complex knowledge exists, and can be extracted, even when at first glance it appears to lack desired properties. (By the way, similarities in BERT are also known to be influenced by “rogue” dimensions, such that the activations of each artificial unit should be z-scored based on a corpus; for instance, see: <https://arxiv.org/abs/2109.04404>. Personally, I don’t think this is going to make much of a difference; I’m mentioning it just in case the authors use similar analyses for other projects).

Second, and more importantly, BERT is a model that is quite good at linguistic representation, but not very good at reasoning (it is also quite small by today’s standards). It does show some evidence of learning “topics”, similarly to those the authors talk about (for instance, <https://arxiv.org/pdf/2203.14680.pdf>), but it is still quite limited. In contrast, models trained using reinforcement learning from human feedback exhibit much better reasoning abilities (for instance: <https://www.nature.com/articles/s41562-023-01659-w>). It is possible that these are the abilities that are required for, e.g., estimating knowledge based on questions. The ability to identify which span of text is relevant for answering a particular question can already be done with high accuracy by modern models (e.g., <https://paperswithcode.com/sota/question-answering-on-squad20>). Moreover, state-of-the-art, publicly available systems might have some important advantages over the authors’ model.

The authors state in their rebuttal “*suppose one were to ask ChatGPT to match up each question with some part of a lecture. What would one “do” with that response?*”. My answer is: whatever one would do with the information in Figure 4. If the data in that figure are used as a test of the authors’ model, then I am assuming those data have some *use*. For instance, you could refer a student who answered a question incorrectly to the relevant part of the lecture for review. The authors also state “*ChatGPT also has no built-in mechanism for keeping track of what the student knows, or how that knowledge might relate to the content of a course the student is learning from*”; and “*ChatGPT has no internal machinery for representing or tracking the learner’s knowledge,*

nor does it maintain a “theory of mind” of the learner, nor does it (in and of itself, to the best of our own understanding) have any deep understanding of the material itself. The authors’ model also does not have any deep understanding of the material (and, if anything, understands *less* because it does not capture the structural dependencies between words in a text). The model also has no substantial theory of mind: what it does is implement a notion of “knowledge = vectors in a continuous space”, but ChatGPT and similar models rely on the same assumption, and hence implement this same notion. Being a high-dimensional embedding, ChatGPT has the capacity to hold and process all the information that the authors’ model stores; the fact that it is very hard (at least for me) to conceive of how it might do so, does not mean that it cannot do so or does not do so. There is active work on the ability of large language models to have a “situation model” or a “discourse model” and update it as they process more input. Also, professors are using these models to read students’ papers and fill out detailed rubrics with feedback, so these models can convey high-resolution insights about student performance. Models like ChatGPT are also *generative*, so they can go well beyond the authors’ model by, e.g., writing new questions. In this sense, these models could be more *useful* in practice because they can translate insights into actions.

As an anecdotal example, I fed ChatGPT the text of the *Four Fundamental Forces* lecture, followed by some prompts and questions. Below are the first responses that I received in this interaction, unless otherwise specified, so there was no “tweaking” (sure, slightly different prompts might have given different results, but that is beside the point in my opinion; once we know the right prompt to use, there is no reason to use others). .

[the input below was given immediately after the text of the lecture]

Answer the question below:

Why don't you and your computer experience any attraction or repulsion due to the Weak Interaction?

- A. The weak interaction only acts over extremely small distances
- B. The weak interaction between you and your computer is counteracted by the other forces
- C. You and your computer have no net charge
- D. Neither you nor your computer has enough mass to induce a significant Weak Interaction

ChatGPT

- A. The weak interaction only acts over extremely small distances

You

Please type the part of the lecture that helped you answer the question above

ChatGPT

The part of the lecture that helps answer the question is:

"Now, the next strongest force-- and just to give a sense of how weak gravity is even relative to the weak interaction, the weak interaction is 10 to the 25th times the strength of gravity. And you might be saying, if this is so strong, how come this does it operate on planets or us relative to the Earth? Why doesn't this apply to intergalactic distances the way gravity does? And the reason is the weak interaction really applies to very small distances, very, very small distances. So it can be much stronger than gravity, but only over very, very-- and it really only applies on the subatomic scale. You go anything beyond that, it kind of disappears as an actual force, as an actual interaction."

[note that the above is a better than the authors' algorithm, which does not identify the relevant lecture snippet for this question; in addition, the snippet is ~1min long, compared to the author's algorithm which identified ~2.5min of text]

You

A student read the lecture above, and answered the following questions correctly:

1. Why is the gravitational attraction between you and your computer too small for you to notice?
2. Roughly how many times stronger is the Weak Interaction than gravity?
3. Why don't you and your computer experience any attraction or repulsion due to the Weak Interaction?

But that student got the following questions wrong:

1. Why does the universe have a very uneven distribution of mass but a relatively equal distribution of charge?
2. In your body, there are a tremendous amount of negatively-charged electrons. Your computer also contains a huge number of negatively-charged electrons. We know that like charges repel, but you and your computer are not repelled apart. Why?
3. What does the Coulomb Force refer to?

Based on this pattern, which concepts did the student successfully learn, and which concepts did the student not learn well?

[I first tried to get ChatGPT to immediately generate new questions based on the above prompt, instead of asking first which concepts the student learned well; that did not work well, which is why I added this "intermediate" step above]

ChatGPT

The student seems to have a good understanding of the concepts related to gravitational attraction, the relative strengths of the weak interaction compared to gravity, and why the weak interaction is not noticeable between everyday objects. However, there are challenges with concepts related to the distribution of mass and charge in the universe, the behavior of like charges in your body and the computer, and the specific reference to the Coulomb Force.

Concepts the student successfully learned:

1. **Gravitational Attraction:** The student understands that the gravitational attraction between objects like you and your computer is too small to be noticeable in everyday situations.
2. **Weak Interaction Strength:** The student correctly grasps the idea that the weak interaction is much stronger than gravity, but its effects are limited to very small distances.

Concepts the student did not learn well:

1. **Uneven Distribution of Mass and Charge:** The student seems to struggle with the concept that the universe has a very uneven distribution of mass but a relatively equal distribution of charge. This may indicate a misunderstanding of the distribution of mass and charge on a cosmic scale.
2. **Behavior of Like Charges:** The student has difficulty explaining why, despite the abundance of negatively-charged electrons in both your body and the computer, there is no repulsion. This suggests a lack of understanding of the principles of charge distribution and behavior.
3. **Coulomb Force:** The student may not be familiar with or did not grasp the concept of the Coulomb Force, which is related to the electrostatic force between charged particles. This is a crucial aspect of electromagnetism.

To improve understanding, the student might benefit from revisiting these concepts, seeking additional explanations or examples, and practicing related problems to reinforce their comprehension.

You

Given what this specific student seems to know vs. what they do not know, write one new question that you expect them to answer correctly and one new question that you expect them to answer wrong.

ChatGPT

Expected Correct Answer:

Question: Why is the weak interaction more noticeable on a subatomic scale rather than on larger scales like intergalactic distances?

Expected Incorrect Answer:

Question: What is the primary factor responsible for the attractive forces between positively charged protons in an atomic nucleus?

Another example:

You

Here is another question: Which force is stronger than the Electromagnetic Force? Could you please write which part of the lecture would help someone answer that question?

ChatGPT

The force that is stronger than the Electromagnetic Force is the Strong Force. The part of the lecture that provides this information is:

"Now, the strongest of the force is probably the best named of them all. And that's the strong force. That is the strong force... And once again, just to keep it in mind relative to gravity, it is 10 to the 38th times the strength of gravity. Or it's about 100 times stronger than the electromagnetic force."

This part of the lecture compares the strength of the Strong Force to that of the Electromagnetic Force, highlighting that the Strong Force is much stronger than the Electromagnetic Force.

[This is Question 11, for which the authors' algorithm did not provide the correct snippet]

These examples are, of course, anecdotal. I did not conduct a thorough evaluation of ChatGPT. But I am showing to suggest that the authors are perhaps too hasty in dismissing large language models as lacking high-resolution insights because their semantic spaces are so large. These models might be able to do a lot of what the authors' model does, and even go beyond it by, e.g., constructing new questions. True, they cannot provide visualizations of continuous maps of a student's knowledge (Figure 5, 8), but such maps are only interpretable and useful insofar as they lead to some action (e.g., identifying which concepts to focus on for re-learning).

All this brings me back to the concern I expressed in my first review: the authors' algorithm is cool, but does it sufficiently advance us beyond the capabilities that already exist in large language

models, which are available to students and instructors? I think that answering this question is important for deciding whether the paper merits publication in this journal.

Some minor thoughts:

- The word-overlap analysis (Supp. Figure 5) is very helpful!
- Perhaps change the naming of the quizzes to Quiz 0, Quiz 1, and Quiz 2? That way, the number corresponds to the number of lectures the students viewed (I found myself having to do a “minus one” operation every time I was reading something about the quizzes to figure out their position in the experiment). I realize this means re-generating all the figures, so I’m not sure it’s worth it...
- The authors test whether “*knowledge about a given concept implies knowledge about related concepts*” and “*knowledge about a given concept is predictive of knowledge about similar or related concepts*”. It would be helpful to the reader if the authors explicitly articulated which theories from psychology and/or education do not predict this pattern of knowledge (otherwise, this is a test of the validity of the model, not a psychological implication of the findings).
- In Figure 3, and Supp Figure 2, the negative correlation between Lecture A and questions from Lecture B might in part be an “artifact” of the inclusion of only two lectures in the model. The more lectures are included, leading to more diverse topics, the more this pattern might decrease? (It is still expected that questions about a lecture will correlate with that lecture more than with other lectures, but the difference might not be as strong, especially for lectures on related topics).